# Brain-like Variational Inference

**Hadi Vafaii**[1,*]      **Dekel Galor**[1,*]      **Jacob L. Yates**[1]

\* co-first authors,   correspondence to `vafaii@berkeley.edu`

[1]Redwood Center for Theoretical Neuroscience, UC Berkeley

## Abstract

Inference in both brains and machines can be formalized by optimizing a shared objective: maximizing the evidence lower bound (ELBO) in machine learning, or minimizing variational free energy ($\mathcal{F}$) in neuroscience (ELBO $= -\mathcal{F}$). While this equivalence suggests a unifying framework, it leaves open how inference is implemented in neural systems. Here, we introduce FOND (*Free energy Online Natural-gradient Dynamics*), a framework that derives neural inference dynamics from three principles: (1) natural gradients on $\mathcal{F}$, (2) online belief updating, and (3) iterative refinement. We apply FOND to derive i$\mathcal{P}$-VAE (*iterative Poisson variational autoencoder*), a recurrent spiking neural network that performs variational inference through membrane potential dynamics, replacing amortized encoders with iterative inference updates. Theoretically, i$\mathcal{P}$-VAE yields several desirable features such as emergent normalization via lateral competition, and hardware-efficient integer spike count representations. Empirically, i$\mathcal{P}$-VAE outperforms both standard VAEs and Gaussian-based predictive coding models in sparsity, reconstruction, and biological plausibility, and scales to complex color image datasets such as CelebA. i$\mathcal{P}$-VAE also exhibits strong generalization to out-of-distribution inputs, exceeding hybrid iterative-amortized VAEs. These results demonstrate how deriving inference algorithms from first principles can yield concrete architectures that are simultaneously biologically plausible and empirically effective.

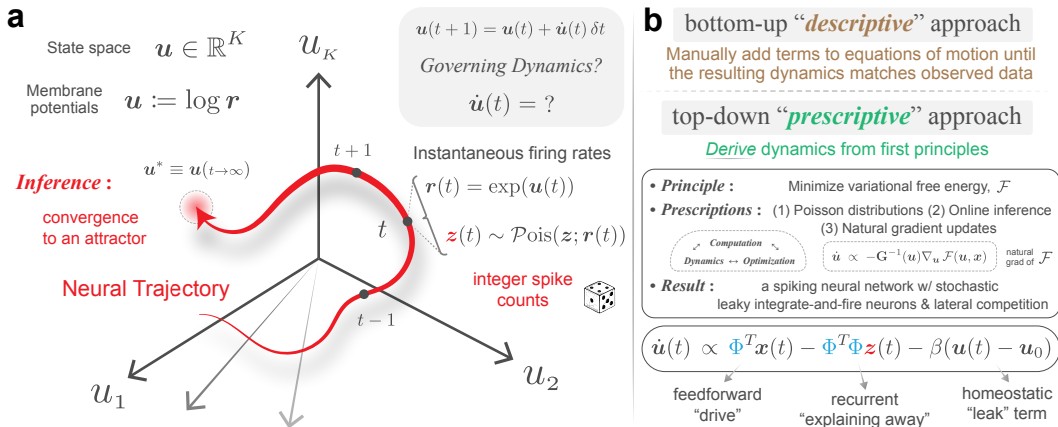

Figure 1: Inferential and dynamical accounts of perception are unified under variational inference. **(a)** Perception is framed as a dynamical process of convergence to attractors in a neural state space, where membrane potentials evolve and generate spikes along the way. **(b)** Our prescriptive approach derives neural dynamics by minimizing free energy via natural gradient descent, yielding a spiking network with lateral competition. The resulting architectures are principled and empirically effective. Code, data, and model checkpoints are available here: https://github.com/hadivafaii/IterativeVAE

39th Conference on Neural Information Processing Systems (NeurIPS 2025).

# 1 Introduction

Artificial intelligence and theoretical neuroscience share a foundational principle: understanding the world requires inferring hidden causes behind sensory observations (early work: [1–3]; modern interpretations: [4–9]). This principle is formalized through Bayesian posterior inference [10–12], with *variational inference* [4, 13–17] offering an optimization-based approximation. Variational inference employs an identical objective function across both fields, albeit under different names. In machine learning, it is called *evidence lower bound* (ELBO), which forms the basis of variational autoencoders (VAEs; [18–21]); whereas, neuroscience refers to it as *variational free energy* ($\mathcal{F}$), which subsumes predictive coding [22, 23] and is presented as a unified theory of brain function [24]. ELBO is exactly equal to negative free energy (ELBO $= -\mathcal{F}$; [25]), and this mathematical equivalence offers a powerful common language for describing both artificial and biological intelligence [26].

Despite this promising connection, neither ELBO nor $\mathcal{F}$ have proven particularly prescriptive for developing specific algorithms or architectures [27, 28]. Instead, the pattern often works in reverse: effective methods are discovered empirically, then later recognized as instances of these principles. In machine learning, diffusion models were originally motivated by non-equilibrium statistical mechanics [29], and only later understood as a form of ELBO maximization [30, 31]. Similarly, in neuroscience, *predictive coding* (PC; [22]) was first proposed as a heuristic model for minimizing prediction errors [32], before being reinterpreted as $\mathcal{F}$ minimization [23, 33]. This pattern of post-hoc theoretical justification, rather than theory-driven derivation, is common across both disciplines.

Then why pursue a general theory at all? A unifying framework offers several advantages: it clarifies the design choices underlying successful models, explains why they work, and guides the development of new models. The recent *Bayesian learning rule* (BLR; [34]) exemplifies this potential. It re-interprets a broad class of learning algorithms as instances of variational inference, optimized via natural gradient descent [35] over approximate posteriors $q_{\boldsymbol{\lambda}}$. By varying the form of $q_{\boldsymbol{\lambda}}$ and the associated approximations, BLR not only recovers a wide range of algorithms—from SGD and Adam to Dropout—but, crucially, it also offers a principled recipe for designing new ones [34]. Thus, BLR transforms a post-hoc theoretical justification into a prescriptive framework for algorithm design.

In variational inference, the prescriptive choices are specifying $q_{\boldsymbol{\lambda}}$ and how to optimize the variational parameters, $\boldsymbol{\lambda}$. The machine learning community has embraced *amortized inference* [18, 19, 36–38], where a neural network is trained to produce $\boldsymbol{\lambda}$ in a single forward pass for each input sample. In contrast, neuroscience models have traditionally favored *iterative inference* methods [22, 39], which align more naturally with the recurrent [40–44] and adaptive [45–49] processing observed in cortical circuits (but see Gershman and Goodman [36] for a counterpoint). Despite this fundamental divide in methodology, systematic comparisons between iterative and amortized inference variants remain scarce, particularly when evaluated against biologically relevant metrics.

In this paper, we introduce *Free energy Online Natural-gradient Dynamics* (**FOND**), a framework for deriving brain-like inference dynamics from $\mathcal{F}$ minimization. As a concrete application of FOND, we derive a novel family of iterative VAE architectures that perform inference via natural gradient descent on $\mathcal{F}$—including a spiking variant, the *iterative Poisson VAE* (**i$\mathcal{P}$-VAE**)—with distinct computational and biological advantages. The paper is organized as follows:

- In section 2 and the associated appendix A, we review and synthesize existing models from machine learning and neuroscience, showing how they can be unified under the variational inference framework via specific choices of distributions and inference methods (Fig. 2).

- In sections 3 and 4, and the associated appendix B, we derive neural dynamics as natural gradient descent on free energy. This yields a novel family of iterative VAEs, including the i$\mathcal{P}$-VAE, which performs online Bayesian inference through membrane potential dynamics, with emergent lateral competition, explaining away dynamics, and divisive normalization.

- In section 5, we show that iterative inference consistently outperforms amortized methods, with i$\mathcal{P}$-VAE achieving the best reconstruction-sparsity trade-off, while using $25\times$ fewer parameters and integer-valued spike counts. Appendix C shows that i$\mathcal{P}$-VAE: (i) exhibits emergent cortical response properties, (ii) generalizes better to out-of-distribution data, and (iii) scales to complex color image datasets such as ImageNet32, CIFAR-10, and CelebA.

- In section 6, and the associated appendix D, we relate our results to recent advances in sequence modeling, and discuss how expressive nonlinearities, emergent normalization, and effective stochastic depth contribute to i$\mathcal{P}$-VAE's computational strengths.

## 2   Background and Related Work

**Notation and conventions.**   We start with a generative model that assigns probabilistic beliefs to observations, $\boldsymbol{x} \in \mathbb{R}^M$ (e.g., images), through invoking $K$-dimensional latent variables, $\boldsymbol{z}$: $p_\theta(\boldsymbol{x}) = \int p_\theta(\boldsymbol{x}, \boldsymbol{z}) \, d\boldsymbol{z} = \int p_\theta(\boldsymbol{x}|\boldsymbol{z}) p_\theta(\boldsymbol{z}) \, d\boldsymbol{z}$, where $\theta$ are its adaptable parameters (e.g., neural network weights or synaptic connection strengths in the brain). Throughout this work, we color-code the generative and inference model components as blue and red, respectively.

**Variational inference and the ELBO objective.**   Following the *perception-as-inference* framework [1, 2], we formalize perception as Bayesian posterior inference. In appendix A.1, we provide a historical background, and in appendix A.2, we review the challenges associated with posterior inference. We overcome these challenges by approximating the true but often intractable posterior, $p_\theta(\boldsymbol{z}|\boldsymbol{x})$, using another distribution, $q_\phi(\boldsymbol{z}|\boldsymbol{x})$, referred to as the *approximate* (or *variational*) posterior. In appendix A.3, we provide more details about how *variational inference* [15, 17] approximates the inference process, and derive the standard Evidence Lower BOund (ELBO) objective [13, 15, 17]:

$$\underbrace{\log p_\theta(\boldsymbol{x})}_{\text{model evidence}} = \underbrace{\mathbb{E}_{\boldsymbol{z} \sim q_\phi(\boldsymbol{z}|\boldsymbol{x})}\left[\log \frac{p_\theta(\boldsymbol{x}, \boldsymbol{z})}{q_\phi(\boldsymbol{z}|\boldsymbol{x})}\right]}_{\text{ELBO}(\boldsymbol{x};\theta,\phi)} + \underbrace{\mathcal{D}_{\text{KL}}\Big(q_\phi(\boldsymbol{z}|\boldsymbol{x}) \, \| \, p_\theta(\boldsymbol{z}|\boldsymbol{x})\Big)}_{\text{Kullback-Leibler (KL) divergence}}. \tag{1}$$

Importantly, maximizing the ELBO minimizes the intractable Kullback-Leibler (KL) divergence between the approximate and true posterior. This can be seen by taking gradients of eq. (1) with respect to $\phi$. The model evidence on the left-hand side does not depend on $\phi$, making the gradients of the two terms on the right additive inverses. Thus, ELBO maximization directly enhances the quality of posterior inference. In theoretical neuroscience, the negative ELBO is referred to as variational free energy ($\mathcal{F}$), which is the mathematical quantity central to the *free energy principle* [24, 25].

In the remainder of this section, we demonstrate how diverse models across machine learning and neuroscience emerge as instances of $\mathcal{F}$ minimization through two fundamental design choices:

> **1. Choice of distributions** (appendix A.4):
> (i) approximate posterior $q_\phi(\boldsymbol{z}|\boldsymbol{x})$, (ii) prior $p_\theta(\boldsymbol{z})$, and (iii) likelihood $p_\theta(\boldsymbol{x}|\boldsymbol{z})$
>
> **2. Choice of inference method** (appendix A.8):
> (i) *amortized* (e.g., learned neural network)  vs.  (ii) *iterative* (e.g., gradient descent)

**Variational Autoencoder (VAE) model family.**   Variational Autoencoders (VAEs) transform the abstract ELBO objective into practical deep learning architectures [18–20]. The standard Gaussian VAE ($\mathcal{G}$-VAE) exemplifies this approach by assuming factorized Gaussian distributions for all three distributions, with the approximate posterior $q_\phi(\boldsymbol{z}|\boldsymbol{x})$ implemented as a neural network that maps each input $\boldsymbol{x}$ to posterior parameters: $\text{enc}(\boldsymbol{x}; \phi) \to (\boldsymbol{\mu}(\boldsymbol{x}), \boldsymbol{\sigma}^2(\boldsymbol{x}))$. This *amortization* of inference—using a single network to approximate posteriors across the entire dataset—is a defining characteristic of VAEs. Alternative distribution choices are also possible; for instance, replacing both prior and posterior with Poisson distributions yields the Poisson VAE ($\mathcal{P}$-VAE; [50]), which better aligns with neural spike-count statistics [51–53]. We derive the VAE loss in appendix A.5, and discuss both $\mathcal{G}$-VAE and $\mathcal{P}$-VAE extensively in appendices A.6 and A.7.

**Sparse coding and predictive coding as variational inference.**   Two major cornerstones of theoretical neuroscience, sparse coding (SC; [54]) and predictive coding (PC; [22]), can also be derived as instances of ELBO maximization (or equivalently, $\mathcal{F}$ minimization), given specific distributional choices [33, 55, 56]. SC and PC share two key characteristics that distinguish them from standard VAEs. First, they both use a Dirac-delta distribution for the approximate posterior, effectively collapsing it to a point estimate. But they differ in their prior assumptions: PC employs a Gaussian prior, while SC uses a sparsity-promoting prior (e.g., Laplace; Fig. 2). Second, instead of amortized inference as in VAEs, both PC and SC employ iterative inference, better aligning with the recurrent nature of neural computation [40–44]. See appendix A.8 for an in-depth comparison, and appendix A.9 for a pedagogical derivation of the Rao and Ballard [22] objective as $\mathcal{F}$ minimization.

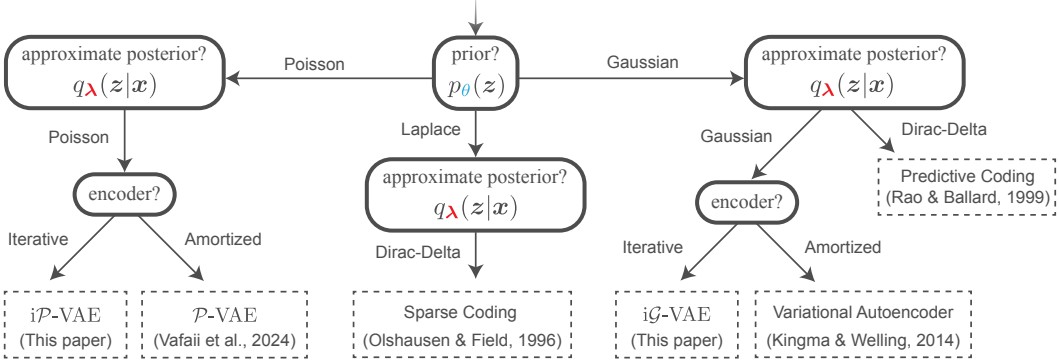

Figure 2: A wide range of models across machine learning and theoretical neuroscience can be unified under free energy ($\mathcal{F}$) minimization. Different distributional and optimization choices result in different models (appendix A.10). Motivated by this unification potential, we introduce **FOND**, a framework for deriving brain-like inference algorithms from first principles (section 3). We apply FOND to derive a family of iterative VAE architectures, including the spiking **i$\mathcal{P}$-VAE** (section 4).

**Online inference through natural gradient descent.** In a recent landmark paper, Khan and Rue [34] proposed the *Bayesian learning rule* (BLR), unifying seemingly disparate learning algorithms as instances of variational inference optimized via natural gradient descent [35, 57] on a Bayesian objective which reduces to the ELBO when the likelihood is known [58, 59]. BLR's generality extends naturally to sequential settings [60], where beliefs must be updated online as data arrives. In such settings, a rolling update scheme—where the posterior at each step becomes the prior for the next (Fig. 7)—provides a simple yet effective approach to continual belief revision [60].

**Summary so far.** In this background section, and the corresponding appendix A, we reviewed how seemingly different models across machine learning and neuroscience can be understood as instances of $\mathcal{F}$ minimization (Fig. 2). The differences among them arise from the choices we make on the distributions and the inference methods (appendix A.10).

## 3 FOND: Free energy Online Natural-gradient Dynamics

Building on the unification potential shown above, we introduce FOND, a framework for deriving inference dynamics from $\mathcal{F}$ minimization. Unlike post-hoc interpretations of existing models, FOND adopts a top-down approach: it starts from theoretical principles and derives architectures, making explicit the connection between design choices and computational properties.

**A two-level framework: flexible choices and fixed prescriptions.** FOND operates at two distinct levels. First, the modeler makes *flexible choices* about distributions and their parameterizations—decisions that define the inference problem. Second, once these choices are made, FOND applies *fixed prescriptions* that fully determine the inference dynamics.

This structure mirrors prescriptive theories in physics [61–64]. To derive dynamics from first principles, one must specify: (a) the dynamical variables of interest, and (b) the equations governing their evolution (see appendix B.1). In FOND, (a) corresponds to flexible choices, while (b) follows from fixed prescriptions.

**Flexible choices — what the modeler decides.** The modeler must choose:

- **Distributions**: Variational family and conditional likelihood (appendix A.4)
- **Parameterization**: Choice of dynamical variables that parameterize the variational distribution

These modeling decisions determine *what* is being optimized, but not *how* optimization proceeds.

**Fixed prescriptions — what FOND determines.** Once the flexible choices are made, the inference dynamics are *fully determined* by FOND's three core prescriptions:

> **FOND prescriptions**
>
> **(1)** **Natural gradients**: Dynamics follow natural gradients of $\mathcal{F}$
> **(2)** **Online**: Inference updates beliefs sequentially from streaming data
> **(3)** **Iterative**: Inference refines estimates through recurrent updates

**Why natural gradients?** Natural gradient descent follows the steepest descent path in the space of probability distributions [35], achieving maximal free energy reduction per unit of distance (measured via Fisher information metric [57, 65]). This provides the most efficient optimization trajectory.

**Why online?** Biological perception requires adaptive priors that evolve with experience. Static inference cannot account for phenomena like *serial dependence*, where recent stimuli systematically bias current perception [46], or *learned helplessness*, where beliefs about agency are updated by experience [66, 67]. Online inference enables time-evolving priors that capture such effects (Fig. 7).

**Why iterative?** Perception follows an "analysis-by-synthesis" loop [68]: hypotheses generate predictions (synthesis), errors refine hypotheses (analysis). Iterative inference implements this closed-loop error-correction, adapting estimates when initial predictions fail. Amortized inference, by contrast, commits to a single forward pass—an open-loop guess that cannot recover when it errs, making it brittle on novel or ambiguous inputs. The brain appears to implement this kind of iterative refinement: macaque visual cortex requires an additional $\sim 30\text{ms}$ of recurrent processing to recognize challenging images [42], and neural codes for faces dynamically evolve over hundreds of milliseconds—a process termed "code switching" [44]. Thus, iterative refinement is motivated both computationally (enabling error correction) and empirically (matching neural dynamics).

**Demonstrating FOND: iterative VAE architectures.** To demonstrate FOND's utility, we derive three iterative VAE models by making different flexible choices while applying the same fixed prescriptions:

- $i\mathcal{P}$-VAE: Poisson distributions with log-rate parameterization

- $i\mathcal{G}$-VAE: Gaussian distributions with mean and log-std parameterization

- $i\mathcal{G}_{\varphi}$-VAE: like Gaussian, but with post-sampling nonlinearity $\varphi(\cdot)$

See Table 1 for complete specifications. In what follows, we present the theoretical derivations, evaluate empirical performance, and discuss implications for neuroscience and machine learning.

## 4 $i\mathcal{P}$-VAE: Deriving a Spiking Inference Model from Variational Principles

In this section, we apply the FOND framework to derive brain-like inference dynamics from first principles. We begin by identifying the key prescriptive choices required for deriving these dynamics. We then introduce Poisson assumptions and derive membrane potential updates as natural gradient descent on variational free energy ($\mathcal{F}$), leading to the iterative Poisson VAE ($i\mathcal{P}$-VAE) architecture.

**Three distributional choices.** To fully specify $\mathcal{F}$, we must choose an approximate posterior, a prior, and a conditional likelihood. We use Poisson distributions for both posterior and prior, as it leads to more brain-like integer-valued spike count representations (see appendix B.2 for a discussion). Later in the appendix, we show how this derivation extends to Gaussian posteriors with optional nonlinearities applied after sampling, illustrating the generality and flexibility of FOND.

For the likelihood, we assume factorized Gaussians throughout this work (a standard modeling choice in both sparse coding [54] and predictive coding [22]). In the main text, for pedagogical clarity, we focus on linear decoders, $p_{\theta}(\boldsymbol{x}|\boldsymbol{z}) = \mathcal{N}(\boldsymbol{x}; \Phi\boldsymbol{z}, \mathbf{I})$, where $\Phi$ is commonly referred to as the *dictionary* in sparse coding literature. In the appendix, we show how our results naturally extend to nonlinear decoders with learned variance [69], and we explore these extensions both theoretically and empirically.

**Parameterization choices.** For a neurally plausible model, we choose the dynamic variables to be real-valued membrane potentials, $\boldsymbol{u} \in \mathbb{R}^K$, where $K$ is the number of neurons. To generate spike counts from $\boldsymbol{u}$, we assume the canonical Poisson parameterization, defining firing rates as $\mathrm{r} := \exp(\boldsymbol{u})$ (Fig. 1a). This choice is both mathematically convenient, since it avoids constrained optimization over a strictly positive variable, and biologically motivated, as the spiking threshold of real neurons is well-approximated by expansive nonlinearities [70, 71].

Under the assumption that dynamics, computation, and optimization are three expressions of the same underlying *process* [72–75] (i.e., *inference $\Leftrightarrow$ dynamics*), membrane potentials evolve to minimize variational free energy ($\mathcal{F}$). To respect the curved geometry of distribution space, this optimization is implemented via natural gradient descent [34, 57, 76] (Fig. 1b).

**Poisson variational free energy.** Given a Poisson posterior and prior, and a factorized Gaussian likelihood with a linear decoder, the free energy (negative ELBO from eq. (1)) takes the form:

$$\mathcal{F}(\boldsymbol{x}; \Phi, \boldsymbol{u}_0, \boldsymbol{u}) = \underbrace{\mathbb{E}_{\boldsymbol{z} \sim q(\boldsymbol{z}|\boldsymbol{x})} \left[ \frac{1}{2} \|\boldsymbol{x} - \Phi\boldsymbol{z}\|_2^2 \right]}_{\mathcal{L}_{\text{recon.}}: \text{ reconstruction term } (\textit{distortion})} + \underbrace{\beta \sum_{i=1}^K \left( e^{\boldsymbol{u}} \odot (\boldsymbol{u} - \boldsymbol{u}_0) - (e^{\boldsymbol{u}} - e^{\boldsymbol{u}_0}) \right)_i}_{\mathcal{L}_{\text{KL}}: \text{ KL term } (\textit{coding rate})}, \quad (2)$$

where $\boldsymbol{u}_0 \in \mathbb{R}^K$ and $\boldsymbol{u} \in \mathbb{R}^K$ are the prior and posterior membrane potentials, $K$ is the latent dimensionality, $\odot$ represents the element-wise (Hadamard) product, and $\beta$ is a positive coefficient that controls the *rate-distortion* trade-off [77, 78]. See appendix B.3 for a detailed derivation.

Next, we compute the gradient of $\mathcal{F}$ with respect to the variational parameters, i.e., $\nabla_{\boldsymbol{u}} \mathcal{F}$.

**Free energy gradient: the reconstruction term.** $\mathcal{L}_{\text{recon.}}$ is the expected value of the prediction errors under the approximate posterior. Since this expectation is intractable in general, we follow standard practice in variational inference and approximate it using a single Monte Carlo sample [34, 60]: $\mathcal{L}_{\text{recon.}} \approx \frac{1}{2} \|\boldsymbol{x} - \Phi\boldsymbol{z}(\boldsymbol{u})\|_2^2$, where $\boldsymbol{z}(\boldsymbol{u}) \sim q_{\boldsymbol{u}}(\boldsymbol{z}|\boldsymbol{x})$.

Since the reconstruction loss depends on the sample $\boldsymbol{z}$, which is a stochastic function of the firing rate $\mathrm{r} = \exp(\boldsymbol{u})$, we propagate gradients through both the exponential nonlinearity and the sampling process (i.e., $\boldsymbol{u} \to \mathrm{r} \to \boldsymbol{z} \to \mathcal{L}_{\text{recon.}}$). We apply the chain rule twice to get:

$$\nabla_{\boldsymbol{u}} \mathcal{L}_{\text{recon.}} = \frac{\partial \mathrm{r}}{\partial \boldsymbol{u}} \frac{\partial \boldsymbol{z}}{\partial \mathrm{r}} \frac{\partial}{\partial \boldsymbol{z}} \mathcal{L}_{\text{recon.}} \approx e^{\boldsymbol{u}} \odot \frac{\partial \boldsymbol{z}}{\partial \mathrm{r}} \odot \left( -\Phi^T (\boldsymbol{x} - \Phi\boldsymbol{z}(\boldsymbol{u})) \right). \quad (3)$$

We further simplify the expression by applying the straight-through gradient estimator [79], treating $\partial \boldsymbol{z}/\partial \mathrm{r} \approx \mathbf{I}$ for the purpose of deriving inference equations. During model training, however, we update the generative model parameters by utilizing the Poisson reparameterization algorithm [50]. This straight-through approximation of the sample gradient yields:

$$\nabla_{\boldsymbol{u}} \mathcal{L}_{\text{recon.}} \approx e^{\boldsymbol{u}} \odot \left( -\Phi^T (\boldsymbol{x} - \Phi\boldsymbol{z}(\boldsymbol{u})) \right). \quad (4)$$

**Free energy gradient: the KL term.** The KL-term gradient is easily computed from eq. (2): $\nabla_{\boldsymbol{u}} \mathcal{L}_{\text{KL}} = e^{\boldsymbol{u}} \odot (\boldsymbol{u} - \boldsymbol{u}_0)$. Combine it with the reconstruction-term gradient from eq. (4) to get:

$$\nabla_{\boldsymbol{u}} \mathcal{F}(\boldsymbol{x}; \Phi, \boldsymbol{u}_0, \boldsymbol{u}) \approx e^{\boldsymbol{u}} \odot \left[ -\Phi^T (\boldsymbol{x} - \Phi\boldsymbol{z}(\boldsymbol{u})) + \beta(\boldsymbol{u} - \boldsymbol{u}_0) \right]. \quad (5)$$

**Natural gradients: applying Fisher preconditioning.** FOND prescribes that neural dynamics follow natural gradients of the free energy, $\dot{\boldsymbol{u}} := -\mathbf{G}^{-1}(\boldsymbol{u}) \nabla_{\boldsymbol{u}} \mathcal{F}$, where $\mathbf{G}$ is the Fisher information matrix. For Poisson distributions in the canonical form, we have: $\mathbf{G}(\boldsymbol{u}) = \exp(\boldsymbol{u})$ (see appendix B.4 for a derivation). Combining this result with eq. (5) yields:

$$\dot{\boldsymbol{u}} \quad \propto \quad \underbrace{\Phi^T \boldsymbol{x}}_{\text{feedforward "drive"}} \quad - \quad \underbrace{\Phi^T \Phi \, \boldsymbol{z}(\boldsymbol{u})}_{\text{recurrent "explaining away"}} \quad - \quad \underbrace{\beta(\boldsymbol{u} - \boldsymbol{u}_0)}_{\text{homeostatic "leak" term}} \quad (6)$$

This concludes the derivation of our core theoretical result underlying the i$\mathcal{P}$-VAE architecture. Natural gradient descent on $\mathcal{F}$ yields circuit dynamics that share important characteristics with neural models developed since the early 1970s [80–83], while offering a notable biological advantage over

standard predictive coding (PC). Namely, recurrent interactions in i$\mathcal{P}$-VAE (eq. (6)) occur through discrete spikes $\boldsymbol{z}$, rather than continuous membrane potentials as in PC (eq. (16)), better aligning with how real neurons communicate [84]. In appendix B.5, we interpret each term in the dynamics and explain the rationale behind its naming. In appendix B.6, we extend the derivation to nonlinear decoders; and in appendix B.7, we extend to Gaussian posteriors with optional nonlinearities. Finally, appendix B.8 explores this biological distinction with standard PC in more detail.

**Online inference: adding time.** Perception is an ongoing process that requires continually updating beliefs as data arrives, rather than reverting to a fixed prior in the distant past. To model this sequential process, we adopt a rolling update scheme, where the current posterior becomes the next prior (Fig. 7, [60]). Combining this online structure with the dynamics from eq. (6), we arrive at the following discrete-time update rule:

$$\boldsymbol{u}_{t+1} \;=\; \boldsymbol{u}_t \;+\; \Phi^T \boldsymbol{x} \;-\; \Phi^T \Phi \, \boldsymbol{z}_t, \tag{7}$$

where $\boldsymbol{u}_t$ and $\boldsymbol{u}_{t+1}$ are interpreted as the prior and posterior membrane potentials at time $t$ (Fig. 5).

Notably, the leak term in eq. (6)—which stems from the KL term in eq. (2)—disappears in this discrete-time update. This is because we perform only a single step per time point in the online setting, where the prior evolves continuously over time (Fig. 7). In appendix B.9, we derive the general form of free energy for sequences, and in appendix B.10, we explain why the KL contribution vanishes in the single-update limit.

**Lateral competition as a stabilizing mechanism.** The recurrent connectivity matrix, $W := \Phi^T \Phi$, has a stabilizing effect. To demonstrate this, we transform the discrete-time dynamics in eq. (7) from membrane potentials to firing rates, yielding the following multiplicative update for neuron $i$:

$$\mathrm{r}_{t+1,i} \;=\; \mathrm{r}_{t,i} \, \frac{\exp\left(\Phi^T \boldsymbol{x}\right)_i}{\exp\left(W_{ii} \, \boldsymbol{z}_{t,i}\right) \prod_{j=1, j \neq i}^{K} \exp\left(W_{ij} \, \boldsymbol{z}_{t,j}\right)}, \tag{8}$$

where $\mathrm{r}_t = \exp(\boldsymbol{u}_t)$ are firing rates, and $\boldsymbol{z}_t \sim \mathcal{P}\mathrm{ois}(\boldsymbol{z}; \mathrm{r}_t)$ are sampled spikes. The resulting denominator reveals a form of multiplicative divisive normalization: co-active neurons suppress each other proportionally to their spike output.

As seen in eq. (8), the recurrent term $W \boldsymbol{z}_t$ acts as a stabilizing force in two ways. First, its diagonal entries are positive (i.e., $W_{ii} = \|\Phi_{\cdot i}\|_2^2 > 0$); therefore, neurons that spike strongly receive self-suppression, dampening activity at the next time point. Second, the off-diagonal entries couple neurons with overlapping tuning. Due to this coupling, vigorously active neurons extend suppressive influence toward their similarly-tuned neighbors (i.e., when $W_{ij} > 0$), preventing excess activity. In addition to stabilization, such competitive interactions also underlie the sparsification dynamics observed in cortical circuits [85–87], and are widely regarded as a hallmark of sparse coding [39, 88].

In sum, we have shown that natural gradient descent on free energy, combined with biologically motivated assumptions, leads to principled and interpretable neural dynamics. As our next step, we apply the theoretical derivations (eq. (6), appendix B.7, and Fig. 5) to design specific instances of iterative VAEs within the broader FOND framework. These include: the iterative Poisson VAE (i$\mathcal{P}$-VAE), the iterative Gaussian VAE (i$\mathcal{G}$-VAE), and the iterative Gaussian-relu VAE (i$\mathcal{G}_{\mathrm{relu}}$-VAE).

## 5 Experiments

We evaluate the family of iterative VAE models introduced in this work (i$\mathcal{P}$-VAE, i$\mathcal{G}$-VAE, i$\mathcal{G}_{\mathrm{relu}}$-VAE; Table 1). These models share identical inference dynamics derived in section 4 and appendix B.7, differing only in their latent variable distributions (Poisson vs. Gaussian) and an optional nonlinearity (e.g., relu [89]) applied after sampling from the posterior. This systematic comparison helps isolate the relative influence of each component on learning brain-like representations that generalize. To evaluate the impact of inference methods, we compare these iterative models to their amortized counterparts ($\mathcal{P}$-VAE, $\mathcal{G}$-VAE, $\mathcal{G}_{\mathrm{relu}}$-VAE), all implemented with identical convolutional encoders.

We also compare to classic normative models in neuroscience, including two predictive coding models, standard PC [22] and incremental PC (iPC; [90]), and the locally competitive algorithm (LCA; [39]), a sparse coding model that can be viewed as a deterministic, non-spiking precursor to i$\mathcal{P}$-VAE. See Table 1 for a summary of models, and Fig. 2 for a visualization of the model tree. Additional comparisons to hybrid iterative-amortized VAEs [91, 92] are presented in the appendix.

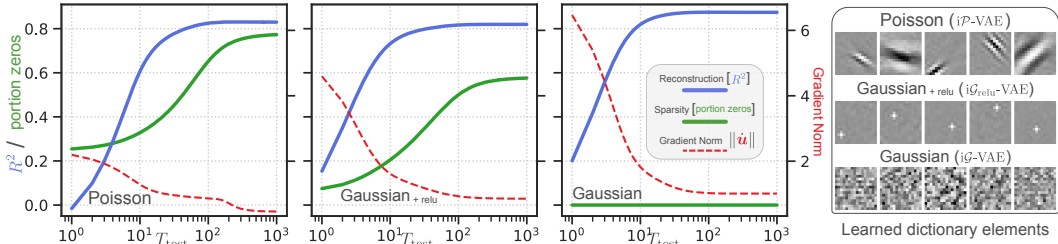

Figure 3: All iterative VAEs converge beyond the training regime ($T_{\text{train}} = 16$). i$\mathcal{P}$-VAE outperforms i$\mathcal{G}_{\text{relu}}$-VAE in sparsity, while i$\mathcal{G}$-VAE achieves superior reconstruction but with dense representations. i$\mathcal{P}$-VAE maintains reasonable reconstruction performance, despite using constrained representations (sparse, integer-valued spike count). The sparsification dynamics of both i$\mathcal{P}$-VAE and i$\mathcal{G}_{\text{relu}}$-VAE resemble those observed in the mouse visual cortex [106] (appendix C.3). All models are trained on $16 \times 16$ natural image patches, and the traces are averages over the entire test set. The right panel displays representative dictionary elements ($\Phi$); see Fig. 11 for the complete set of $K = 512$ features.

**Learning to infer.** To learn the model parameters, we use a training scheme that accumulates gradients across the entire inference trajectory and applies a single update at the end. Specifically, we perform $T_{\text{train}}$ inference steps per input batch and optimize model weights using the accumulated gradients across all steps (not just the final state). This procedure is analogous to backpropagation through time [93, 94], and $T_{\text{train}}$ can be interpreted as the effective model "depth" [43, 95–97] (Fig. 5). We find that the choice of $T_{\text{train}}$ has a significant effect on quantitative metrics, making it an important hyperparameter to track. We explore various $T_{\text{train}}$, but report performance using the same $T_{\text{test}} = 1,000$ steps.

**Hyperparameters.** In addition to $T_{\text{train}}$, we explore various $\beta$ values [78], which controls the reconstruction-KL trade-off (eq. (2)). For i$\mathcal{P}$-VAE and $\mathcal{P}$-VAE [50], varying $\beta$ traces out a reconstruction-sparsity landscape, loosely analogous to a rate-distortion curve [77]. In the main paper, all models have a latent dimensionality of $K = 512$, with linear decoders comprised of a single dictionary, $\Phi \in \mathbb{R}^{M \times K}$, where $M$ is the input dimensionality (i.e., number of pixels). In the appendix, we explore more general nonlinear decoder architectures and different latent dimensionalities. See appendix C.1 for additional implementation, hyperparameter, and training details.

**Tasks and Datasets.** We evaluate models in several ways, including convergence behavior, reconstruction–sparsity trade-off, downstream classification, and out-of-distribution (OOD; [98, 99]) generalization. We train and evaluate models on two datasets: (1) whitened $16 \times 16$ natural image patches from the van Hateren dataset [100], used to assess convergence and reconstruction–sparsity trade-offs; (2) MNIST [101], used for reconstruction, classification, and OOD generalization tests. A good portion of these results, including the OOD ones, are presented in the appendix, as our primary focus in this paper is theoretical, and these results warrant their focused exploration in future work.

**Performance metrics.** We define model *representations* as samples drawn from the posterior at each time point (e.g., $z(t)$ in Fig. 1a), and quantify performance using the following metrics. For reconstruction, we compute the coefficient of determination ($R^2$) between the input image, $x$, and its reconstruction, $\hat{x} = \Phi z$. $R^2(x, \hat{x})$ quantifies the proportion of input variance explained and yields a dimensionality-independent score bounded from above ($R^2 \leq 1$, typically in $[0, 1]$ for effective models). For sparsity, we use the proportion of zeros in the sampled latents: `torch.mean(z == 0)`. This is a simple but informative proxy for energy efficiency in hardware implementations [102–105].

**i$\mathcal{P}$-VAE converges to sparser states while maintaining competitive reconstruction performance.** We frame inference as convergence to attractor states (Fig. 1a) that faithfully represent inputs, achieving high-fidelity reconstruction with sparsity. To evaluate convergence quality, we train iterative VAEs using $T_{\text{train}} = 16$, and monitor inference process across $T_{\text{test}} = 1,000$ iterations, tracking three metrics averaged over the test set: (1) reconstruction fidelity via $R^2$, (2) sparsity via the proportion of zeros, and (3) proximity to the attractor via gradient norm $\|\dot{u}\| \propto \|\mathbf{G}^{-1}(u) \nabla_u \mathcal{F}\|$ calculated across the latent dimension ($K = 512$). These metrics collectively characterize convergence dynamics and allow us to evaluate attractor quality (Fig. 3).

We determine convergence by detecting when the $R^2$ trace flattens and remains stable (details in appendix C.2). For i$\mathcal{P}$-VAE, i$\mathcal{G}_{\text{relu}}$-VAE, and i$\mathcal{G}$-VAE, respectively, we observe convergence times of 95, 75, and 69 iterations. At the final state ($t = 1,000$), these models achieve $R^2$ values of 0.83, 0.82, and 0.87, with latents exhibiting 77%, 58%, and 0% zeros (Fig. 3). i$\mathcal{P}$-VAE achieves superior sparsity compared to both Gaussian models, while i$\mathcal{G}$-VAE achieves the best overall $R^2$. Throughout inference, the gradient norm $\|\boldsymbol{u}\|$ steadily decreases but plateaus around 0.5, 0.9, and 1.0, respectively, likely due to gradient variance induced by stochastic inference dynamics.

**i$\mathcal{P}$-VAE learns V1-like features.** Analysis of the learned features reveals that i$\mathcal{P}$-VAE develops V1-like Gabor filters [107–110], while i$\mathcal{G}_{\text{relu}}$-VAE learns localized pixel-like patterns and i$\mathcal{G}$-VAE learns unstructured features (see right panel in Fig. 3, and Fig. 11 for complete dictionaries). Additionally, when tested with drifting gratings of varying contrasts, i$\mathcal{P}$-VAE exhibits contrast-dependent response latency characteristic of V1 neurons [111, 112] (Fig. 8), possibly a consequence of the normalization dynamics that emerge from our theoretical derivations (eq. (8)). We explore these cortex-like properties further in appendix C.3. In summary, these results suggest i$\mathcal{P}$-VAE learns brain-like representations while achieving the best reconstruction-sparsity compromise.

**i$\mathcal{P}$-VAE achieves the best overall reconstruction-sparsity trade-off.** To assess the robustness of i$\mathcal{P}$-VAE's performance across hyperparameter settings, we systematically explore combinations of $T_{\text{train}} \in [8, 16, 32]$, and $\beta$ values proportional to training iterations (ranging from $0.5\times$ to $4.0 \times T_{\text{train}}$). To compare across inference methods, we include amortized counterparts with identical $\beta$ selection criteria but using $T_{\text{train}} = 1$. Finally, we also include LCA [39] due to its theoretical similarity to i$\mathcal{P}$-VAE (eq. (6)). See appendices C.1.1 to C.1.4 for architectural and training details for all models.

Figure 4a positions all models within a unified sparsity-reconstruction landscape, enabling direct comparison of performance metrics and revealing hyperparameter sensitivity. This representation can be interpreted through the lens of rate-distortion theory [77], where $R^2$ corresponds to inverse distortion and sparsity to inverse coding rate. The optimal point represents perfect reconstruction ($R^2 = 1.0$) using only zeros (sparsity $= 1.0$). This unachievable ideal is marked with a gold star.

For i$\mathcal{P}$-VAE, varying $T_{\text{train}}$ and $\beta$ traces a characteristic curve where increased sparsity trades off against reconstruction quality at $T_{\text{train}} = 8$, with this trade-off improving at higher training iterations ($T_{\text{train}} = 16, 32$). LCA and i$\mathcal{G}_{\text{relu}}$-VAE exhibit a similar pattern, but i$\mathcal{G}_{\text{relu}}$-VAE consistently underperforms i$\mathcal{P}$-VAE in both metrics, confirming Fig. 3. i$\mathcal{G}$-VAE achieves comparable $R^2$ values to high-$T_{\text{train}}$ i$\mathcal{P}$-VAE fits but with zero sparsity. These extensive evaluations confirm that both i$\mathcal{P}$-VAE and LCA achieve the best overall reconstruction-sparsity trade-offs among the tested models.

Finally, for fair comparison with LCA's *maximum a posteriori* (MAP) estimation, we evaluated VAE models with deterministic decoding, where i$\mathcal{P}$-VAE slightly outperforms LCA (Fig. 9). We also confirm that higher $\beta$ values increase i$\mathcal{P}$-VAE sparsity as theoretically predicted [50] (Fig. 10 and appendix C.4). For the effect of latent dimensionality on performance, see appendix C.5.

**Iterative VAEs unanimously outperform their amortized counterparts.** To quantify overall performance as a single metric, we compute the Euclidean distance from each model to the optimal point (the gold star at $R^2 = 1.0$, sparsity $= 1.0$), with lower distances indicating better performance. Figure 4b shows that iterative VAE models (i$\mathcal{G}$-VAE, i$\mathcal{G}_{\text{relu}}$-VAE, i$\mathcal{P}$-VAE) consistently outperform their amortized counterparts ($\mathcal{G}$-VAE, $\mathcal{G}_{\text{relu}}$-VAE, and $\mathcal{P}$-VAE), despite the latter employing deep convolutional encoders with orders of magnitude more parameters. As predicted by theory (comparing eq. (6) with the LCA dynamics [39]), the performance of LCA and i$\mathcal{P}$-VAE are statistically indistinguishable, though i$\mathcal{P}$-VAE exhibits lower variability across hyperparameter settings.

For clarity in the reconstruction-sparsity analysis, we omit PC and iPC as their dense Gaussian latents are effectively represented by i$\mathcal{G}$-VAE. Extended experiments on MNIST (appendix C.6) show that iterative VAEs outperform PCNs in reconstruction metrics, with i$\mathcal{P}$-VAE achieving the best reconstruction-sparsity trade-off (Table 3). Interestingly, in downstream classification, $\mathcal{P}$-VAE reaches $\sim 98\%$ accuracy, comparable to supervised PCNs [113]. See appendix C.6 for more details.

**Iterative VAEs match the inference runtime of their amortized counterparts.** Despite requiring $\sim 2\times$ training time (appendix C.1.4), iterative models achieve competitive inference speeds for large batches, matching amortized performance before continuing to superior trade-offs (appendix C.7).

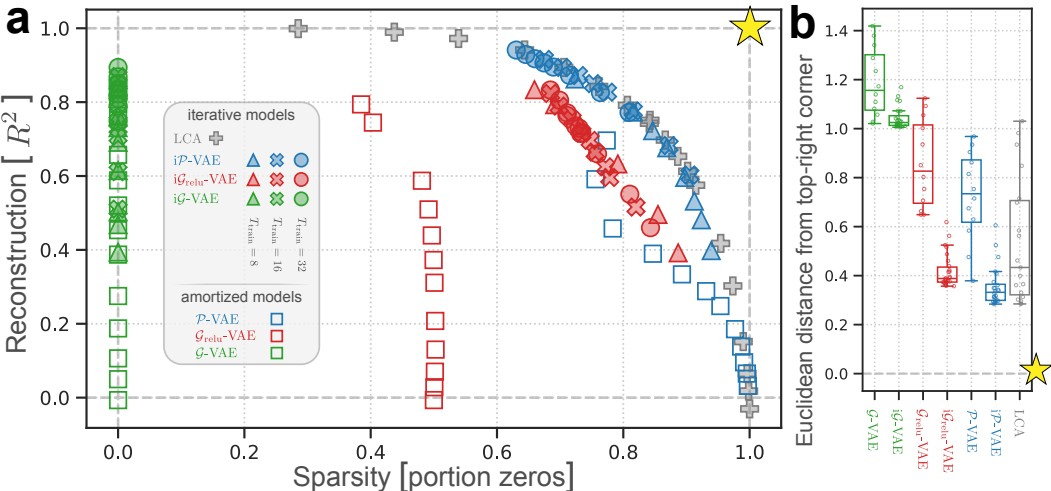

Figure 4: Reconstruction-sparsity trade-off across model families. **(a)** Performance landscape showing reconstruction quality ($R^2$) versus sparsity (proportion of zeros). Different symbols indicate model variants (triangles: $T_{\text{train}} = 8$, crosses: $T_{\text{train}} = 16$, circles: $T_{\text{train}} = 32$, empty squares: amortized VAEs, plus: LCA), with colors denoting model architectures. The gold star marks the theoretical optimum. **(b)** Overall performance measured as Euclidean distance from the optimum point. Iterative models (right side) consistently outperform their amortized counterparts (left side), with i$\mathcal{P}$-VAE and LCA achieving the best overall performance. See also related Figs. 9 and 10.

**i$\mathcal{P}$-VAE exhibits strong out-of-distribution (OOD) generalization.** While the main results use linear decoders ($\hat{\boldsymbol{x}} = \Phi\boldsymbol{z}$), our framework easily extends to deep, nonlinear decoders. In appendix B.6, we develop the theory; and in appendix C.8, we train nonlinear i$\mathcal{P}$-VAE models with multilayer perceptron and convolutional decoders—interpretable as *deep sparse coding*—and compare them to hybrid iterative-amortized VAEs [91, 92]. For both within-dataset perturbations (Fig. 14) and cross-dataset settings (Figs. 13, 15 and 16), i$\mathcal{P}$-VAE consistently outperforms alternatives in OOD reconstruction and classification accuracy (Fig. 14). These results suggest that i$\mathcal{P}$-VAE learns a compositional code (Fig. 17), a hypothesis we aim to study more systematically in future work.

**i$\mathcal{P}$-VAE scales to complex color image datasets and generates realistic images.** In appendix C.9, we demonstrate practical utility by scaling i$\mathcal{P}$-VAE to high-dimensional color images (CelebA, CIFAR-10, tiny ImageNet); and, in appendix C.10, we show that i$\mathcal{P}$-VAE successfully generates realistic MNIST samples through an iterative procedure.

## 6   Conclusions and Discussion

In this paper, we introduced FOND, a framework for deriving brain-like inference dynamics from variational principles. We then applied FOND to derive a new family of iterative VAE models, including the i$\mathcal{P}$-VAE, a spiking model that performs inference in its membrane potential dynamics. i$\mathcal{P}$-VAE's success likely stems from three key attributes: (1) exponential nonlinearities and (2) emergent normalization, similar to recent advances in sequence modeling [114]; and (3) effective stochastic depth through temporal unrolling, which also explains the effectiveness of hierarchical VAEs [115]. Further, i$\mathcal{P}$-VAE's weight reuse and its sparse, integer-valued spike count representations enable efficient hardware implementation [116]. While we focused on variational inference, equally important sampling-based approaches offer complementary perspectives [117–122]. Future work should explore acceleration techniques for iterative inference [114], biologically plausible learning rules [123], predictive dynamics for non-stationary sequences [124], hierarchical extensions [125, 126], and neural data applications [127–129]. We extend upon these points in appendices D.1 to D.5.

In sum, this work connects the prescriptive model development philosophy of FOND with practical algorithms that exhibit brain-like behavior and deliver empirical benefits in machine learning.

## 7 Code and data

Our code, data, and model checkpoints are available here: https://github.com/hadivafaii/IterativeVAE.

## 8 Author contributions

All authors conceived of the project and developed the core theoretical framework. H.V. conducted the formal analysis, implemented the primary models, and generated figures and tables. D.G. implemented baseline comparisons. All authors contributed to writing the manuscript.

## 9 Acknowledgments

We would like to thank Bruno Olshausen, Richard D. Lange, and Ralf M. Haefner for fruitful discussions. This work was supported by the National Institute of Health under award number NEI EY032179. This material is also based upon work supported by the National Science Foundation Graduate Research Fellowship Program under Grant No. DGE-1752814 (DG). Any opinions, findings, conclusions, or recommendations expressed in this material are those of the author(s) and do not necessarily reflect the views of the National Science Foundation. Additionally, DG was funded by the Center for Innovation in Vision and Optics. We thank the developers of the software packages used in this project, including PyTorch [130], NumPy [131], SciPy [132], scikit-learn [133], pandas [134], matplotlib [135], and seaborn [136].

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

# Appendix: Brain-like Variational Inference

# A    Extended Background: Philosophical, Historical, and Theoretical

In this appendix, we present additional background information that complements the main text (section 2). First, we explore the historical and philosophical perspectives underlying our work. We then connect these foundations to recent developments while providing pedagogical explanations of essential statistical concepts. Finally, we review existing machine learning and neuroscience models as instances of free energy minimization, culminating in a model taxonomy shown in Fig. 2.

## A.1    Perception as inference

The idea that perception is an inferential (rather than passive) process has been around for centuries. Early thinkers described it as a process of "unconscious inference" about the hidden causes of sensory inputs [1, 2]. This inference relies on two key components: (i) incoming sensory data, and (ii) *a priori* knowledge about the environment [137]. Prior world knowledge can be acquired either through subjective experience [138–140], or innately built in through evolutionary and developmental processes [141–143]. As recognized in early psychology [144], and later formalized in theoretical and computational frameworks, perception emerges when these two sources—sensory evidence and prior expectations—are integrated to infer the most likely state of the world. Thus, brains actively "construct" their perceptions [3, 145].

This framework remains a major cornerstone of theoretical neuroscience, and has inspired many influential theories across the field [5–7, 22–24, 32, 54, 146–149]. Although these theories go by different names, they are unified by the core idea of "*perception-as-inference*". Of particular relevance to our work are *sparse coding* [54, 55, 150, 151], *predictive coding* [22, 33, 146, 152], and the *free energy principle* [23–25, 148, 149, 153].

Ideas such as *variational inference* from statistics [4, 14–17] and *variational autoencoders* (VAEs) from machine learning [18–20] closely parallel those from theoretical neuroscience. While there has been increasing dialogue between these fields, opportunities remain for deeper integration. Our work speaks directly to this overlap by offering a theoretical synthesis of these ideas (Fig. 2), as well as introducing a new family of brain-inspired machine learning architectures with practical benefits.

## A.2    Bayesian posterior inference

In statistical terms, perception can be modeled as a problem of Bayesian posterior inference [10–12]. We start with a generative model, $p_\theta(\boldsymbol{x})$, that assigns probabilistic beliefs to observations, $\boldsymbol{x} \in \mathbb{R}^M$ (e.g., images), through invoking $K$-dimensional latent variables, $\boldsymbol{z}$:

$$\underbrace{p_\theta(\boldsymbol{x})}_{\substack{\text{marginal likelihood} \\ \text{(a.k.a model evidence)}}} = \int \underbrace{p_\theta(\boldsymbol{x}, \boldsymbol{z})}_{\substack{\text{joint} \\ \text{distribution}}} d\boldsymbol{z} = \int \underbrace{p_\theta(\boldsymbol{x}|\boldsymbol{z})}_{\substack{\text{conditional} \\ \text{likelihood}}} \underbrace{p_\theta(\boldsymbol{z})}_{\text{prior}} d\boldsymbol{z}, \tag{9}$$

where $\theta$ are the adaptable parameters of the generative model (e.g., neural network weights or synaptic connection strengths in the brain). Here, $p_\theta(\boldsymbol{z})$ represents prior beliefs over latent causes, and $p_\theta(\boldsymbol{x}|\boldsymbol{z})$ is the conditional likelihood of observations given latents.

Posterior inference inverts this generative process to determine which latent configurations are most likely given an observation. Apply Bayes' rule to get:

$$\underbrace{p_\theta(\boldsymbol{z}|\boldsymbol{x})}_{\text{posterior}} = \frac{p_\theta(\boldsymbol{x}, \boldsymbol{z})}{p_\theta(\boldsymbol{x})} = \frac{p_\theta(\boldsymbol{x}|\boldsymbol{z})p_\theta(\boldsymbol{z})}{p_\theta(\boldsymbol{x})} = \frac{p_\theta(\boldsymbol{x}|\boldsymbol{z})p_\theta(\boldsymbol{z})}{\int p_\theta(\boldsymbol{x}|\boldsymbol{z})p_\theta(\boldsymbol{z}) \, d\boldsymbol{z}}. \tag{10}$$

This posterior distribution is analytically intractable in all but the simplest cases due to the integration over all latent configurations in the denominator. This necessitates approximate inference methods.

## A.3    Variational inference and deriving the ELBO objective

Variational inference is a powerful approximate inference method that transforms the posterior inference problem, eq. (10), into an optimization task [15–17]. Specifically, one introduces a variational density, $q_\phi(\boldsymbol{z}|\boldsymbol{x})$, parameterized by $\phi$, which approximates the true posterior. The goal is to minimize the Kullback-Leibler (KL) divergence between $q_\phi(\boldsymbol{z}|\boldsymbol{x})$ and the true posterior $p_\theta(\boldsymbol{z}|\boldsymbol{x})$.

This optimization leads to the standard *Evidence Lower BOund* (ELBO) objective [15, 17], which can be derived starting from the model evidence as follows:

$$
\begin{aligned}
\log p_\theta(\boldsymbol{x}) &= \log p_\theta(\boldsymbol{x}) \int q_\phi(\boldsymbol{z}|\boldsymbol{x}) d\boldsymbol{z} \\
&= \int q_\phi(\boldsymbol{z}|\boldsymbol{x}) \log p_\theta(\boldsymbol{x}) d\boldsymbol{z} \\
&= \mathbb{E}_{\boldsymbol{z} \sim q_\phi(\boldsymbol{z}|\boldsymbol{x})} \left[ \log p_\theta(\boldsymbol{x}) \right] \\
&= \mathbb{E}_{\boldsymbol{z} \sim q_\phi(\boldsymbol{z}|\boldsymbol{x})} \left[ \log p_\theta(\boldsymbol{x}) + \log p_\theta(\boldsymbol{z}|\boldsymbol{x}) - \log p_\theta(\boldsymbol{z}|\boldsymbol{x}) \right] \\
&= \mathbb{E}_{\boldsymbol{z} \sim q_\phi(\boldsymbol{z}|\boldsymbol{x})} \left[ \log \frac{p_\theta(\boldsymbol{x}) p_\theta(\boldsymbol{z}|\boldsymbol{x})}{p_\theta(\boldsymbol{z}|\boldsymbol{x})} \right] \\
&= \mathbb{E}_{\boldsymbol{z} \sim q_\phi(\boldsymbol{z}|\boldsymbol{x})} \left[ \log \frac{p_\theta(\boldsymbol{x}, \boldsymbol{z})}{p_\theta(\boldsymbol{z}|\boldsymbol{x})} \right] \\
&= \mathbb{E}_{\boldsymbol{z} \sim q_\phi(\boldsymbol{z}|\boldsymbol{x})} \left[ \log \frac{p_\theta(\boldsymbol{x}, \boldsymbol{z}) q_\phi(\boldsymbol{z}|\boldsymbol{x})}{p_\theta(\boldsymbol{z}|\boldsymbol{x}) q_\phi(\boldsymbol{z}|\boldsymbol{x})} \right] \\
&= \mathbb{E}_{\boldsymbol{z} \sim q_\phi(\boldsymbol{z}|\boldsymbol{x})} \left[ \log \frac{p_\theta(\boldsymbol{x}, \boldsymbol{z})}{q_\phi(\boldsymbol{z}|\boldsymbol{x})} + \log \frac{q_\phi(\boldsymbol{z}|\boldsymbol{x})}{p_\theta(\boldsymbol{z}|\boldsymbol{x})} \right] \\
&= \mathbb{E}_{\boldsymbol{z} \sim q_\phi(\boldsymbol{z}|\boldsymbol{x})} \left[ \log \frac{p_\theta(\boldsymbol{x}, \boldsymbol{z})}{q_\phi(\boldsymbol{z}|\boldsymbol{x})} \right] + \mathbb{E}_{\boldsymbol{z} \sim q_\phi(\boldsymbol{z}|\boldsymbol{x})} \left[ \log \frac{q_\phi(\boldsymbol{z}|\boldsymbol{x})}{p_\theta(\boldsymbol{z}|\boldsymbol{x})} \right] \\
&= \underbrace{\mathbb{E}_{\boldsymbol{z} \sim q_\phi(\boldsymbol{z}|\boldsymbol{x})} \left[ \log \frac{p_\theta(\boldsymbol{x}, \boldsymbol{z})}{q_\phi(\boldsymbol{z}|\boldsymbol{x})} \right]}_{\mathrm{ELBO}(\boldsymbol{x}; \theta, \phi)} + \mathcal{D}_{\mathrm{KL}}\Big( q_\phi(\boldsymbol{z}|\boldsymbol{x}) \,\|\, p_\theta(\boldsymbol{z}|\boldsymbol{x}) \Big).
\end{aligned}
\tag{11}
$$

This concludes our derivation of eq. (1). All we did was multiply by one twice (once by inserting $\int q_\phi(\boldsymbol{z}|\boldsymbol{x}) d\boldsymbol{z}$ and once by introducing the ratio $\frac{q_\phi(\boldsymbol{z}|\boldsymbol{x})}{q_\phi(\boldsymbol{z}|\boldsymbol{x})}$) followed by a few algebraic rearrangements.

Importantly, eq. (1) holds for any $q_\phi(\boldsymbol{z}|\boldsymbol{x})$ that defines a proper probability density. Compared to the typical derivation using Jensen's inequality [15], this derivation yields a stronger result [20]: An exact decomposition of the model evidence into the ELBO plus the KL divergence between the approximate and true posterior.

A direct consequence of this equality is that taking gradients of eq. (1) with respect to $\phi$ shows that maximizing the ELBO minimizes the KL divergence, since the model evidence on the left-hand side is independent of $\phi$. Thus, ELBO maximization improves posterior inference quality, corresponding to more accurate perception under the perception-as-inference framework (appendix A.1).

### A.4 Three distributions, one ELBO

To fully specify the ELBO in eq. (1), we must make choices about the approximate posterior $q_\phi(\boldsymbol{z}|\boldsymbol{x})$ and the joint distribution $p_\theta(\boldsymbol{x}, \boldsymbol{z})$, which further decomposes as $p_\theta(\boldsymbol{x}, \boldsymbol{z}) = p_\theta(\boldsymbol{z}) p_\theta(\boldsymbol{x}|\boldsymbol{z})$. Therefore, we are faced with a total of three independent distributional choices: approximate posterior, prior, and likelihood [1].

In both neuroscience and machine learning, it is common practice to assume all three distributions are Gaussian. These critical assumptions are frequently adopted by convention rather than through theoretical justification. We emphasize that these distributions are entirely at the discretion of the modeler, and there are no fundamental constraints requiring them to be Gaussian in all cases.

---

[1] In the online inference setting, both the prior and approximate posterior (typically) become one family, reducing the independent choices into two: approximate posterior, and likelihood.

## A.5 Two ways to carve up the ELBO

There are two common ways of expressing and interpreting the ELBO. The first, more popular in the machine learning community, is the VAE loss decomposition [20]:

$$
\begin{aligned}
\text{ELBO}(\boldsymbol{x}; \theta, \phi) &= \mathbb{E}_{\boldsymbol{z} \sim q_\phi(\boldsymbol{z}|\boldsymbol{x})} \left[ \log \frac{p_\theta(\boldsymbol{x}|\boldsymbol{z}) p_\theta(\boldsymbol{z})}{q_\phi(\boldsymbol{z}|\boldsymbol{x})} \right] \\
&= \mathbb{E}_{\boldsymbol{z} \sim q_\phi(\boldsymbol{z}|\boldsymbol{x})} \left[ \log p_\theta(\boldsymbol{x}|\boldsymbol{z}) \right] + \mathbb{E}_{\boldsymbol{z} \sim q_\phi(\boldsymbol{z}|\boldsymbol{x})} \left[ \log \frac{p_\theta(\boldsymbol{z})}{q_\phi(\boldsymbol{z}|\boldsymbol{x})} \right] \qquad (12) \\
&= \underbrace{\mathbb{E}_{\boldsymbol{z} \sim q_\phi(\boldsymbol{z}|\boldsymbol{x})} \left[ \log p_\theta(\boldsymbol{x}|\boldsymbol{z}) \right]}_{\text{Reconstruction term (\textit{distortion})}} - \underbrace{\mathcal{D}_{\text{KL}} \Big( q_\phi(\boldsymbol{z}|\boldsymbol{x}) \,\|\, p_\theta(\boldsymbol{z}) \Big)}_{\text{KL term (\textit{coding rate})}}.
\end{aligned}
$$

This view emphasizes reconstruction fidelity and latent space regularization. The reconstruction term can be interpreted as a *distortion measure*, quantifying how well the latent code $\boldsymbol{z}$ can explain the input $\boldsymbol{x}$ through the generative model. The KL term, by contrast, acts as an *information rate*, measuring how much input-dependent information is encoded in the posterior $q_\phi(\boldsymbol{z}|\boldsymbol{x})$ beyond what is already present in the prior $p_\theta(\boldsymbol{z})$. In other words, the KL quantifies the *coding cost* of representing $\boldsymbol{x}$ via $\boldsymbol{z}$, and reflects the capacity of the latent space to capture novel, stimulus-specific structure. This interpretation is closely related to classical *rate-distortion theory* [154, 155], and has been formalized in the context of VAEs by Alemi et al. [77].

The second view, more aligned with theoretical neuroscience and physics, splits (negative) ELBO as:

$$
-\text{ELBO}(\boldsymbol{x}; \theta, \phi) \equiv \mathcal{F}(\boldsymbol{x}; \theta, \phi) = \underbrace{\mathbb{E}_{\boldsymbol{z} \sim q_\phi(\boldsymbol{z}|\boldsymbol{x})} \Big[ -\log p_\theta(\boldsymbol{x}, \boldsymbol{z}) \Big]}_{\text{Energy}} - \underbrace{\mathcal{H} \big[ q_\phi(\boldsymbol{z}|\boldsymbol{x}) \big]}_{\text{Entropy}}, \qquad (13)
$$

where $\mathcal{H}[q] = - \int q \log q$ is the Shannon entropy.

This carving is analogous to the concept of *Helmholtz free energy* from statistical physics [156–158], where minimizing free energy involves reducing energy while preserving entropy (i.e., maintaining uncertainty). Below, we will use this decomposition to show that the predictive coding objective of Rao and Ballard [22] can be directly derived from eq. (13) under specific distributional assumptions.

## A.6 Gaussian Variational Autoencoder ($\mathcal{G}$-VAE)

The standard Gaussian VAE ($\mathcal{G}$-VAE) represents a foundational model in the VAE family, where all three distributions are factorized Gaussians [20]. To simplify the model, the prior is typically fixed as a standard normal distribution with zero mean and unit variance: $p_\theta(\boldsymbol{z}) = \mathcal{N}(\boldsymbol{0}, \boldsymbol{1})$.

The key innovation of VAEs lies in how they parameterize the approximate posterior and likelihood distributions using neural networks:

**Approximate posterior.** The encoder neural network transforms each input $\boldsymbol{x}$ into parameters for the posterior distribution. Specifically, it outputs both a mean vector and a variance vector: $\text{enc}(\boldsymbol{x}; \phi) = \boldsymbol{\lambda}(\boldsymbol{x}) \equiv (\boldsymbol{\mu}(\boldsymbol{x}), \boldsymbol{\sigma}^2(\boldsymbol{x}))$. These parameters fully specify the approximate posterior: $q_\phi(\boldsymbol{z}|\boldsymbol{x}) = \mathcal{N}(\boldsymbol{z}; \boldsymbol{\mu}(\boldsymbol{x}), \boldsymbol{\sigma}^2(\boldsymbol{x}))$. Typically, neural networks are made to output $\log \boldsymbol{\sigma}$, which is exponentiated to obtain the variance. This choice ensures the variance always remains positive.

**Likelihood.** The decoder neural network maps sampled latent variables back to reconstructions in the input space, producing the mean of the likelihood distribution: $\boldsymbol{\mu}(\boldsymbol{z}) = \text{dec}(\boldsymbol{z}; \theta)$.

**Likelihood variance.** There are several approaches for handling the variance of the likelihood:

- Fixed variance: Setting a constant variance for all dimensions (this is bad practice: [159]).
- Input-dependent variance: Using a separate neural net to predict variance as a function of $\boldsymbol{z}$.
- Learned global variance: Learning a set of global variance parameters $\boldsymbol{\sigma}^2$ for the entire dataset [69].

In our implementation, we use learned global variance parameters for the decoder, meaning: $p_\theta(\boldsymbol{x}|\boldsymbol{z}) = \mathcal{N}(\boldsymbol{x}; \text{dec}(\boldsymbol{z}; \theta), \boldsymbol{\sigma}^2)$.

**Training.** A critical component enabling end-to-end training of VAEs is the reparameterization trick [18]. This technique allows gradients to flow through the sampling operation by expressing a sample $z$ from $q_\phi(z|x)$ as a deterministic function of $x$ and a noise variable $\epsilon \sim \mathcal{N}(\mathbf{0}, \boldsymbol{I})$:

$$z = \boldsymbol{\mu}(x) + \boldsymbol{\sigma}(x) \odot \boldsymbol{\epsilon},$$

With this formulation, both encoder and decoder parameters can be jointly optimized by maximizing the ELBO objective shown in eq. (12).

**Extensions and modifications.** Since the advent of VAEs [18, 19], numerous proposals have extended or modified the standard Gaussian framework. These efforts can be broadly categorized into three main directions: (i) developing more expressive or learnable priors, (ii) replacing the likelihood function with non-Gaussian alternatives, and (iii) altering the latent distribution.

In terms of priors $p_\theta(z)$, researchers have introduced hierarchical variants [115, 160, 161], structured priors such as VampPrior [162], and nonparametric approaches like the stick-breaking process [163].

For the likelihood function $p_\theta(x|z)$, alternatives to the standard Gaussian have been proposed to better accommodate binary, count, or highly structured data. Examples include the Bernoulli distribution for binary data [18, 164], Poisson [165] and negative binomial [166] for count data, and mixtures of discretized Logistic distributions for natural images [161, 167].

Finally, for inference, many have enhanced the expressiveness of the variational posterior $q_\phi(z|x)$ by applying normalizing flows [168] or inverse autoregressive flows [169], thereby relaxing the mean-field assumption. Others have replaced the Gaussian latents altogether, exploring alternative distributions such as categorical [170, 171], Bernoulli [172, 173], Laplace [174], Dirichlet [175], hyperbolic normal [176], von Mises-Fisher [177], Student's t [178], and negative binomial [179].

### A.7   Poisson Variational Autoencoder ($\mathcal{P}$-VAE)

A particularly relevant departure from Gaussian latent variables is the Poisson VAE ($\mathcal{P}$-VAE; [50]). Motivated by findings in neuroscience that neural spike counts follow Poisson statistics over short temporal windows [51, 180], Vafaii et al. [50] introduced a VAE variant in which posterior inference is performed over discrete spike counts. In $\mathcal{P}$-VAE, both the prior and approximate posterior distributions of the standard Gaussian VAE are replaced with Poisson distributions. The authors also developed a novel Poisson reparameterization trick and derived a corresponding ELBO for Poisson-distributed VAEs.

An important theoretical insight arises from the $\mathcal{P}$-VAE ELBO: the KL divergence term induces a firing rate penalty that resembles *sparse coding* [54], a fundamental and widely documented property of biological neural representations [151]. When trained on natural image patches, $\mathcal{P}$-VAE learns sparse latent representations with Gabor-like basis vectors [107–110], much like classic sparse coding [54]. $\mathcal{P}$-VAE has also been extended to model psychophysical response times [181].

Despite these advances, $\mathcal{P}$-VAE underperformed the *locally competitive algorithm* (LCA; [39])—a classic iterative sparse coding algorithm from 2008—in both reconstruction accuracy and sparsity. The authors attributed this performance gap to a key difference in inference mechanisms: VAEs rely on *amortized* inference; whereas, sparse coding uses *iterative* optimization at test time.

### A.8   Amortized versus iterative variational inference

Once we select our approximate posterior distribution—such as a Gaussian ($\mathcal{G}$-VAE) or Poisson ($\mathcal{P}$-VAE)—we must decide how to parameterize and optimize it. Let $\boldsymbol{\lambda}$ denote the distributional parameters of the approximate posterior (distinct from neural network weights). For example, a Gaussian posterior requires both a mean $\boldsymbol{\mu}$ and variance $\boldsymbol{\sigma}^2$, i.e., $\boldsymbol{\lambda} \equiv (\boldsymbol{\mu}, \boldsymbol{\sigma}^2)$; whereas, a Poisson posterior is specified by a vector of rates, $\boldsymbol{\lambda} \equiv \mathrm{r}$. We now describe three strategies for optimizing $\boldsymbol{\lambda}$: amortized, iterative, and hybrid iterative-amortized inference.

**Amortized inference.** In *amortized variational inference*, an encoder neural network $\mathrm{enc}(x; \phi)$, parameterized by weights $\phi$, maps each observation $x$ to its corresponding posterior parameters in a single forward pass: $\boldsymbol{\lambda}(x) = \mathrm{enc}(x; \phi)$. This approach, central to VAEs, is efficient at test time, as the cost of inference is "amortized" over training, much like spreading out a fixed expense over time in accounting [36].

The amortized approach is partially motivated by the universal approximation capability of neural networks [182]: if there exists a mapping from samples to their optimal posterior parameters, $\boldsymbol{x} \to \boldsymbol{\lambda}^*(\boldsymbol{x})$, then a sufficiently large and deep neural network should theoretically be able to learn it.

However, despite this theoretical promise, amortized inference often suffers from an *amortization gap*—a systematic mismatch between the inferred posterior parameters, $\boldsymbol{\lambda}(\boldsymbol{x}) = \mathrm{enc}(\boldsymbol{x}; \phi)$, and the optimal variational parameters, $\boldsymbol{\lambda}^*(\boldsymbol{x})$. This gap can significantly degrade model performance [183, 184] and can be conceptualized in two equivalent ways.

First, within a given family of distributions, the optimal variational parameters $\boldsymbol{\lambda}^*(\boldsymbol{x})$ are by definition those that minimize the KL divergence from the approximate posterior to the true posterior (the last term in eq. (1)). Second, due to the equality in eq. (1), this KL divergence is mathematically equivalent to the difference between the ELBO and the model evidence, $\log p_\theta(\boldsymbol{x})$. The closer the variational approximation is to the true posterior distribution in terms of KL, the smaller the gap between ELBO and $\log p_\theta(\boldsymbol{x})$ becomes, thereby increasing the "tightness" of the lower bound.

In summary, the amortization gap reveals that despite the promise of the universal approximation theorem [182], even highly parameterized deep neural networks often fail to find the optimal variational parameters [183]. This results in both poor inference quality, as well as substantial discrepancy between the ELBO and the model evidence.

**Iterative inference.**   In contrast to amortized inference, *iterative variational inference* optimizes the posterior parameters $\boldsymbol{\lambda}$ separately for each data point by directly maximizing the ELBO, typically via gradient-based optimization. Although more computationally intensive at test time, this approach often achieves tighter variational bounds and more accurate posterior approximations [183, 185].

A canonical example is stochastic variational inference (SVI; [186]), which updates local variational parameters via stochastic gradient ascent. Iterative optimization is also central to classical Bayesian approaches such as expectation maximization (EM; [13, 187]), coordinate ascent variational inference [188], mean-field variational inference [17, 189], and expectation propagation [190], all of which refine posterior estimates through successive updates.

In neuroscience-inspired models, iterative inference emerges naturally in biologically plausible frameworks such as the *locally competitive algorithm* (LCA; [39]) for sparse coding (SC), and in predictive coding models (PC; [22, 23]). Both SC and PC implement inference through local, recurrent dynamics. While these models are not always formulated explicitly in terms of ELBO maximization, as reviewed in the main text, appendix A.9, and illustrated in Fig. 2, both SC and PC can be interpreted as performing variational inference via iterative refinement of latent variables in response to sensory input.

In summary, iterative inference methods can substantially reduce the amortization gap by producing more accurate posterior approximations. But they incur increased computational cost at test time, often requiring multiple optimization steps per observation.

**Hybrid iterative-amortized inference.**   Several hybrid approaches have been proposed to bridge the gap between amortized and iterative inference, aiming to combine the speed and scalability of amortized methods with the accuracy and flexibility of iterative refinement. The *Helmholtz machine* [4] is an early example, using a recognition network for approximate inference (similar to amortization) but trained through the *wake-sleep* algorithm. This approach alternates between updating the generative model in the "wake" phase and the recognition model in the "sleep" phase, combining aspects of both amortized and iterative approaches in an EM-like fashion.

More recent examples include *iterative amortized inference* [91], which trains a neural network to predict parameter updates rather than the posterior parameters themselves. This approach draws inspiration from meta-learning [191], where the inference procedure is learned as a dynamical process conditioned on data. By learning to iteratively improve posterior estimates, such models can adapt inference trajectories to individual data points while still leveraging amortization during training.

Another example is the *semi-amortized variational autoencoder* [92], which uses a standard encoder network to produce an initial guess for the approximate posterior, and then performs a small number of gradient-based updates, as in stochastic variational inference (SVI; [186]), to refine the estimate. This two-stage approach retains the test-time efficiency of amortization while reducing the amortization gap via local adaptation.

Our models are fully iterative, in the spirit of LCA, but remain conceptually related to these hybrid strategies. We compare to them thoroughly in our extended results below.

**Iterative versus amortized inference: which approach is more brain-like?**    Unlike one-shot amortized inference, neural dynamics unfold over time [75], allowing for progressive refinement of perceptual representations [42, 44]. This temporal unfolding is consistent with recurrent processing and resonance phenomena observed in neural circuits [40, 41, 43], suggesting that the brain engages in a form of iterative inference. This idea is reflected in biologically inspired algorithms such as sparse coding [39] and predictive coding [22, 33], where inference is implemented through ongoing dynamical interactions that minimize error over time.

A counterpoint comes from Gershman and Goodman [36], who argues that the brain may instead employ amortized inference, using fast, feedforward mappings (e.g., learned neural networks) that approximate posterior beliefs without requiring iterative updates. This view emphasizes the efficiency of inference through learned function approximation, contrasting with the flexibility and adaptivity of iterative approaches.

However, this efficiency comes at the cost of adaptability and biological realism. Given the strong evidence for recurrent, time-evolving neural dynamics, we view iterative inference as the more plausible default. Though in practice, the brain may implement a hybrid of both strategies.

The jury is still out.

## A.9   Predictive coding as variational inference

Having established the variational framework for perception, we now demonstrate how the classic predictive coding model of Rao and Ballard [22] can also be understood as a form of variational inference with specific distributional choices. Here, we focus on a single-layer predictive coding network (PCN) with a linear decoder. See Millidge et al. [33] for a comprehensive review, covering more general cases including hierarchical PCNs.

The key insight is that predictive coding is mathematically identical to free energy minimization (or ELBO maximization), with three specific distributional choices: (1) factorized Gaussian prior, (2) factorized Gaussian likelihood, and (3) a Dirac-delta distribution for the approximate posterior, combined with iterative inference.

Consider images $\boldsymbol{x} \in \mathbb{R}^M$ and latent variables $\boldsymbol{z} \in \mathbb{R}^K$. Starting from the free energy formulation in eq. (13), we substitute $q_\phi(\boldsymbol{z}|\boldsymbol{x}) = \delta(\boldsymbol{z} - \boldsymbol{\mu})$, where $\phi \equiv \boldsymbol{\mu}$ are the encoding parameters. With this delta distribution, the entropy term vanishes, simplifying the free energy to:

$$\mathcal{F}(\boldsymbol{x}; \theta, \boldsymbol{\mu}) \; = \; \mathbb{E}_{\boldsymbol{z} \sim \delta(\boldsymbol{z}-\boldsymbol{\mu})}\Big[-\log p_\theta(\boldsymbol{x}, \boldsymbol{z})\Big] \; = \; -\log p_\theta(\boldsymbol{x}, \boldsymbol{\mu}). \tag{14}$$

For the generative model, we adopt factorized Gaussians:

$$\text{Gaussian prior:} \quad p_\theta(\boldsymbol{\mu}) \; = \; \mathcal{N}(\boldsymbol{\mu}; \boldsymbol{\mu}_0, \mathbf{I}),$$
$$\text{Gaussian likelihood:} \quad p_\theta(\boldsymbol{x}|\boldsymbol{\mu}) \; = \; \mathcal{N}(\boldsymbol{x}; \Phi\boldsymbol{\mu}, \mathbf{I}),$$

where $\boldsymbol{\mu}_0 \in \mathbb{R}^K$ are learnable prior parameters, and $\Phi \in \mathbb{R}^{M \times K}$ is a linear decoder (i.e., the dictionary in sparse coding). Therefore, the generative model parameters are $\theta \equiv (\boldsymbol{\mu}_0, \Phi)$.

Substituting these Gaussians into eq. (14) yields:

$$\begin{aligned} \mathcal{F}(\boldsymbol{x}; \theta, \boldsymbol{\mu}) \; &= \; -\log p_\theta(\boldsymbol{x}|\boldsymbol{\mu}) - \log p_\theta(\boldsymbol{\mu}) \\ &= \; \frac{1}{2}\big[(\boldsymbol{x} - \Phi\boldsymbol{\mu})^2 + (\boldsymbol{\mu} - \boldsymbol{\mu}_0)^2\big] + \dots, \end{aligned} \tag{15}$$

where the dots represent constant terms irrelevant to optimization. The two squared terms capture prediction errors at different levels of the hierarchy.

In predictive coding, the neural dynamics emerge directly from gradient descent on $\mathcal{F}$. Specifically, predictive coding assumes neural activity $\boldsymbol{\mu}$ evolves according to $\dot{\boldsymbol{\mu}} = -\nabla_{\boldsymbol{\mu}} \mathcal{F}$, which gives:

$$\dot{\boldsymbol{\mu}} \; = \; \underbrace{\Phi^T \boldsymbol{x}}_{\text{feedforward drive}} \; - \; \underbrace{\Phi^T \Phi \boldsymbol{\mu}}_{\text{lateral connections}} \; - \; \underbrace{(\boldsymbol{\mu} - \boldsymbol{\mu}_0)}_{\text{leak term}}. \tag{16}$$

Table 1: Models considered in this paper. Variational autoencoders (VAEs), sparse coding (SC), and predictive coding networks (PCNs) are different instantiations of variational inference with specific "prescriptive" choices. Our theoretical synthesis explicitly identifies these critical choices—including distribution families for priors and approximate posteriors, as well as the inference optimization method (amortized versus iterative)—that define and distinguish each model. Iterative VAEs, SC, and PCNs optimize encoding parameters $\boldsymbol{\lambda}$ through iteration, while amortized VAEs perform one-shot inference using neural networks with parameters $\phi$. The main paper (sections 4 and 5, Fig. 5) focuses on linear decoders (dictionary $\Phi$), with the general nonlinear decoder case explored in appendices B.6 and C.8 to C.10, Figs. 6 and 13 to 17, and tables 4 and 5. For an illustration of this table as a model tree, see Fig. 2. $\varphi(\cdot)$: nonlinear activation function (e.g., relu), Nat. GD: natural gradient descent.

| Model family | Model | Posterior $q_{\boldsymbol{\lambda}}(\boldsymbol{z}\|\boldsymbol{x})$ | Prior $p_\theta(\boldsymbol{z})$ | Likelihood $p_\theta(\boldsymbol{x}\|\boldsymbol{z})$ | Architecture Encoder | Architecture Decoder | Reference |
|---|---|---|---|---|---|---|---|
| iterative VAE | i$\mathcal{P}$-VAE | $\mathcal{P}$oisson | $\mathcal{P}$oisson | $\mathcal{G}$aussian | Nat. GD | $\Phi\boldsymbol{z}$ | This paper |
| | i$\mathcal{G}$-VAE | $\mathcal{G}$aussian | $\mathcal{G}$aussian | | | $\Phi\boldsymbol{z}$ | |
| | i$\mathcal{G}_\varphi$-VAE | $\mathcal{G}$aussian | $\mathcal{G}$aussian | | | $\Phi\varphi(\boldsymbol{z})$ | |
| amortized VAE | $\mathcal{P}$-VAE | $\mathcal{P}$oisson | $\mathcal{P}$oisson | $\mathcal{G}$aussian | enc$(\boldsymbol{x};\phi)$ | dec$(\boldsymbol{z};\theta)$ | Vafaii et al. [50] |
| | $\mathcal{G}$-VAE | $\mathcal{G}$aussian | $\mathcal{G}$aussian | | | dec$(\boldsymbol{z};\theta)$ | Kingma and Welling [18] |
| | $\mathcal{G}_\varphi$-VAE | $\mathcal{G}$aussian | $\mathcal{G}$aussian | | | dec$(\varphi(\boldsymbol{z});\theta)$ | Whittington et al. [89] |
| PCN | PC | Dirac-delta | $\mathcal{G}$aussian | $\mathcal{G}$aussian | GD | $\varphi(\Phi\boldsymbol{z})$ | Rao and Ballard [22] |
| | iPC | | | | | | Salvatori et al. [90] |
| Sparse coding | LCA | Dirac-delta | $\mathcal{L}$aplace | $\mathcal{G}$aussian | GD | $\Phi\boldsymbol{z}$ | Olshausen and Field [54] |
| | | | | | | | Rozell et al. [39] |

## A.10  Variational free energy minimization unifies machine learning and neuroscience

Throughout this extended background, we reviewed how variational free energy ($\mathcal{F}$) minimization serves as a unifying theoretical framework that connects seemingly disparate approaches across machine learning and theoretical neuroscience. This synthesis reveals that models as diverse as variational autoencoders (VAEs), predictive coding, and sparse coding can all be derived from the same fundamental principle of $\mathcal{F}$ minimization, differing only in their specific prescriptive choices.

Our synthesis emphasizes two critical dimensions of choice that determine a model's characteristics:

1. **Distributional choices**: The selection of three distributions
    - *approximate posterior* : $q_{\boldsymbol{\lambda}}(\boldsymbol{z}|\boldsymbol{x})$,
    - *prior* : $p_\theta(\boldsymbol{z})$,
    - *likelihood* : $p_\theta(\boldsymbol{x}|\boldsymbol{z})$.

2. **Inference optimization strategy**: The decision between
    - *amortized inference*: training a neural network to compute posterior parameters in a single forward pass, versus
    - *iterative inference*: optimizing parameters through iterative (e.g., gradient-based) refinement.

These choices define a rich taxonomy of models, illustrated in Fig. 2. For example, combining amortized inference with factorized Gaussian distributions yields the standard VAE ($\mathcal{G}$-VAE; [18]). Replacing the Gaussian prior and posterior with Poisson distributions leads to the Poisson VAE ($\mathcal{P}$-VAE; [50]).

On the neuroscience side, models based on iterative inference with Dirac-delta posteriors and Gaussian likelihoods give rise to predictive coding [22] and sparse coding [54]. Predictive coding typically assumes a Gaussian prior, while sparse coding often uses a Laplace prior to encourage sparsity (Fig. 2). See Friston [23] and Millidge et al. [33] for derivations of predictive coding as variational inference, and Olshausen [55] and Chapter 10 of Dayan and Abbott [56] for sparse coding.

Overall, this systematic mapping clarifies how models originally developed in separate domains and described with different terminology can be understood through a common theoretical lens.

Building on this synthesis, our contribution is introducing the FOND framework (*Free energy Online Natural-gradient Dynamics*), which derives brain-like inference dynamics as natural gradient descent on free energy with specific distributional and structural assumptions. Unlike post-hoc theoretical interpretations, FOND starts with principles and derives algorithms in a top-down manner. We apply FOND to derive a new family of iterative VAE architectures: the iterative Poisson VAE (i$\mathcal{P}$-VAE), the iterative Gaussian VAE (i$\mathcal{G}$-VAE), and the iterative Gaussian-relu VAE (i$\mathcal{G}_{\text{relu}}$-VAE).

These models exemplify FOND's prescriptive power, while combining the strengths of two theoretical traditions: the probabilistic foundation of machine learning and the recurrent, adaptive processing characteristic of neuroscience models. By making our prescriptive choices explicit and theoretically grounded, FOND provides not just a synthesis of existing approaches, but a principled framework for developing new algorithms that address limitations in both fields. See Table 1 for a summary of models considered in this work.

# B  Extended Theory: Motivations, Derivations, and Discussions

In this appendix, we present additional motivations for our modeling choices, detailed derivations, and occasional commentary on key outcomes, complementing the theoretical exposition in the main paper (sections 3 and 4).

We begin by clarifying what we mean by "prescriptive" using an analogy from fundamental physics. We then motivate the Poisson assumptions, which led us to the i$\mathcal{P}$-VAE architecture. In the remainder of this appendix, we further develop the theoretical foundations and present generalizations of our framework beyond what is covered in the main paper, along with interpretive insights that clarify its connection to principles of neural computation.

## B.1  What do we mean by "prescriptive"?

In physics, prescriptive theories derive dynamics from first principles rather than hand-tuning to fit data. Notably, the *Principle of Least Action* [61] states that physical systems follow trajectories minimizing the *Action* (time integral of the Lagrangian). However, this principle alone is insufficient without addressing two fundamental questions:

> **Deriving dynamics from first principles**
>
> **(Q1) Dynamical Objects**: What are the mathematical objects of interest which exhibit the dynamics we seek to understand?
>
> **(Q2) Dynamical Equations**: What differential equations govern these objects' time evolution?

For instance, effective field theory [62] addresses these questions most elegantly, where symmetry principles alone resolve both questions (e.g, see Chapter 17 in Schwichtenberg [63]). Thus, in prescriptive theories, the arrow goes from *principles → dynamics*, and not the other way around.

While the elegant derivations of physics (where everything follows from symmetry alone [64]) may not be fully achievable in the messy world of NeuroAI, we can still apply this prescriptive philosophy.

In this work, we follow the same arrow—*from principles to dynamics*—to derive brain-like inference dynamics from first principles. Specifically: (1) we choose real-valued neural membrane potentials as our dynamical objects (Fig. 1a). This is because membrane potentials are the source of information processing in the brain and they directly generate spiking activity [70, 84], which in turn causally drive perception and behavior [192], and (2) we derive inference dynamics as natural gradient descent on variational free energy with Poisson assumptions in an online inference setting (Fig. 1b, FOND).

## B.2 Why Poisson?

To fully specify the variational free energy objective, one must make concrete decisions about distributional assumptions. Our choice of Poisson distributions for both the approximate posterior and prior is motivated by both theoretical principles and empirical observations from neuroscience.

From a theoretical perspective, Poisson is a natural choice for neural coding because spike counts are, by definition, discrete, non-negative integers. When implementing variational inference, distributional choices should reflect the natural constraints of the variables being modeled, and the Poisson distribution inherently respects these constraints for neural spike counts.

Empirically, the Poisson assumption aligns with extensive neurophysiological observations. Neurons exhibit variable responses across repeated presentations of identical stimuli, with spike-count variance proportional to the mean [51–53]. For brief time windows, this proportionality approaches unity [193–195], approximately matching Poisson statistics. While longer windows and higher cortical areas often show super-Poisson variability, this can be attributed to modulation of an underlying inhomogeneous Poisson process [180].

In other words, neurons can be modeled as conditionally Poisson, even if not marginally so [196].

The apparent precision observed in certain neuronal responses [197] is not inconsistent with Poisson processes. Modern models demonstrate that inhomogeneous Poisson rate codes can capture precise spike timing [198], and the high-precision examples often cited against rate coding are effectively captured by Poisson-process Generalized Linear Models [199].

Overall, the Poisson assumption offers mathematical convenience, and it is not too biologically implausible. Neurons are not literally Poisson in all contexts, but this distributional choice provides a reasonable approximation that respects the discrete nature of spike counts while capturing key statistical properties of neural activity. By grounding our theory in these biologically realistic assumptions, we derive dynamics that naturally reproduce important features of cortical computation.

## B.3 Poisson Kullback–Leibler (KL) divergence

We provide a closed-form derivation of the KL divergence between two Poisson distributions. Recall that the Poisson distribution for a single spike count variable $z$, with rate $r \in \mathbb{R}_{>0}$, is given by:

$$\mathcal{P}\mathrm{ois}(z; r) = \frac{r^z}{z!} e^{-r}. \tag{17}$$

Now suppose $p = \mathcal{P}\mathrm{ois}(z; r_0)$ is the prior distribution for this single neuron, and $q = \mathcal{P}\mathrm{ois}(z; r)$ is the approximate posterior. The KL divergence is given by:

$$
\begin{aligned}
\mathcal{D}_{\mathrm{KL}}\Big(q \,\big\|\, p\Big) &= \mathbb{E}_{z \sim q}\left[\log \frac{q}{p}\right] \\
&= \mathbb{E}_{z \sim q}\left[\log \frac{r^z e^{-r}/z!}{r_0^z e^{-r_0}/z!}\right] \\
&= \mathbb{E}_{z \sim q}\left[\log\left(\left(\frac{r}{r_0}\right)^z e^{-(r-r_0)}\right)\right] \\
&= \mathbb{E}_{z \sim q}\left[z \log\left(\frac{r}{r_0}\right) - (r - r_0)\right] \\
&= \mathbb{E}_{z \sim q}\left[z \log\left(\frac{r}{r_0}\right)\right] - r + r_0 \\
&= r \log\left(\frac{r}{r_0}\right) - r + r_0 \\
&= r_0 + r\left(\log\left(\frac{r}{r_0}\right) - 1\right).
\end{aligned}
\tag{18}
$$

We now express the final result in terms of the membrane potentials, $u := \log r$, to get:

$$\mathcal{D}_{\text{KL}}\Big(\mathcal{P}\text{ois}(z; \exp(u)) \,\big\|\, \mathcal{P}\text{ois}(z; \exp(u_0))\Big) \;=\; e^{u_0} + e^u \left(u - u_0 - 1\right). \tag{19}$$

This concludes our derivation of KL divergence for a single Poisson variable. It's easy to see how this result would generalize to $K$ latents, which is what we show in eq. (2).

### B.4 Fisher information matrix

To derive inference dynamics as natural gradient descent on free energy, we must account for the geometry of distributional parameter space. The Fisher information matrix defines this geometry by acting as a Riemannian metric over the space of probability distributions, measuring how sensitively a distribution changes with its parameters.

When small parameter changes produce large shifts in the distribution, the Fisher information is high, meaning those parameters are farther apart in distribution space. Natural gradient descent leverages this structure, adjusting updates to reflect the true curvature of the space; whereas, standard gradient descent implicitly assumes a flat, Euclidean geometry. These geometry-aware updates lead to more efficient and stable optimization, especially in ill-conditioned settings [35, 57, 76, 200].

In what follows, we formalize this idea and derive explicit Fisher information matrices for Poisson and Gaussian distributions.

#### B.4.1 Fisher information matrix as a second-order approximation of the KL divergence

To derive the Fisher information matrix, we begin with a local approximation of the KL divergence between two nearby distributions. Consider a distribution $p_{\boldsymbol{\lambda}}(x)$ parameterized by $\boldsymbol{\lambda}$, and the same distribution with slightly perturbed parameters, $p_{\boldsymbol{\lambda}+\Delta\boldsymbol{\lambda}}(x)$. The KL divergence between these distributions can be expanded using a second-order Taylor approximation:

$$\begin{aligned}
\mathcal{D}_{\text{KL}}\Big(p_{\boldsymbol{\lambda}} \,\big\|\, p_{\boldsymbol{\lambda}+\Delta\boldsymbol{\lambda}}\Big) &= \int p_{\boldsymbol{\lambda}}(x) \log \frac{p_{\boldsymbol{\lambda}}(x)}{p_{\boldsymbol{\lambda}+\Delta\boldsymbol{\lambda}}(x)} dx \\
&\approx \frac{1}{2}\Delta\boldsymbol{\lambda}^T \mathbf{G}(\boldsymbol{\lambda})\Delta\boldsymbol{\lambda} + \mathcal{O}(\|\Delta\boldsymbol{\lambda}\|^3),
\end{aligned} \tag{20}$$

where $\mathbf{G}(\boldsymbol{\lambda})$ is the Fisher information matrix defined as:

$$\boxed{\mathbf{G}(\boldsymbol{\lambda}) \;=\; \mathbb{E}_{x \sim p_{\boldsymbol{\lambda}}(x)}\Big[\nabla_{\boldsymbol{\lambda}} \log p_{\boldsymbol{\lambda}}(x)\nabla_{\boldsymbol{\lambda}} \log p_{\boldsymbol{\lambda}}(x)^T\Big]} \tag{21}$$

This matrix appears as the Hessian in the second-order approximation of the KL divergence, revealing its role as the natural metric for measuring distances between nearby probability distributions.

From this perspective, natural gradient descent updates take the form:

$$\boldsymbol{\lambda}_{t+1} = \boldsymbol{\lambda}_t - \eta \mathbf{G}^{-1}(\boldsymbol{\lambda}_t)\nabla_{\boldsymbol{\lambda}}\mathcal{L}(\boldsymbol{\lambda}_t) \tag{22}$$

where $\mathcal{L}(\boldsymbol{\lambda})$ is the loss function (in our case, variational free energy), and $\eta$ is the learning rate. By incorporating the inverse Fisher information matrix, the update accounts for the curvature of distribution space, leading to more effective optimization.

#### B.4.2 Poisson Fisher information matrix

In this paper, we considered the canonical parameterization of the Poisson distribution in terms of log rate, $u := \log r$, interpreted as membrane potentials. We now derive the Fisher information matrix for a single Poisson variable in this log rate parameterization, where the probability mass function is:

$$\mathcal{P}\text{ois}(z; e^u) \;=\; \frac{e^{uz}}{z!} e^{-e^u} \tag{23}$$

The log-likelihood as a function of $u$ is:

$$\log \mathcal{P}\text{ois}(z; e^u) \;=\; uz - e^u - \log z! \tag{24}$$

Taking the derivative with respect to $u$:

$$\frac{\partial}{\partial u} \log \mathcal{P}\text{ois}(z; e^u) = z - e^u \tag{25}$$

Plug this expression in eq. (21) to get:

$$
\begin{aligned}
\mathbf{G}(u) &= \mathbb{E}_{z \sim \mathcal{P}\text{ois}(z; e^u)}\left[\left(\frac{\partial}{\partial u} \log \mathcal{P}\text{ois}(z; e^u)\right)^2\right] \\
&= \mathbb{E}_{z \sim \mathcal{P}\text{ois}(z; e^u)}\left[(z - e^u)^2\right] \\
&= \mathbb{E}_{z \sim \mathcal{P}\text{ois}(z; e^u)}\left[z^2 - 2ze^u + e^{2u}\right] \\
&= \mathbb{E}_{z \sim \mathcal{P}\text{ois}(z; e^u)}\left[z^2\right] - 2e^u \, \mathbb{E}_{z \sim \mathcal{P}\text{ois}(z; e^u)}\left[z\right] + e^{2u}.
\end{aligned}
\tag{26}
$$

For a Poisson distribution with rate $e^u$, we know $\mathbb{E}[z] = e^u$ and $\text{Var}[z] = e^u$, which means $\mathbb{E}[z^2] = \text{Var}[z] + \mathbb{E}[z]^2 = e^u + e^{2u}$. Substituting, we get:

$$
\begin{aligned}
\mathbf{G}(u) &= (e^u + e^{2u}) - 2e^u \cdot e^u + e^{2u} \\
&= e^u + e^{2u} - 2e^{2u} + e^{2u} \\
&= e^u.
\end{aligned}
\tag{27}
$$

The final result reveals a remarkably simple form for the Fisher information matrix in the membrane potential parameterization:

$$\boxed{\text{Poisson:} \qquad \mathbf{G}(u) = e^u} \tag{28}$$

For the multivariate case with independent Poisson components, the Fisher information matrix becomes diagonal with entries $\mathbf{G}_{ii}(\mathbf{u}) = e^{u_i}$, or in vector notation, $\mathbf{G}(\boldsymbol{u}) = \text{diag}(e^{\boldsymbol{u}})$.

The natural gradient update for the membrane potentials thus takes the form:

$$\dot{\boldsymbol{u}} = -\mathbf{G}^{-1}(\boldsymbol{u})\nabla_{\boldsymbol{u}}\mathcal{F} = -e^{-\boldsymbol{u}} \odot \nabla_{\boldsymbol{u}}\mathcal{F}. \tag{29}$$

This derivation leads directly to the dynamic equations shown in eq. (6), providing a principled foundation for our spiking neural inference model.

### B.4.3 Gaussian Fisher information matrix

For the Gaussian distribution, we define the log standard deviation, $\xi := \log \sigma$, and derive the Fisher information matrix using the $(\mu, \xi)$ parameterization. The probability density function is:

$$\mathcal{N}(x; \mu, e^{2\xi}) = \frac{1}{\sqrt{2\pi e^{2\xi}}} \exp\left(-\frac{(x-\mu)^2}{2e^{2\xi}}\right) \tag{30}$$

And the log-likelihood is:

$$\log \mathcal{N}(x; \mu, e^{2\xi}) = -\frac{1}{2}\log(2\pi) - \xi - \frac{(x-\mu)^2}{2e^{2\xi}} \tag{31}$$

We compute the partial derivatives with respect to the parameters:

$$
\begin{aligned}
\frac{\partial}{\partial \mu} \log \mathcal{N}(x; \mu, e^{2\xi}) &= \frac{x - \mu}{e^{2\xi}}, \\
\frac{\partial}{\partial \xi} \log \mathcal{N}(x; \mu, e^{2\xi}) &= -1 + \left(\frac{x - \mu}{e^{\xi}}\right)^2.
\end{aligned}
\tag{32}
$$

The Fisher information matrix components are:

$$
\begin{aligned}
\mathbf{G}_{\mu\mu} &= \mathbb{E}_{x\sim\mathcal{N}(x;\mu,e^{2\xi})}\left[\left(\frac{\partial}{\partial\mu}\log\mathcal{N}(x;\mu,e^{2\xi})\right)^2\right] \\
&= \mathbb{E}_{x\sim\mathcal{N}(x;\mu,e^{2\xi})}\left[\frac{(x-\mu)^2}{e^{4\xi}}\right] \\
&= \frac{1}{e^{4\xi}}\mathbb{E}_{x\sim\mathcal{N}(x;\mu,e^{2\xi})}\left[(x-\mu)^2\right] \\
&= \frac{1}{e^{4\xi}}\cdot e^{2\xi} \\
&= e^{-2\xi}.
\end{aligned}
\tag{33}
$$

And for the $\xi$ component:

$$
\begin{aligned}
\mathbf{G}_{\xi\xi} &= \mathbb{E}_{x\sim\mathcal{N}(x;\mu,e^{2\xi})}\left[\left(\frac{\partial}{\partial\xi}\log\mathcal{N}(x;\mu,e^{2\xi})\right)^2\right] \\
&= \mathbb{E}_{x\sim\mathcal{N}(x;\mu,e^{2\xi})}\left[\left(-1+\left(\frac{x-\mu}{e^{\xi}}\right)^2\right)^2\right] \\
&= \mathbb{E}_{x\sim\mathcal{N}(x;\mu,e^{2\xi})}\left[1-2\left(\frac{x-\mu}{e^{\xi}}\right)^2+\left(\frac{x-\mu}{e^{\xi}}\right)^4\right].
\end{aligned}
\tag{34}
$$

For a Gaussian, we have $\mathbb{E}[(x-\mu)^2]=e^{2\xi}$ and $\mathbb{E}[(x-\mu)^4]=3e^{4\xi}$. After expansion and computing the expectations, we get the final result:

$$
\mathbf{G}_{\xi\xi} = 2.
\tag{35}
$$

Remarkably, when using the log standard deviation parameterization, the corresponding Fisher information metric becomes constant.

Finally, for the cross-terms we have:

$$
\begin{aligned}
\mathbf{G}_{\mu\xi} &= \mathbb{E}_{x\sim\mathcal{N}(x;\mu,e^{2\xi})}\left[\frac{\partial}{\partial\mu}\log\mathcal{N}(x;\mu,e^{2\xi})\cdot\frac{\partial}{\partial\xi}\log\mathcal{N}(x;\mu,e^{2\xi})\right] \\
&= \mathbb{E}_{x\sim\mathcal{N}(x;\mu,e^{2\xi})}\left[\frac{x-\mu}{e^{2\xi}}\cdot\left(-1+\left(\frac{x-\mu}{e^{\xi}}\right)^2\right)\right].
\end{aligned}
\tag{36}
$$

Since $\mathbb{E}[x-\mu]=\mathbb{E}[(x-\mu)^3]=0$ for a Gaussian, this cross-term vanishes:

$$
\mathbf{G}_{\mu\xi} = 0.
\tag{37}
$$

Therefore, the Fisher information matrix for a univariate Gaussian with $(\mu,\xi)$ parameterization is:

$$
\text{Gaussian:} \quad \mathbf{G}(\mu,\xi) = \begin{pmatrix} e^{-2\xi} & 0 \\ 0 & 2 \end{pmatrix} = \begin{pmatrix} \frac{1}{\sigma^2} & 0 \\ 0 & 2 \end{pmatrix}
\tag{38}
$$

For the multivariate case with a diagonal covariance matrix, the Fisher information becomes block-diagonal with entries corresponding to each dimension's parameters. This structure leads to separate natural gradient updates for means and log standard deviations.

Equation (38) shows that natural gradient descent in Gaussian parameter space involves scaling the mean gradients by the variance, while the log standard deviation updates require a simple constant scaling factor. In appendix B.7, we will use this result to derive the inference dynamics for the case of Gaussian posterior distributions.

## B.5 Interpreting the terms in Poisson dynamics equations

In this section, we interpret each term in the neural dynamics, eq. (6). We explain their biological significance and draw connections to canonical models in computational neuroscience.

### B.5.1 Feedforward drive

The first term, $\Phi^T \boldsymbol{x}$, represents the feedforward excitatory input to neurons (i.e., "feedforward drive"). It projects the incoming stimulus $\boldsymbol{x}$ into the linear subspace spanned by each neuron's receptive field (the columns of $\Phi$). This operation loosely corresponds to the selective tuning of synaptic weights, with each neuron responding strongly to stimuli matching its preferred feature. In sensory cortices, for example, this feedforward drive could correspond to thalamocortical projections that relay sensory information to cortical neurons [201, 202].

### B.5.2 Recurrent explaining away

The second term, $\Phi^T \Phi \boldsymbol{z}(\boldsymbol{u})$, with spikes sampled from the posterior $\boldsymbol{z}(\boldsymbol{u}) \sim q_{\boldsymbol{u}}(\boldsymbol{z}|\boldsymbol{x})$, captures recurrent interactions within the neural population. Crucially, this interaction occurs through spiking activity $\boldsymbol{z}(\boldsymbol{u})$, not membrane potentials, thereby preserving the biological principle that neurons communicate via discrete spikes rather than continuous voltages [82, 84].

The recurrent connectivity matrix, $\Phi^T \Phi$, allows lateral interactions among latent dimensions. This is known as "explaining away" in probabilistic inference contexts [7]; or as "lateral competition" or "lateral inhibition" in neural circuit models [81, 86, 203–207].

Two distinct suppressive mechanisms emerge: (1) self-suppression via diagonal terms ($[\Phi^T \Phi]_{ii} > 0$), where neurons with high activity suppress their own future responses, acting as a form of spike-frequency adaptation [208]; and (2) lateral inhibition through off-diagonal terms ($[\Phi^T \Phi]_{ij} \neq 0$), where neurons with similar feature preferences compete [85–87]. While most interactions are inhibitory when features are positively correlated ($[\Phi^T \Phi]_{ij} > 0$), excitatory effects can also arise when features are anti-correlated ($[\Phi^T \Phi]_{ij} < 0$).

### B.5.3 Homeostatic leak

The final term, $\beta(\boldsymbol{u} - \boldsymbol{u}_0)$, effectively functions as a homeostatic "leak" that pulls membrane potentials back toward their prior values. This establishes a direct formal connection between a fundamental biophysical property of neurons—their leaky integrator dynamics [81]—and the Kullback-Leibler divergence in variational inference, which measures the *relative information* [209] between posterior and prior distributions.

From a biological perspective, this leak term reflects passive membrane properties that restore neural excitability to baseline in the absence of input, preventing hyperactivity and ensuring metabolic efficiency [210]. From an information-theoretic standpoint, it implements a regularization pressure that penalizes deviations from the prior, balancing the reconstruction objective against representational complexity [77, 78].

The strength of our framework lies in unifying these perspectives: the biophysical constraint of "*don't deviate too much from your baseline state*" aligns nicely with the information-theoretic principle that "*encoding more stimulus-specific information incurs a coding cost.*" This duality offers a principled explanation for why neural systems operate in sparse regimes: they naturally minimize free energy by balancing reconstruction fidelity against metabolic constraints [50].

Overall, the derived i$\mathcal{P}$-VAE dynamics, eq. (6), reveals how neural circuits can implement Bayesian inference through local computations and spiking communication, in a way that mimics established principles of neural physiology. The emergence of these biologically plausible mechanisms directly from variational principles suggests that evolution may have indeed optimized neural systems to perform approximate Bayesian inference [6, 74, 148].

## B.6 Extension to nonlinear decoders

In the main derivation (section 4), we made two simplifying assumptions to enhance clarity: (1) we assumed a linear decoder, and (2) we ignored the variance of the decoder. In this section, we relax these assumptions and provide a full derivation that reduces to those in the main text.

Throughout this work, we assume a Gaussian likelihood function for all models considered. The mean is produced by a decoder neural network, $\boldsymbol{\mu}(\boldsymbol{z}) = \mathrm{dec}(\boldsymbol{z}; \theta) \equiv f_\theta(\boldsymbol{z})$, with parameters $\theta$. For the main derivation, we assumed a unit covariance. But in fact, in the code we have a set of global learnable parameters, $\boldsymbol{\sigma}^2$, that we learn end-to-end alongside other parameters [69]. Given these assumptions, the likelihood is given by:

$$ p_\theta(\boldsymbol{x}|\boldsymbol{z}) = \mathcal{N}(\boldsymbol{x}; f_\theta(\boldsymbol{z}), \boldsymbol{\sigma}^2). $$

And the negative log-likelihood is:

$$ -\log p_\theta(\boldsymbol{x}|\boldsymbol{z}) = \frac{1}{2} \left\| \frac{\boldsymbol{x} - f_\theta(\boldsymbol{z})}{\boldsymbol{\sigma}} \right\|^2 + \dots, $$

where the dots represent terms that are going to vanish after differentiating with respect to the variational parameters, so we didn't bother writing them out.

Similar to the main text, we approximate the reconstruction term using a single Monte Carlo sample:

$$ \mathcal{L}_{\mathrm{recon.}}(\boldsymbol{x}; \theta, \boldsymbol{\lambda}) \approx \frac{1}{2} \left\| \frac{\boldsymbol{x} - f_\theta(\boldsymbol{z}); \theta)}{\boldsymbol{\sigma}} \right\|^2, $$

where $\boldsymbol{z}(\boldsymbol{\lambda}) \sim q_{\boldsymbol{\lambda}}(\boldsymbol{z}|\boldsymbol{x})$. We see that to compute the gradient of the reconstruction term, we only need to compute the gradient of the sample and simply use chain rule $\partial/\partial\lambda_i = (\partial z_i/\partial\lambda_i)(\partial/\partial z_i)$:

$$ \frac{\partial}{\partial\lambda_i} \mathcal{L}_{\mathrm{recon.}}(\boldsymbol{x}; \theta, \boldsymbol{\lambda}) = -\frac{\partial z_i}{\partial\lambda_i} \sum_{j=1}^{M} [\boldsymbol{J}_\theta]_{ij} \frac{x_j - [f_\theta]_j}{\sigma_j}, $$

where we defined the decoder Jacobian as follows:

$$ [\boldsymbol{J}_\theta(\boldsymbol{z})]_{ij} := \frac{\partial}{\partial z_i} [f_\theta(\boldsymbol{z})]_j. \tag{39} $$

In matrix form, the gradient of the reconstruction term reads:

$$ \nabla_{\boldsymbol{\lambda}} \mathcal{L}_{\mathrm{recon.}}(\boldsymbol{x}; \theta, \boldsymbol{\lambda}) = -\frac{\partial \boldsymbol{z}}{\partial \boldsymbol{\lambda}} \odot \left( \boldsymbol{J}_\theta(\boldsymbol{z}) \left[ \frac{\boldsymbol{x} - f_\theta(\boldsymbol{z})}{\sigma^2} \right] \right), \tag{40} $$

where $\partial\boldsymbol{z}/\partial\boldsymbol{\lambda}$ is a vector of length $K$, the decoder Jacobian $\boldsymbol{J}_\theta$ is a $K \times M$ matrix, the expression inside the bracket is a vector of length $M$, and $\odot$ is the Hadamard (element-wise) product. During inference, we evaluate eq. (40) at the particular sample drawn from the posterior, $\boldsymbol{z} = \boldsymbol{z}(\boldsymbol{\lambda})$.

This concludes our extension of the results to the case of nonlinear decoders with a learned global variance. If the decoder is a neural network with relu activations, then the Jacobian can be interpreted as a dynamic feedforward receptive field. In the case of a linear decoder, $f_\theta(\boldsymbol{z}) = \Phi\boldsymbol{z}$, we get a fixed Jacobian: $\boldsymbol{J}_\theta = \Phi^T$, producing the main paper result: $\nabla_{\boldsymbol{\lambda}} \mathcal{L}_{\mathrm{recon.}} \propto -\left(\Phi^T\boldsymbol{x} - \Phi^T\Phi\boldsymbol{z}\right)$.

Finally, compare Fig. 5 and Fig. 6 for a ResNet-style view of the unrolled encoding algorithms, contrasting linear and nonlinear decoders.

## B.7 Extension to Gaussian posteriors

In the main text, we derived dynamics for the case of Poisson latent variables, resulting in the spiking inference dynamics shown in eq. (6). This led to the corresponding iterative Poisson VAE (i$\mathcal{P}$-VAE) architecture.

Here, we extend these results to Gaussian latents, with an optional nonlinearity, $\varphi(\cdot)$, applied after sampling, corresponding to the i$\mathcal{G}$-VAE and i$\mathcal{G}_\varphi$-VAE architectures, respectively.

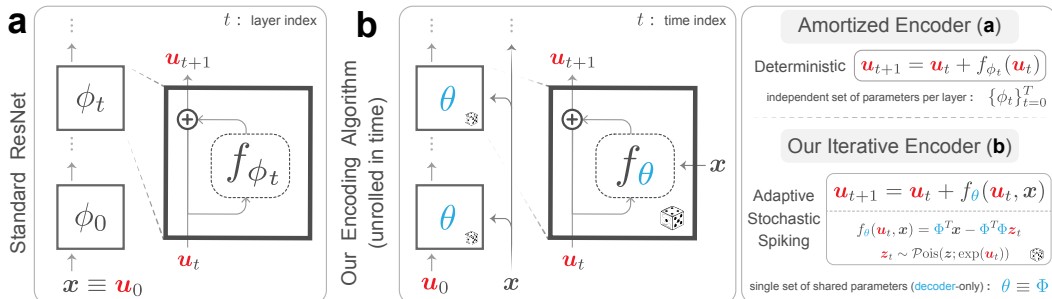

Figure 5: Comparison between standard deep residual neural networks (ResNet; [211]) and our inference algorithm. **(a)** Standard ResNets process information through a series of deterministic layers, each with independent parameters $\phi_t$, where $t$ is the layer index. **(b)** The i$\mathcal{P}$-VAE encoding algorithm (eq. (7)) unrolled in time resembles a ResNet with shared parameters, but incorporates adaptive, stochastic, and spiking dynamics. This recurrent architecture is conceptually similar to *Looped Transformers* [212–214], which use "input injection"—conditioning on the input $x$ at each step—as an effective heuristic to improve stability and performance. In our framework, this mechanism is not a heuristic choice, but emerges directly from the first principles of online variational inference. Finally, unlike conventional networks [95], the i$\mathcal{P}$-VAE uses only decoder parameters $\theta$, enabling online Bayesian inference through iterative sampling and recurrent updates conditioned on both current state $u_t$ and input $x$. This figure illustrates the linear decoder implementation used in the main paper. For the general nonlinear decoder case, see Fig. 6.

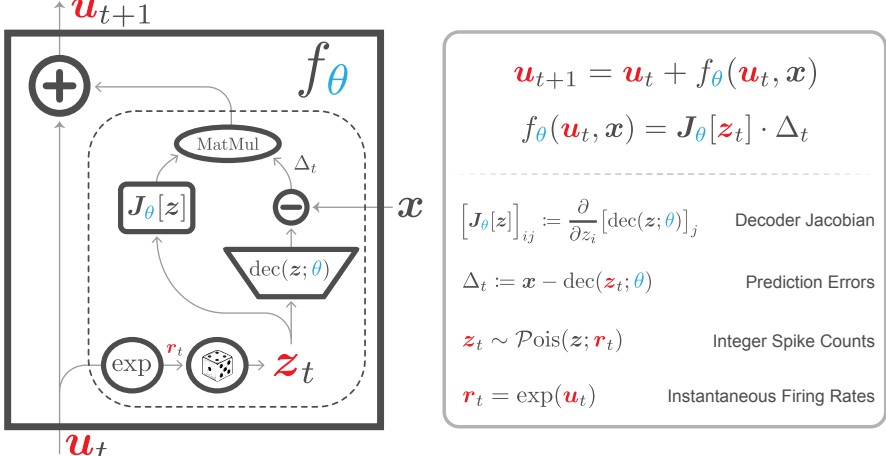

Figure 6: Detailed view of the i$\mathcal{P}$-VAE iterative inference algorithm with a nonlinear decoder. The left panel illustrates the computational flow: membrane potentials ($u_t$) are transformed into firing rates ($r_t$) through an exponential nonlinearity, then sampled to produce integer spike counts ($z_t$). These spikes are passed through a decoder to generate predictions, with errors propagated back through the decoder Jacobian. The right panel shows the corresponding mathematical formulation derived in appendix B.6. For the simpler case of linear decoders used in the main paper, see Fig. 5.

Let's revisit the prescriptive choices needed for deriving dynamics from first principles (appendix B.1). For the Gaussian case, we select the mean, $\mu$, and log standard deviation, $\xi := \log \sigma$, as our dynamical variables. This parameterization avoids constrained optimization that would be necessary with $\sigma$ directly, since $\sigma$ must remain strictly positive while $\xi$ can take any real value. Additionally, as shown in appendix B.4.3, the log standard deviation parameterization yields a constant Fisher scaling, which simplifies our derivations.

Next, we derive Gaussian inference dynamics from natural gradient on free energy. Since we use Gaussian likelihoods across all models in this paper, we've already addressed this component in the main text for linear decoders (eq. (4)) and extended it to general nonlinear decoders in the appendix (eq. (40)). We now need to derive the KL component.

### B.7.1 Gaussian KL divergence

Suppose the posterior and prior parameters for a single latent variable are given by $(\mu, \sigma^2)$ and $(\mu_0, \sigma_0^2)$, respectively. The Gaussian KL term reads:

$$\mathcal{D}_{\mathrm{KL}}\Big(q_{\mu,\sigma^2}(\boldsymbol{z}|\boldsymbol{x}) \,\big\|\, p_{\mu_0,\sigma_0^2}(\boldsymbol{z})\Big) \;=\; \frac{1}{2}\left[\left(\frac{\mu-\mu_0}{\sigma_0}\right)^2 + \left(\frac{\sigma}{\sigma_0}\right)^2 - \log\left(\frac{\sigma}{\sigma_0}\right)^2 - 1\right].$$

With our log standard deviation parameterization, $\xi = \log\sigma$, this becomes:

$$\mathcal{D}_{\mathrm{KL}}\Big(q_{\mu,\xi}(\boldsymbol{z}|\boldsymbol{x}) \,\big\|\, p_{\mu_0,\xi_0}(\boldsymbol{z})\Big) \;=\; \frac{1}{2}\left[\left(\frac{\mu-\mu_0}{e^{\xi_0}}\right)^2 + e^{2(\xi-\xi_0)} - 2\,(\xi-\xi_0) - 1\right].$$

### B.7.2 Gradient of the Gaussian KL term

For the general factorized $K$-variate Gaussian, we compute derivatives with respect to the variational parameters to find:

$$\begin{aligned}
\nabla_{\boldsymbol{\mu}}\mathcal{D}_{\mathrm{KL}}(q\,\|\,p) &= \frac{\boldsymbol{\mu}-\boldsymbol{\mu}_0}{e^{2\xi_0}}, \\
\nabla_{\boldsymbol{\xi}}\mathcal{D}_{\mathrm{KL}}(q\,\|\,p) &= e^{2(\xi-\xi_0)} - 1.
\end{aligned} \tag{41}$$

The next step is to estimate the reconstruction gradient from eq. (40). This requires computing the partial derivative of the sample with respect to each parameter. Using the Gaussian reparameterization trick [20], we have:

$$\boldsymbol{z} \;=\; \boldsymbol{\mu} \,+\, \exp(\boldsymbol{\xi}) \odot \boldsymbol{\epsilon}, \tag{42}$$

with $\boldsymbol{\epsilon} \sim \mathcal{N}(\boldsymbol{0}, \boldsymbol{1})$. Therefore, the sample derivatives are:

$$\frac{\partial \boldsymbol{z}}{\partial \boldsymbol{\mu}} \;=\; 1, \quad \frac{\partial \boldsymbol{z}}{\partial \boldsymbol{\xi}} \;=\; \exp(\boldsymbol{\xi}) \odot \boldsymbol{\epsilon}. \tag{43}$$

### B.7.3 The Gaussian inference dynamics

Following our prescriptive approach:

$$\begin{aligned}
\dot{\boldsymbol{\lambda}} &= -\mathbf{G}^{-1}(\boldsymbol{\lambda})\nabla_{\boldsymbol{\lambda}}\mathcal{F} \\
&= -\mathbf{G}^{-1}(\boldsymbol{\lambda})\left[\nabla_{\boldsymbol{\lambda}}\mathcal{L}_{\mathrm{recon.}} + \beta\nabla_{\boldsymbol{\lambda}}\mathcal{L}_{\mathrm{KL}}\right].
\end{aligned} \tag{44}$$

Combining eqs. (38), (40), (41), (43) and (44) yields the final dynamics for the Gaussian case:

$$\boxed{\begin{aligned}
\dot{\boldsymbol{\mu}} &= e^{2\boldsymbol{\xi}} \odot \left(\boldsymbol{J}_\theta(\boldsymbol{z})\left[\frac{\boldsymbol{x}-f_\theta(\boldsymbol{z})}{\sigma^2}\right]\right) - \beta\,e^{2(\xi-\xi_0)} \odot (\boldsymbol{\mu}-\boldsymbol{\mu}_0) \\
\dot{\boldsymbol{\xi}} &= \frac{1}{2}\,e^{\boldsymbol{\xi}} \odot \boldsymbol{\epsilon} \odot \left(\boldsymbol{J}_\theta(\boldsymbol{z})\left[\frac{\boldsymbol{x}-f_\theta(\boldsymbol{z})}{\sigma^2}\right]\right) - \frac{1}{2}\,\beta\left(e^{2(\xi-\xi_0)}-1\right)
\end{aligned}} \tag{45}$$

Remember that we had $\boldsymbol{z} = \boldsymbol{\mu} + \exp(\boldsymbol{\xi}) \odot \boldsymbol{\epsilon}$, and $\boldsymbol{\epsilon} \sim \mathcal{N}(\boldsymbol{0}, \boldsymbol{1})$.

This completes our derivation of the full Gaussian inference dynamics. Next, we examine how this general equation simplifies for linear decoders and when applying nonlinearities after sampling.

### B.7.4 The iterative Gaussian VAE (i$\mathcal{G}$-VAE)

In the main paper, we focus on linear decoders, defining the iterative Gaussian VAE (i$\mathcal{G}$-VAE) model. For a linear decoder, we have $f_\theta(\boldsymbol{z}) = \Phi\boldsymbol{z}$ and $\boldsymbol{J}_\theta = \Phi^T$. Substituting these into eq. (45) gives us the i$\mathcal{G}$-VAE dynamics:

$$\begin{aligned}
\dot{\boldsymbol{\mu}} &= e^{2\boldsymbol{\xi}} \odot \left[\frac{\Phi^T\boldsymbol{x} - \Phi^T\Phi\boldsymbol{z}}{\sigma^2}\right] - \beta\,e^{2(\xi-\xi_0)} \odot (\boldsymbol{\mu}-\boldsymbol{\mu}_0), \\
\dot{\boldsymbol{\xi}} &= \frac{1}{2}\,e^{\boldsymbol{\xi}} \odot \boldsymbol{\epsilon} \odot \left[\frac{\Phi^T\boldsymbol{x} - \Phi^T\Phi\boldsymbol{z}}{\sigma^2}\right] - \frac{1}{2}\,\beta\left(e^{2(\xi-\xi_0)}-1\right).
\end{aligned} \tag{46}$$

### B.7.5 The iterative Gaussian-$\varphi$ VAE (i$\mathcal{G}_\varphi$-VAE)

We can further extend our framework by applying a nonlinearity, $\varphi(\cdot)$, immediately after sampling:

$$\tilde{z} := \varphi(z) = \varphi(\mu + \exp(\xi) \odot \epsilon). \tag{47}$$

This modification affects the dynamics equations in two ways. First, we decode from $\tilde{z} = \varphi(z)$ rather than $z$ directly: $f_\theta(z) \rightarrow f_\theta(\varphi(z))$. In the linear decoder case, this simplifies to: $\Phi z \rightarrow \Phi \varphi(z)$ (see Table 1). Second, we now have to apply the element-wise product of the derivative of the nonlinearity, evaluated at the sample point. This stems from the sample derivative term from eq. (40).

With these two changes applied, the i$\mathcal{G}_\varphi$-VAE dynamics becomes:

$$
\begin{aligned}
\dot{\mu} &= e^{2\xi} \odot \varphi'(z) \odot \left( J_\theta(\varphi(z)) \left[ \frac{x - f_\theta(\varphi(z))}{\sigma^2} \right] \right) - \beta \, e^{2(\xi - \xi_0)} \odot (\mu - \mu_0), \\
\dot{\xi} &= \frac{1}{2} e^{\xi} \odot \epsilon \odot \varphi'(z) \odot \left( J_\theta(\varphi(z)) \left[ \frac{x - f_\theta(\varphi(z))}{\sigma^2} \right] \right) - \frac{1}{2} \beta \left( e^{2(\xi - \xi_0)} - 1 \right).
\end{aligned} \tag{48}
$$

This approach provides a general recipe that works with any differentiable function. In this paper, we specifically use $\varphi(\cdot) = \mathrm{relu}(\cdot)$ for two key reasons. First, Whittington et al. [89] demonstrated that applying $\mathrm{relu}$ to amortized Gaussian VAEs, combined with activity penalties, produces disentangled representations that can even outperform beta-VAEs [78]. Second, Bricken et al. [215] showed that combining a $\mathrm{relu}$ activation with noisy inputs is sufficient to approximate sparse coding.

In our implementation, we use the exact derivative of the $\mathrm{relu}$ function:

$$\mathrm{relu}'(z) = \Theta(z),$$

where $\Theta(\cdot)$ is the Heaviside step function.

Finally, the i$\mathcal{G}_{\mathrm{relu}}$-VAE dynamics under a linear decoder are given by:

$$
\begin{aligned}
\dot{\mu} &= e^{2\xi} \odot \Theta(z) \odot \left[ \frac{\Phi^T x - \Phi^T \Phi \, \mathrm{relu}(z)}{\sigma^2} \right] - \beta \, e^{2(\xi - \xi_0)} \odot (\mu - \mu_0), \\
\dot{\xi} &= \frac{1}{2} e^{\xi} \odot \epsilon \odot \Theta(z) \odot \left[ \frac{\Phi^T x - \Phi^T \Phi \, \mathrm{relu}(z)}{\sigma^2} \right] - \frac{1}{2} \beta \left( e^{2(\xi - \xi_0)} - 1 \right).
\end{aligned} \tag{49}
$$

### B.8 Biological interpretation of the standard predictive coding dynamics

Due to Gaussian assumptions, the predictive coding inference dynamics (eq. (16), appendix A.9) reveal a rate model where the variables $\mu$ are real-valued and only loosely interpretable as neural activity. In contrast, biological neurons communicate through discrete spikes, not continuous values. One possible interpretation is that $\mu$ represents membrane potentials resulting from the summation of many stochastic synaptic inputs. By the central limit theorem, these could plausibly follow a Gaussian distribution.

Additionally, the dynamics include a recurrent term of the form $\Phi^T \Phi \, \mu$, which can be interpreted as each neuron being directly influenced by the membrane potentials of other neurons, which is unrealistic. Membrane potentials of real neurons are typically thought to be internal to each neuron and not accessible to others. Communication between neurons occurs primarily through spikes [84]. One solution to this is to stay in the space of rate models, but apply a nonlinearity to map from the membrane potentials to rates. In canonical circuit models—starting with Amari [80] (1972), and developed further in biologically inspired frameworks [81–83]—it is standard to apply nonlinearities to neuronal activity before it influences other neurons, thereby maintaining biological plausibility.

While some predictive coding formulations incorporate nonlinearities, they typically apply them after the decoder weights [33]: $\Phi \mu \rightarrow \varphi(\Phi \mu)$. This yields recurrent terms like $\Phi^T \varphi(\Phi \mu)$, which are still at odds with principles of local computation and communication. A more biologically plausible formulation would instead apply nonlinearities before lateral communication: $\Phi^T \Phi \, \varphi(\mu)$.

Our framework addresses these limitations in two distinct ways. In the Poisson case, communication occurs through sampled spike counts, so each neuron only transmits spikes to its neighbors, preserving the privacy of membrane potentials (eq. (6)). In the Gaussian case, we apply a nonlinearity such as

relu after sampling from the posterior, allowing only rectified activity to propagate laterally (eq. (49)). Unlike predictive coding dynamics (eq. (16)), in both of these formulations, membrane potentials remain local to each neuron, in line with biological constraints.

## B.9 Free energy for sequences

Consider a sequence of input data, $\{\boldsymbol{x}_t\}_{t=0}^T$, where $\boldsymbol{x}_t \in \mathbb{R}^M$ (e.g., frames of a video). For such sequential data, we must account for the temporal dependencies in our inference process. After observing the first $t$ frames, we form a prior belief:

$$p_\theta(\boldsymbol{z}|\boldsymbol{x}_{<t}), \tag{50}$$

where $\boldsymbol{x}_{<t} = \{\boldsymbol{x}_0, \boldsymbol{x}_1, \ldots, \boldsymbol{x}_{t-1}\}$ represents all previously observed frames. Upon receiving the next observation $\boldsymbol{x}_t$, we update our beliefs by computing the posterior distribution $q(\boldsymbol{z}_t|\boldsymbol{u})$.

This posterior update can be formulated through the minimization of a local-in-time free energy:

$$\mathcal{F}_t(\boldsymbol{u}) = \underbrace{\mathbb{E}_{\boldsymbol{z}_t \sim q(\boldsymbol{z}_t|\boldsymbol{u})}\Big[-\log p_\theta(\boldsymbol{x}_t|\boldsymbol{z})\Big]}_{\text{reconstruction term}} + \underbrace{\mathcal{D}_{\text{KL}}\Big(q(\boldsymbol{z}_t|\boldsymbol{u}) \,\big\|\, p_\theta(\boldsymbol{z}_t|\boldsymbol{x}_{<t})\Big)}_{\text{KL divergence from prior}}. \tag{51}$$

This formulation captures the essence of Bayesian belief updating: the posterior balances fidelity to the current observation (reconstruction term) against deviation from prior beliefs shaped by all the previous observations (KL term).

Applying natural gradient descent on this time-dependent free energy, we derive the general update equation for sequential inference:

$$\boldsymbol{u}_{t+1} = \boldsymbol{u}_t + \Phi^T \boldsymbol{x}_t - \Phi^T \Phi \boldsymbol{z}_t, \tag{52}$$

where $\boldsymbol{z}_t \sim q(\boldsymbol{z}_t|\boldsymbol{u}_t)$. This equation reflects a rolling update scheme in which the posterior at time $t$ serves as the prior at time $t+1$. We examine this update rule in more detail in the next section.

For stationary sequences (i.e., repeated presentations of the same image, $\boldsymbol{x}_t \to \boldsymbol{x}$), this general update equation reduces to the simplified form presented in the main text (eq. (7)). In this paper, we focus exclusively on such stationary sequences, leaving richer non-stationary dynamics for future work. Future extensions could incorporate more sophisticated predictive dynamics beyond the simple rolling update scheme, potentially allowing the model to *anticipate* future frames rather than just adapt to current ones.

The overall free energy for the entire sequence is the sum of local-in-time free energies:

$$\mathcal{F} = \sum_{t=0}^T \mathcal{F}_t. \tag{53}$$

During model training (i.e., learning the generative model parameters $\theta$), we perform iterative inference updates, computing $\mathcal{F}_t$ at each time point while accumulating gradients. After processing the entire sequence, a single gradient update is then applied to $\theta$ to minimize the overall $\mathcal{F}$, similar to meta-learning approaches [191].

This sequential formulation reveals how our encoding algorithms can be interpreted as unrolled residual networks (ResNets), where each time step corresponds to a network layer with shared parameters (Figs. 5 and 6).

## B.10 Static versus online inference

Traditional variational inference operates on static datasets, assuming samples are drawn independently from a fixed distribution. In sharp contrast, biological perception involves continuous belief updating from streaming observations. This fundamental difference motivates our preference for online inference approaches with an evolving prior.

This section contrasts static and online approaches to inference and their mathematical implications. A key distinction is that online inference features an evolving prior, as opposed to a fixed prior in

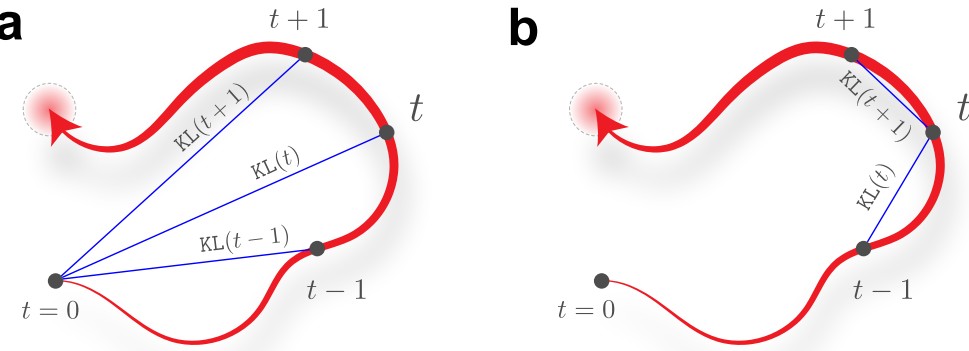

Static iterative inference:
fixed prior, multiple gradient updates

Online inference:
evolving prior, single gradient step per time point

Figure 7: Comparison of static and online inference frameworks. **(a)** In static iterative inference, all posterior distributions (points along the red curve) are regularized toward a fixed prior at $t = 0$. Each blue line represents a KL divergence term computed relative to this same unchanging reference point. **(b)** In online inference, the prior evolves over time, with each posterior becoming the prior for the next time step. KL divergence is computed only between adjacent time points (consecutive points on the red curve), representing a rolling update scheme. This fundamentally different regularization structure leads to the vanishing KL term phenomenon when performing single gradient updates at each time step (e.g., compare eq. (6) with eq. (7)).

static inference. This evolving prior means the KL term (eq. (12)) is always computed relative to the previous time step (Fig. 7b). We show how this observation explains why a single gradient update in online settings naturally leads to dropping the KL contribution to free energy, providing intuition for why BONG [60] advocates using only the expected log-likelihood term in gradient updates.

In static inference (Fig. 7a), the time index $t$ reflects gradient iterations, with each posterior being regularized toward the same fixed prior. As optimization progresses, the KL term consistently pulls the distribution back toward this initial prior point, preventing excessive drift regardless of how many iterations are performed.

In online inference (Fig. 7b), the time index represents sequential data points rather than optimization iterations. The crucial difference is that prior beliefs themselves get updated as new data arrives. When we follow the BONG approach [60] and perform a single gradient update per time step, we only compute the KL divergence between adjacent time points, rather than regularizing all the way back to an initial prior.

Why does the KL regularization term effectively vanish in our discrete-time online setting with single gradient updates? To understand this, we revisit the continuous differential equation from eq. (6):

$$\dot{\boldsymbol{u}} \; \propto \; \Phi^T \boldsymbol{x} - \Phi^T \Phi \, \boldsymbol{z}(\boldsymbol{u}) - \beta(\boldsymbol{u} - \boldsymbol{u}_0)$$

When we discretize this equation for a single update step, we get the following general update rule:

$$\boldsymbol{u}_{t+\delta t} = \boldsymbol{u}_{\text{init}} + \delta t \left[ \Phi^T \boldsymbol{x} - \Phi^T \Phi \, \boldsymbol{z}(\boldsymbol{u}_{\text{init}}) - \beta(\boldsymbol{u}_{\text{init}} - \boldsymbol{u}_0) \right],$$

where $\delta t$ is some small time interval. We choose to initialize at the prior from the previous time point, $\boldsymbol{u}_{\text{init}} \leftarrow \boldsymbol{u}_0$. Consequently, the third term, $\beta(\boldsymbol{u}_{\text{init}} - \boldsymbol{u}_0)$, vanishes for this first gradient step:

$$\boldsymbol{u}_{t+\delta t} = \boldsymbol{u}_0 + \delta t \left[ \Phi^T \boldsymbol{x} - \Phi^T \Phi \, \boldsymbol{z}(\boldsymbol{u}_0) - \beta \underbrace{(\boldsymbol{u}_0 - \boldsymbol{u}_0)}_{} \right],$$

Because of the rolling update scheme, we know the prior at time $t$ is the same as the posterior from time $t - 1$, denoted by $\boldsymbol{u}_t$. In other words, we have $\boldsymbol{u}_0 \equiv \boldsymbol{u}_t$, resulting in:

$$\boldsymbol{u}_{t+\delta t} = \boldsymbol{u}_t + \delta t \left[ \Phi^T \boldsymbol{x} - \Phi^T \Phi \, \boldsymbol{z}(\boldsymbol{u}_t) \right].$$

Assuming $\delta t = 1$, we recover the update rule shown in the main text (eq. (7)). This explains why the leak term is absent in eq. (7). It also explains why BONG [60] advocates dropping the KL term entirely when performing single-step online inference: it has no effect on the first gradient step.

More generally, this analysis reveals a fundamental difference between static and online inference paradigms: static approaches maintain stability through persistent regularization to a fixed prior; whereas, online methods continually update the reference point for regularization and are more adaptive as a result. This distinction is particularly relevant for modeling biological perception, which must balance the preservation of accumulated knowledge with flexibility to new observations.

## C Extended Experiments: Methodological Details and Additional Results

This appendix complements the main experimental results from section 5. We provide detailed information about model architectures, hyperparameters, and training protocols in appendix C.1, along with additional methods, analyses, and findings as we summarize below.

We describe our convergence detection methodology (appendix C.2), demonstrate that i$\mathcal{P}$-VAE exhibits cortex-like properties such as contrast-dependent response latency (appendix C.3), extend our reconstruction-sparsity analysis with MAP decoding experiments (appendix C.4), present MNIST results comparing iterative VAEs with predictive coding networks (appendix C.6), and provide evidence for i$\mathcal{P}$-VAE's strong out-of-distribution (OOD) generalization capabilities for both within-dataset perturbations and cross-dataset settings (appendix C.8).

### C.1 Architecture, Hyperparameter, and Training details

This appendix describes the implementation details of the iterative VAE models introduced in this paper. We cover architecture design, hyperparameter selection, and training procedures. We also provide details about all the other models we compare to, including amortized VAEs, predictive coding variants (PC, iPC), and LCA.

#### C.1.1 Architecture details

Our iterative VAE architectures consist of three key components:

**Encoding mechanism.** The encoding algorithm implements natural gradient descent on variational free energy (eq. (7)), visualized in Figs. 5 and 6. Different distributional choices yield distinct dynamics: Poisson encoding (eq. (7)) for i$\mathcal{P}$-VAE versus Gaussian encoding (eq. (46)) for i$\mathcal{G}$-VAE. The encoder is fundamentally coupled to the decoder, performing *analysis-by-synthesis* [68]. We derive results for linear decoders in the main paper and extend to nonlinear decoders in appendix B.6.

Key implementation details include: (1) learned priors (one parameter per latent dimension), and (2) a learned global step size, $\delta t$, interpretable as a time constant: $\boldsymbol{u}_{t+1} = \boldsymbol{u}_t + \delta t\, \dot{\boldsymbol{u}}_t$, with $\dot{\boldsymbol{u}}$ given by eq. (6). These parameters are learned end-to-end alongside decoder parameters $\theta$.

**Sampling mechanism.** To approximate the reconstruction loss (eqs. (4) and (40)), we use differentiable sampling from the posterior distribution. For Gaussian models, we employ the standard reparameterization trick [18] (eq. (42)), while for i$\mathcal{P}$-VAE we use the Poisson reparameterization algorithm [50]. In i$\mathcal{G}_\varphi$-VAE models, we apply a nonlinearity $\varphi(\cdot)$ post-sampling, which effectively modifies the latent distribution. For example, applying relu to Gaussian samples creates a truncated Gaussian distribution, corresponding to the i$\mathcal{G}_{\text{relu}}$-VAE architecture.

**Decoding mechanism.** Our main results use linear decoders ($\Phi$) that map from latent to stimulus space. This enables closed-form theoretical derivations (eqs. (6), (46) and (49)). In appendix B.6 extend our theory to cover nonlinear decoders; and, in appendix C.8, we extend our empirical analysis to MLP and convolutional decoders. Throughout this paper, all models use Gaussian likelihoods (yielding MSE reconstruction loss) with learned global variance parameters [69].

**Implementation details for other models.** For amortized VAEs, we utilized the implementation from [50], including their five-layer ResNet encoders. For LCA, we used the LCA-PyTorch library [216] (BSD-3 license). For PC and iPC models, we employed the PCX library [113] (Apache-2.0 license) with their default MNIST-optimized MLP architectures. But we matched the latent dimensionality to the other models, using a consistent value of $K = 512$ throughout.

### C.1.2 Hyperparameter details

For the reconstruction-sparsity analysis (Fig. 4), we swept across $T_{\text{train}} \in \{8, 16, 32\}$ and $\beta$ values proportional to each $T_{\text{train}}$, with factors $\in \{0.5, 0.75, 1.0, 1.25, 1.5, 2.0, 3.0, 4.0\}$. This can be seen in the x-axis of Fig. 10. For amortized models, we used the same proportional $\beta$ values but with $T_{\text{train}} = 1$. For MNIST experiments, we used $\beta = T_{\text{train}}$.

For the convergence experiment (Fig. 3), we used $T_{\text{train}} = 16$ with $\beta = 24.0$ for i$\mathcal{P}$-VAE and $\beta = 8.0$ for i$\mathcal{G}$-VAE and i$\mathcal{G}_{\text{relu}}$-VAE, based on optimal values identified in our hyperparameter sweep (Fig. 4). The contrast experiments (Fig. 8) used the same i$\mathcal{P}$-VAE configuration.

For LCA, we used a time constant $\tau = 100$ with 1000 inference iterations and a learning rate of 0.01. We optimized the sparsity parameter $\lambda$ through an extensive sweep on the van Hateren dataset, finding $\lambda = 0.5$ performed best.

### C.1.3 Training details

We trained iterative VAEs using PyTorch with the Adamax optimizer [217] and cosine learning rate scheduling [218], but without warm restarts. We implemented KL term annealing during the first 10% of training following standard VAE practices [219, 220], though we observed minimal impact since our models use a single layer.

For i$\mathcal{P}$-VAE, we adopted the Poisson reparameterization algorithm from [50], annealing the temperature from $\text{temp}_{\text{start}} = 1.0$ to $\text{temp}_{\text{stop}} = 0.01$ during the first half of training.

For the van Hateren dataset, we used a learning rate of 0.002, a batch size of 200, and trained for 300 epochs. MNIST [221] models used identical hyperparameters but were trained for 400 epochs. These settings were consistent across both iterative and amortized VAEs.

For LCA, we adapted the training approach from [50], gradually increasing $\lambda$ from 0 to the target value over 100 epochs. For PC and iPC models, we followed the PCX library examples [113], doubling the default epoch count to ensure convergence.

### C.1.4 Training compute details

The computational requirements for iterative VAEs scale with the number of inference iterations ($T_{\text{train}}$). For example, training the models presented in Fig. 3 with $T_{\text{train}} = 16$ requires approximately three hours per model on an NVIDIA A6000 GPU. Memory usage and training time both scale approximately linearly with $T_{\text{train}}$, as each additional inference iteration requires maintaining the computational graph for backpropagation.

Our complete set of experiments, including all hyperparameter sweeps and model variants, was conducted over the course of approximately one week using six NVIDIA A6000 GPUs (48GB VRAM each). To maximize GPU utilization, we often ran multiple smaller experiments concurrently on a single GPU. Linear decoder models were the most memory-efficient, while experiments with nonlinear decoders for the OOD generalization tasks required more resources due to their additional parameters and the need to backpropagate through relatively more complex architectures.

### C.2 Convergence detection procedure

In section 5 and Fig. 3, we determined convergence by finding the first time point where the $R^2$ trace flattens out and stays flat. Here, we provide more details about this convergence detection algorithm.

**The main intuition.** We consider inference to be complete once the $R^2$ trace becomes essentially flat. This is because $R^2$ is monotonically related to the log-likelihood, and when the curve levels off, further improvements in log-likelihood are negligible.

To obtain a robust estimate of the local trend, we fit a straight line to every 60-step segment (a sliding window) of the trace and measure its slope. We call a window *flat* when the absolute slope falls below a fixed tolerance of $10^{-5}$. To protect against short pauses in learning, we require at least five successive windows to be flat before declaring convergence.

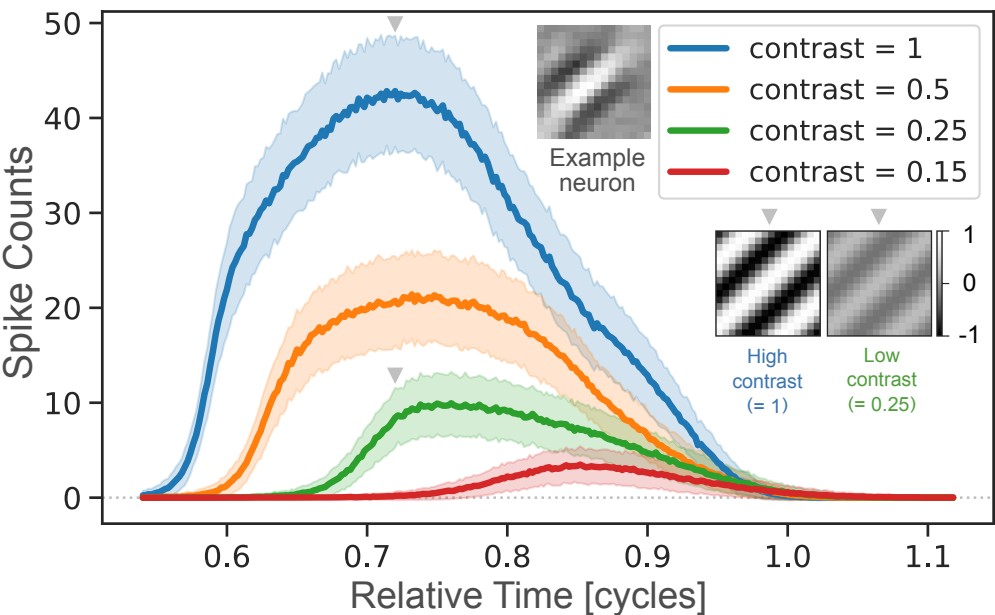

Figure 8: i$\mathcal{P}$-VAE neurons exhibit contrast-dependent response latency, mimicking a well-documented property of V1 neurons. The figure shows trial-averaged firing rates (PSTHs) from a representative model neuron in response to drifting gratings of varying contrast levels. Higher contrast stimuli (blue and orange) elicit responses with shorter onset latencies, earlier peak times, and larger firing rates compared to lower contrast stimuli (green and red). Insets show the learned feature of the selected neuron (top) and the appearance of high contrast (left) versus low contrast (right) grating stimuli. The x-axis represents time in units of grating cycles, while the y-axis shows mean spike counts across $N = 500$ trials. Shaded regions indicate standard deviation. This behavior likely emerges from the divisive normalization in our model dynamics (eq. (8)), even though the model was trained only on static images, suggesting that our theoretical framework captures key aspects of visual cortical computation [111, 112].

**Details of the convergence detection algorithm.** Given $s = (s_1, \ldots, s_{T_{\text{test}}})$, a one-dimensional array of $R^2$ scores in $[0, 1]$, let $\texttt{window} = 60$, $\texttt{tol} = 10^{-5}$, and $\texttt{consecutive} = 5$.

1. **Compute local slopes.** For each start index $i = 0, \ldots, T_{\text{test}} - \texttt{window}$, define

$$\tau = (0, 1, \ldots, \texttt{window} - 1), \qquad y = (s_i, \ldots, s_{i+\texttt{window}-1}),$$

and compute the least-squares slope

$$\beta_i = \frac{\sum_j (\tau_j - \bar{\tau})(y_j - \bar{y})}{\sum_j (\tau_j - \bar{\tau})^2}.$$

Store $|\beta_i|$ in an array $\texttt{slopes}$.

2. **Mark flat windows.** Set $\texttt{flat} = (\texttt{slopes} < \texttt{tol})$.

3. **Require persistence.** Find the smallest index $k$ at which $\texttt{flat}$ is true for $\texttt{consecutive} = 5$ windows in a row. If no such run exists, define $t_{\text{converge}} = T_{\text{test}}$.

4. **Report the convergence index.** Otherwise return

$$t_{\text{converge}} = k + \texttt{window},$$

the first sample that follows the verified flat period (at least $5 \times 60 = 300$ uninterrupted steps).

Please see our code for more info.

## C.3 i$\mathcal{P}$-VAE exhibits cortex-like dynamics: contrast-dependent response latency

The i$\mathcal{P}$-VAE learns spatially localized, oriented, and bandpass (i.e., Gabor-like [107–110]) features (Fig. 11), similar to those learned by classic sparse coding models [54] and LCA [39]. Previous studies have shown that LCA exhibits cortex-like properties, including contrast invariance of orientation tuning, cross-orientation suppression, and so on [222]. Given the theoretical (section 4) and empirical (Fig. 4) similarities between i$\mathcal{P}$-VAE and LCA, we investigated whether i$\mathcal{P}$-VAE also exhibits cortex-like dynamics. We focused on two key neural phenomena with strong experimental support: temporal sparsification dynamics and contrast-dependent response latency.

**V1-like sparsification dynamics.** Recent electrophysiological findings in mouse primary visual cortex (V1) by Moosavi et al. [106] demonstrate that V1 neurons become progressively sparser over time, even after initial response convergence, while maintaining high mutual information with the input stimuli. Remarkably, both i$\mathcal{P}$-VAE and i$\mathcal{G}_{\text{relu}}$-VAE exhibit analogous dynamics in our experiments, continuing to sparsify their representations by an additional ∼10% after functional convergence as measured by reconstruction quality (Fig. 3). For i$\mathcal{P}$-VAE, this post-convergence sparsification coincides with a distinctive dip in the gradient norm $\|\dot{\boldsymbol{u}}\|$ (Fig. 3), suggesting a transition to a new phase of representational refinement. The emergence of this biological property across both models provides evidence that our normative framework captures fundamental principles of neural computation, without being explicitly designed to do so.

**V1-like contrast-dependent response latency.** A significant theoretical outcome of our approach is that a multiplicative form of divisive normalization naturally emerges from the derivations (eq. (8)). The resulting equation differs from canonical normalization that was developed to describe cortical circuits in that there is a product instead of a sum [223]. However, in practice, it captures features of normalization in the cortex. In particular, Carandini et al. [111] demonstrated that V1 neurons exhibit contrast-dependent response latency when presented with drifting gratings of varying contrast. Higher contrast stimuli consistently elicit faster responses, as measured by both onset time and time-to-peak [111, 112].

To investigate whether our model reproduces this phenomenon, we conducted an analogous experiment on the spiking "neurons" of the i$\mathcal{P}$-VAE. We first characterized the tuning properties of individual model neurons to identify their preferred orientation and spatial frequency. We then presented drifting gratings matching these preferred parameters while systematically varying contrast levels $\in \{15\%, 25\%, 50\%, 100\%\}$ (a contrast level of $100\%$ corresponds to a maximum amplitude of 1 in the simulated gratings). We measured i$\mathcal{P}$-VAE neuron responses across $N = 500$ trials, and computed trial-averaged firing rates (i.e., *peristimulus time histograms*, PSTHs) as shown in Fig. 8.

The results reveal response dynamics that are similar to those observed in biological V1 neurons (Fig. 8). At high contrasts $(50\%, 100\%)$, responses exhibit shorter onset latencies and higher peak firing rates compared to lower contrasts $(15\%, 25\%)$. Response peaks also occur earlier in time at higher contrasts, with peak timing progressively delayed as contrast decreases. Quantitatively, we observe a mean latency difference of approximately $0.15$ (in units of relative grating cycles) between the highest $(100\%)$ and lowest $(15\%)$ contrast stimuli, with intermediate contrasts showing progressively longer delays as contrast decreases.

This contrast-dependent latency shift arises naturally from i$\mathcal{P}$-VAE's recurrent dynamics and is likely a direct consequence of the divisive normalization mechanism in eq. (8). Notably, the model was trained solely on static natural images, with no exposure to time-varying stimuli or explicit temporal structure.

## C.4 Reconstruction-sparsity with deterministic MAP decoding, and dependence on $\beta$

In the main text (section 5), we explored the reconstruction-sparsity trade-off, shown in Fig. 4. We found that all iterative VAEs outperformed their amortized counterparts, and that both LCA and i$\mathcal{P}$-VAE achieved the best overall performance (Fig. 4b).

Here we note an important distinction in inference methods: LCA performs *maximum a posteriori* (MAP) estimation [39], while the VAE models maintain uncertainty in their posteriors by sampling. For a more direct comparison, we also evaluated all VAE models using MAP decoding instead of sampling. Specifically, for the Poisson family, we used firing rates directly for reconstruction (instead

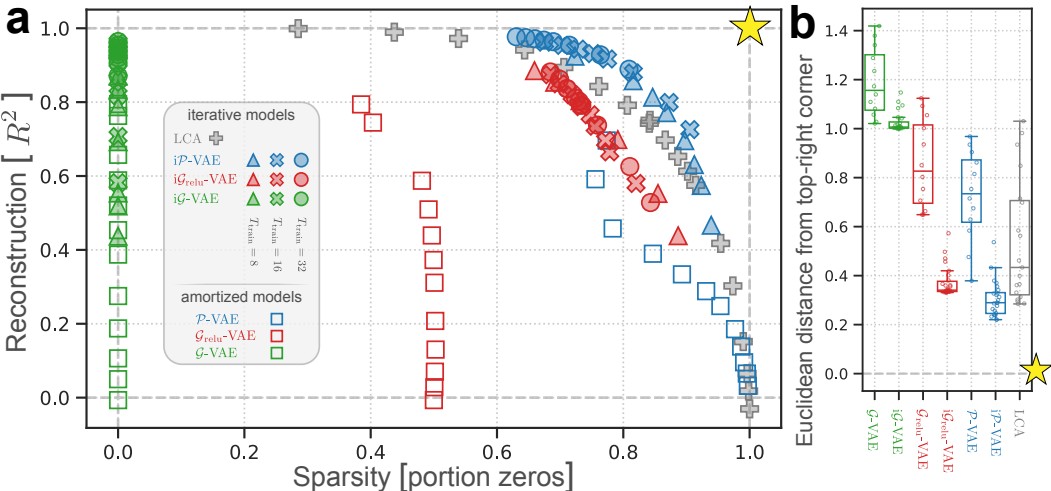

Figure 9: Related to Fig. 4, but using *maximum a posteriori* (MAP) decoding instead of sampling for all VAE-based models. For Poisson models, this means using firing rates directly for reconstruction, while for Gaussian models, we use the posterior means. This approach provides a more direct comparison to LCA, which performs MAP estimation by design [39]. Under these conditions, i$\mathcal{P}$-VAE slightly outperforms LCA while maintaining the benefits of a probabilistic representation.

of spike-count samples), while for the Gaussian family, we used the posterior mean parameters. Note that for iterative VAEs, the inference dynamics was still fully stochastic (eqs. (6), (46) and (49)). The difference here was that for MAP decoding, we simply used the final posterior parameters (instead of samples drawn from it) for the reconstruction.

**MAP decoding.** The results are presented in Fig. 9. As expected, all VAE models improve their reconstruction $R^2$ performance under MAP decoding. The best overall reconstruction is achieved by i$\mathcal{G}$-VAE and i$\mathcal{P}$-VAE with a large $T_{\text{train}} = 32$, approaching $R^2 \approx 1.0$. Notably, i$\mathcal{P}$-VAE achieves this high reconstruction quality while maintaining considerable sparsity levels exceeding 60%, and slightly outperforms LCA in this MAP decoding setting. This result demonstrates that our probabilistic framework can match or exceed the performance of established sparse coding algorithms while maintaining the additional benefits of a fully probabilistic, integer spike-count representation.

**The effects of varying $\beta$.** We also analyze the relationship between the $\beta$ hyperparameter and i$\mathcal{P}$-VAE performance in Fig. 10. The results confirm the theoretical prediction that for Poisson models, the KL divergence term effectively functions as a metabolic cost penalty [50], with higher $\beta$ values inducing greater sparsity. This relationship establishes a direct link between information rate (KL divergence) and our heuristic "proportion zeros" measure of sparsity, partially justifying the interpretation of the reconstruction-sparsity landscape of Fig. 4 as a rate-distortion map [77].

Interestingly, the relationship between $\beta$ and model performance is significantly influenced by the training duration $T_{\text{train}}$. As shown in Fig. 10, models trained with longer inference trajectories ($T_{\text{train}} = 32$) achieve better reconstruction across all $\beta$ values compared to those with shorter training iterations ($T_{\text{train}} = 8$), which tend to be sparser. The overall performance metric (distance from the optimal point, marked by a gold star) reveals different patterns: $T_{\text{train}} = 32$ models show steady improvement that plateaus at higher $\beta$ values, while models with shorter training durations ($T_{\text{train}} = 8, 16$) exhibit a U-shaped curve, initially improving but then degrading as $\beta$ increases beyond an optimal point.

These results suggest that extended inference during training enhances the model's ability to find optimal representations, even when evaluated at a fixed inference length during testing. Our approach bears similarities to meta-learning [191], where the inner loop of optimization affects the outer loop performance. Future work should further explore this dependency on inner-loop iterations to better understand the mechanisms behind these performance differences across training regimes.

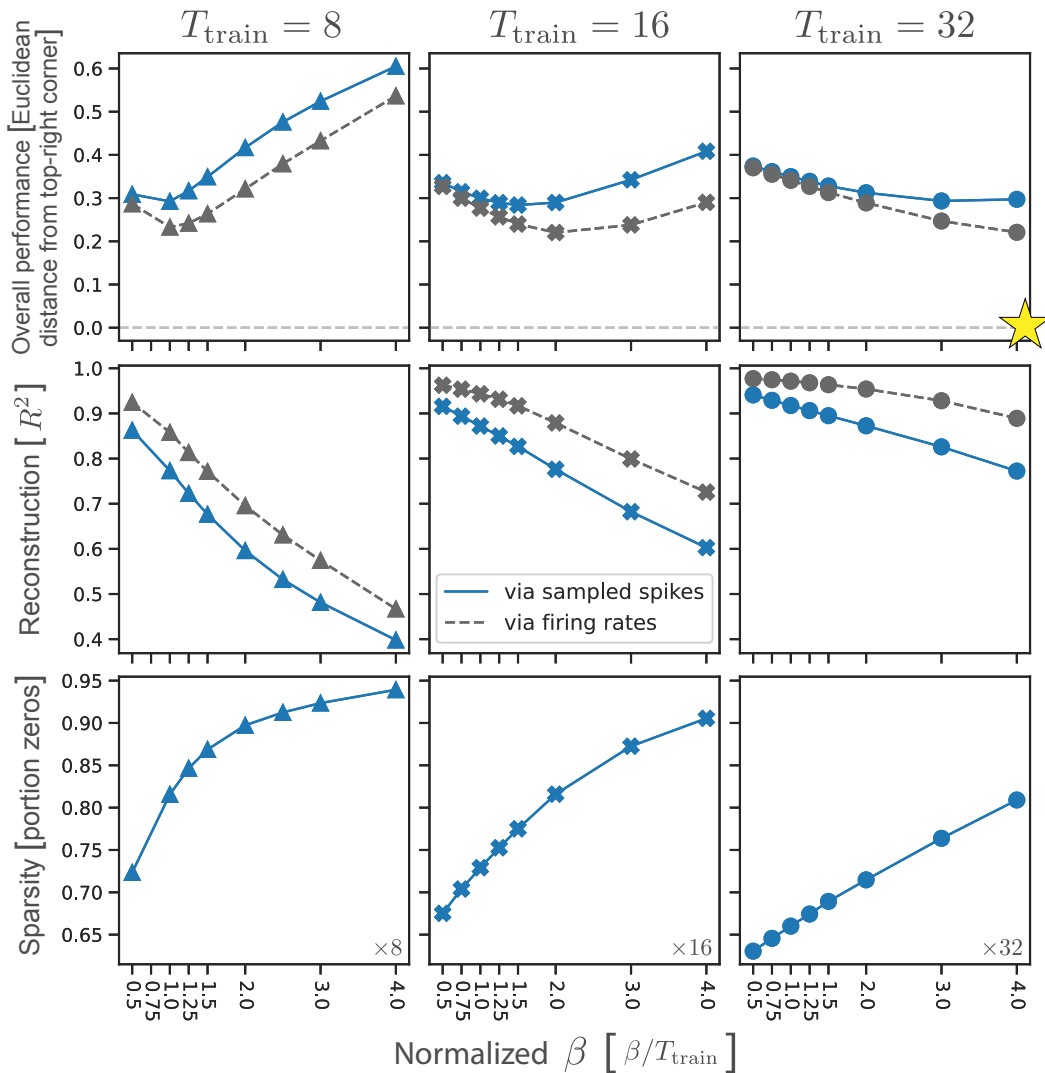

Figure 10: Effect of $\beta$ on i$\mathcal{P}$-VAE performance metrics across different training durations. Each column represents a different training duration ($T_{\text{train}} \in \{8, 16, 32\}$), with performance metrics shown as a function of normalized $\beta$ (scaled by $T_{\text{train}}$). **Top row:** Overall performance improves with higher $T_{\text{train}}$, with longer training allowing higher $\beta$ values before performance degrades. **Middle row:** Reconstruction quality ($R^2$) decreases with higher $\beta$ across all training durations, but MAP decoding (using firing rates, gray dashed line) consistently outperforms sampling-based decoding (blue solid line). **Bottom row:** Sparsity monotonically increases with $\beta$ for all $T_{\text{train}}$ values, confirming the direct relationship between the KL penalty and representational sparsity for Poisson latents [50]. The gold star (defined in Fig. 4) indicates the theoretical optimum: perfect reconstruction ($R^2 = 1.0$) with maximum sparsity (portion zeros $= 1.0$). See Figs. 4 and 9 for a comparison with LCA and other iterative and amortized VAEs, embedded in a shared sparsity-reconstruction landscape.

### C.5 Reconstruction-sparsity performance across latent dimensionalities

In addition to $\beta$ (Fig. 10), latent dimensionality $K$ also influences the reconstruction-sparsity trade-off. To systematically evaluate this effect, we trained i$\mathcal{P}$-VAE models on $16 \times 16$ natural image patches (van Hateren dataset) with linear decoders, varying $K \in \{32, 64, 128, 256, 512, 768, 1024, 2048\}$. All models used $T_{\text{train}} = 16$ and $\beta = 24$, matching the configuration from the model shown in Fig. 3.

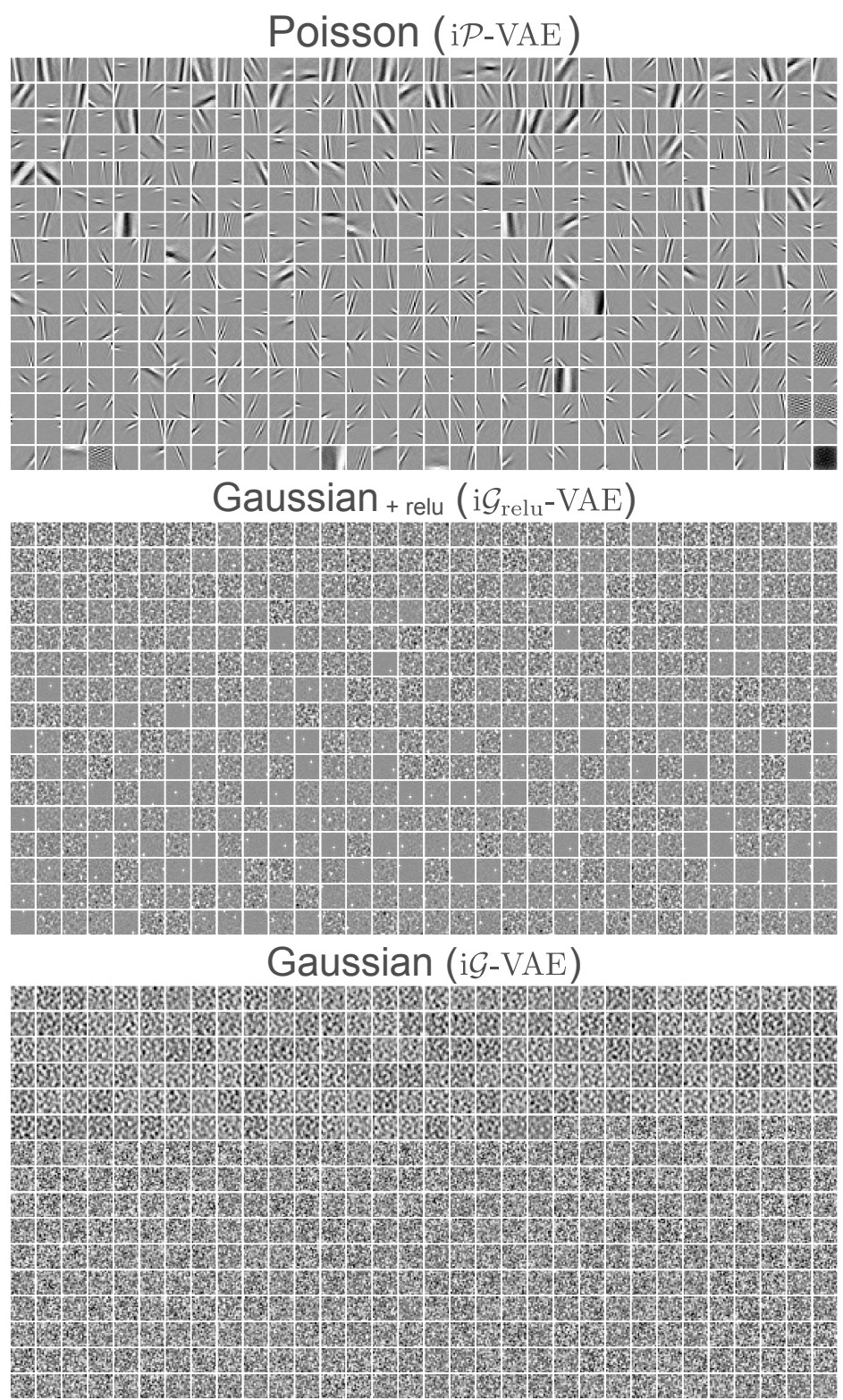

Figure 11: The full set of $K = 512$ dictionary elements, corresponding to the models from Fig. 3. Features are ordered in ascending order of their aggregate posterior membrane potentials (i.e., the final $u_{t=1000}$ for i$\mathcal{P}$-VAE, and $\mu_{t=1000}$ for Gaussian models, averaged over the entire test set).

Table 2: Effect of latent dimensionality ($K$) on reconstruction and sparsity. The i$\mathcal{P}$-VAE model from the main paper used $K = 512$ (highlighted), which balances performance and sparsity.

| $K$ | $R^2 \uparrow$ | Sparsity $\uparrow$ | Overall perf. $\downarrow$ | Active neurons |
|---|---|---|---|---|
| 32 | 0.24 | 0.28 | 1.05 | 23 |
| 64 | 0.42 | 0.28 | 0.93 | 46 |
| 128 | 0.64 | 0.32 | 0.77 | 87 |
| 256 | 0.77 | 0.55 | 0.51 | 116 |
| 512 | 0.83 | 0.77 | 0.28 | 116 |
| 768 | 0.83 | 0.86 | 0.22 | 110 |
| 1024 | 0.83 | 0.90 | 0.20 | 107 |
| 2048 | 0.84 | 0.95 | 0.17 | 103 |

We report reconstruction quality ($R^2$), sparsity (portion zeros), overall performance—Euclidean distance from the ideal point: $R^2 = 1$, sparsity $= 1$, golden star in Fig. 4—and the average number of active neurons, defined as $K \times (1 - \text{portion\_zeros})$. Results are shown in Table 2.

We observe several key patterns:

1. **Overcompleteness improves performance:** Overall performance consistently improves as $K$ increases, with most gains occurring between $K = 32$ and $K = 512$. Performance saturates around $K = 1024$, with diminishing returns beyond this point.

2. **Efficient coding emerges:** The total number of active neurons plateaus around $\sim 110$, even as latent dimensionality grows to 2048. This suggests the model learns an efficient sparse code that maintains a consistent activity budget regardless of available capacity.

3. **Reconstruction-sparsity trade-off:** Beyond $K = 512$, further increases in dimensionality primarily improve sparsity (from 77% to 95% zeros) while reconstruction quality remains stable ($R^2 \approx 0.83$–$0.84$). This validates our choice of $K = 512$ for the main experiments as a reasonable balance point.

These results demonstrate that i$\mathcal{P}$-VAE exhibits the expected overcomplete sparse coding behavior while maintaining computational efficiency through consistent use of approximately 100 active neurons across the natural image patch dataset.

### C.6 Reconstruction performance and downstream classification accuracy on MNIST

In this appendix, we explore the reconstruction fidelity and downstream classification accuracy of iterative and amortized VAEs on MNIST, comparing them to predictive coding networks (PCNs). Specifically, we compare to the original PC model of Rao and Ballard [22], and a recent enhanced version, incremental PC (iPC; Salvatori et al. [90]). See Table 1 for a summary of model details.

### C.6.1 Temporal scalability: i$\mathcal{P}$-VAE scales consistently with training depth on MNIST

As we saw in the main text (section 5, Fig. 4), iterative VAE performance strongly depends on $T_{\text{train}}$. Therefore, we first performed a systematic search to select an appropriate $T_{\text{train}}$ for different iterative VAE models on MNIST, exploring $T_{\text{train}} \in \{4, 8, 16, 32, 64, 128\}$.

We evaluated these models using a consistent $T_{\text{test}} = 1,000$ and examined reconstruction performance. We observed markedly different behavior across variants. i$\mathcal{G}$-VAE showed continuous improvement up until $T_{\text{train}} = 64$, but plateaued thereafter with negligible gains from doubling to $T_{\text{train}} = 128$. i$\mathcal{G}_{\text{relu}}$-VAE initially improved but began to deteriorate and diverge beyond $T_{\text{train}} = 32$. We suspect this instability stems from the absence of the divisive normalization mechanism that naturally emerges in i$\mathcal{P}$-VAE theory (eq. (8)).

Notably, i$\mathcal{P}$-VAE was the only model that exhibited consistent improvements as we increased $T_{\text{train}}$. It likely would continue improving with even longer training sequences, but the computational cost becomes prohibitive beyond what we tested ($T_{\text{train}}$ can be conceptualized roughly as the number of layers in a deep ResNet, see Fig. 5).

In conclusion, i$\mathcal{P}$-VAE demonstrated the most promising scalability characteristics (at least on MNIST). Based on these results, we selected $T_{\text{train}} = 128, 64, 32$ for i$\mathcal{P}$-VAE, i$\mathcal{G}$-VAE, i$\mathcal{G}_{\text{relu}}$-VAE, respectively, for the MNIST experiments reported below.

### C.6.2 Iterative VAEs with an overcomplete latent space outperform PCNs on MNIST

We trained unsupervised models on MNIST and evaluated reconstruction quality using normalized per-dimension MSE, following prior work [113]. Additionally, as in the main text, we report $R^2$, a dimensionality-independent metric bounded between 0 and 1. To evaluate downstream classification performance, we extracted latent representations and trained a logistic regression classifier. Results are shown in Table 3.

Iterative VAEs substantially outperform both PCNs and amortized VAEs in reconstruction fidelity. i$\mathcal{P}$-VAE achieves the lowest per-dim MSE ($4.39 \times 10^{-3}$) and highest $R^2$ (0.951) among all models, significantly surpassing both PC (MSE: $9.24 \times 10^{-3}$, $R^2$: 0.896) and iPC (MSE: $9.08 \times 10^{-3}$, $R^2$: 0.898). Remarkably, i$\mathcal{P}$-VAE accomplishes this favorable reconstruction while maintaining high sparsity (84% zeros), whereas PCNs produce entirely dense representations.

For classification, $\mathcal{P}$-VAE achieved the highest accuracy of 97.89% on downstream unsupervised classification, approaching the performance of supervised PCNs reported in ref. [113]. i$\mathcal{P}$-VAE achieves the second-best classification performance at 96.46%, substantially outperforming both PC (90.74%) and iPC (91.60%).

**Limitations of the PCN comparisons.** We acknowledge several important limitations in our comparative analysis. All iterative and amortized VAE models in this experiment used a single linear decoder, while amortized models employed convolutional encoders. In contrast, PCNs utilize hierarchical architectures with fundamentally different design principles. Rather than conducting a comprehensive architectural optimization for PCNs, we used default settings from the official implementations released by Pinchetti et al. [113], with only minor adjustments (e.g., doubling the number of epochs to ensure convergence, or matching latent dimensionality to other models – see appendix C.1 for more details).

These substantial architectural differences limit the conclusiveness of our PCN versus iterative VAE comparisons presented in Table 3. A more rigorous comparison would require carefully controlling for network capacity and depth. Additionally, our evaluation metrics may inherently favor certain architectural choices. For instance, reconstruction metrics advantage models with overcomplete latent representations with direct pathways between the latents and reconstruction spaces.

Despite these methodological limitations, our results provide valuable initial evidence that iterative VAEs can achieve comparable or superior performance to PCNs in both reconstruction quality and downstream classification tasks. Particularly noteworthy is that i$\mathcal{P}$-VAE accomplishes this while maintaining high sparsity (84% zeros) and using integer-valued spike count representations that better align with biological neural computation. Future work should explore more controlled comparisons across these model families, including matching parameter counts, architectural depth, and computation budgets to establish more definitive performance relationships.

### C.7 Runtime comparison: iterative versus amortized inference

A primary motivation for amortized inference is computational efficiency: a single forward pass through an encoder network should be faster than iterative optimization. However, as we saw in Fig. 4, this potential speed advantage comes at the cost of inferior reconstruction-sparsity performance. Here we quantify the actual runtime trade-off to determine whether iterative models' performance gains justify their computational costs—and whether they are even slower at all.

**Experimental setup.** We compared the iterative VAE models from Fig. 3 ($T_{\text{train}} = 16$) against amortized counterparts trained with standard ELBO ($\beta = 1$). All models used linear decoders trained on $16 \times 16$ natural image patches (van Hateren dataset), with amortized models employing 6-layer convolutional encoders.

We measured inference time using CUDA events for accurate GPU synchronization:

Table 3: Comparison of reconstruction performance, representational sparsity, and downstream classification accuracy on MNIST across model families. Iterative VAEs consistently outperform both PCNs and amortized VAEs in reconstruction metrics while using significantly fewer parameters than their amortized counterparts. Bold values indicate best performance in each category. Amortized VAE models used deep ResNet convolutional encoders, and all VAE models (both iterative and amortized) used a single linear decoder for a consistent comparison with the main text results. Iterative VAE models were trained with optimized iteration counts: $T_{\text{train}} = 128$ for i$\mathcal{P}$-VAE, $T_{\text{train}} = 64$ for i$\mathcal{G}$-VAE, and $T_{\text{train}} = 32$ for i$\mathcal{G}_{\text{relu}}$-VAE, based on convergence properties. Mean squared error (MSE) values are normalized per dimension for consistent comparison with previous benchmarks [113]. Classification accuracy is reported using a linear classifier trained on unsupervised features. Values are mean $\pm$ 99% confidence interval, estimated from 5 random initializations. See appendix C.6.1 for analysis of how $T_{\text{train}}$ affects performance, and appendix C.6.2 for detailed interpretation.

| Model family | Model | Reconstruction | | Portion Zeros ↑ | Classification Accuracy ↑ | Num Params ↓ |
|---|---|---|---|---|---|---|
| | | MSE$_{\times 10^{-3}}$ ↓ | $R^2$ ↑ | | | |
| iterative VAE | i$\mathcal{P}$-VAE | **4.39**$_{\pm.04}$ | **.951**$_{\pm.000}$ | .84$_{\pm.00}$ | 96.46$_{\pm.11}$ | 0.40 $M$ |
| | i$\mathcal{G}$-VAE | 7.63$_{\pm.02}$ | .914$_{\pm.000}$ | .00$_{\pm.00}$ | 92.73$_{\pm.14}$ | 0.40 $M$ |
| | i$\mathcal{G}_{\text{relu}}$-VAE | 8.45$_{\pm.92}$ | .906$_{\pm.011}$ | .78$_{\pm.05}$ | 95.74$_{\pm.43}$ | 0.40 $M$ |
| amortized VAE | $\mathcal{P}$-VAE | 25.48$_{\pm.24}$ | .713$_{\pm.002}$ | .84$_{\pm.00}$ | **97.89**$_{\pm0.34}$ | 6.79 $M$ |
| | $\mathcal{G}$-VAE | 14.40$_{\pm.19}$ | .841$_{\pm.002}$ | .00$_{\pm.00}$ | 95.75$_{\pm2.19}$ | 6.93 $M$ |
| | $\mathcal{G}_{\text{relu}}$-VAE | 17.13$_{\pm.20}$ | .809$_{\pm.002}$ | .50$_{\pm.00}$ | 95.41$_{\pm1.51}$ | 6.93 $M$ |
| PCN | PC [22] | 9.24$_{\pm.03}$ | .896$_{\pm.000}$ | .00$_{\pm.00}$ | 90.74$_{\pm.26}$ | **0.13** $M$ |
| | iPC [90] | 9.08$_{\pm.00}$ | .898$_{\pm.000}$ | .00$_{\pm.00}$ | 91.60$_{\pm.10}$ | **0.13** $M$ |

```
start = torch.cuda.Event(enable_timing=True)
end = torch.cuda.Event(enable_timing=True)

start.record()
output = model(x)
end.record()

torch.cuda.synchronize()
duration = start.elapsed_time(end)
```

Since iterative methods distribute computation across many timesteps, they may underutilize available parallelism for small batches but benefit substantially from large-batch parallelization. We therefore evaluated two extreme scenarios: single-sample inference (batch size = 1) and large-batch inference (batch size = 10,000).

### C.7.1 Inference runtime results: iterative models match or exceed amortized speed.

Figure 12 shows reconstruction quality, sparsity, and overall performance as functions of wall-clock time. The key question is: how long does it take iterative models to reach the same overall performance as amortized models? The answer challenges conventional assumptions:

**Small batches (batch size = 1):** Amortized models complete inference in $\sim$3ms (marked by X's). Iterative models reach equivalent performance (distance from optimal point) within 3–15ms. Iterative models then continue improving, converging to superior final performance by $\sim$400ms.

**Large batches (batch size = 10,000):** Parallel computation reveals an unexpected advantage for iterative inference. Amortized models complete inference in $\sim$50ms, but iterative models achieve the same performance (distance from optimal point) in approximately the same duration. After matching amortized performance, iterative models continue refining their representations, converging to substantially better reconstruction-sparsity trade-offs by $\sim$2000ms.

This pattern holds across all three iterative variants: i$\mathcal{P}$-VAE, i$\mathcal{G}$-VAE, and i$\mathcal{G}_{\text{relu}}$-VAE all reach amortized-level performance within comparable speed to their amortized counterparts when batch size

is large. The iterative models then leverage additional compute to achieve superior final performance that amortized models cannot match.

**Training time considerations.** While inference speed is competitive between the two VAE types, training iterative models requires more time. As noted in appendix C.1.4, training iterative VAEs with $T_{\text{train}} = 16$ takes $\sim 3$ hours—approximately twice as long as amortized models with 6-layer encoders. This reflects the effective depth of iterative models: backpropagation through time across $T_{\text{train}}$ steps creates computational graphs comparable to deep ResNets (Fig. 5). Training costs scale with both $T_{\text{train}}$ and dataset size, though inference remains efficient.

**Implications.** These results challenge the assumption that iterative inference is prohibitively slow. For small batches, iterative models match amortized performance at slightly slower speeds. For large batches, iterative models are actually *comparable* at reaching equivalent performance levels, then continue improving to achieve superior reconstruction-sparsity trade-offs. Combined with their better out-of-distribution generalization (appendix C.8) and $25\times$ parameter efficiency, iterative models offer a compelling alternative to amortized inference. The increased training time appears to be the only meaningful trade-off, making iterative models particularly attractive for applications where inference performance matters more than training speed.

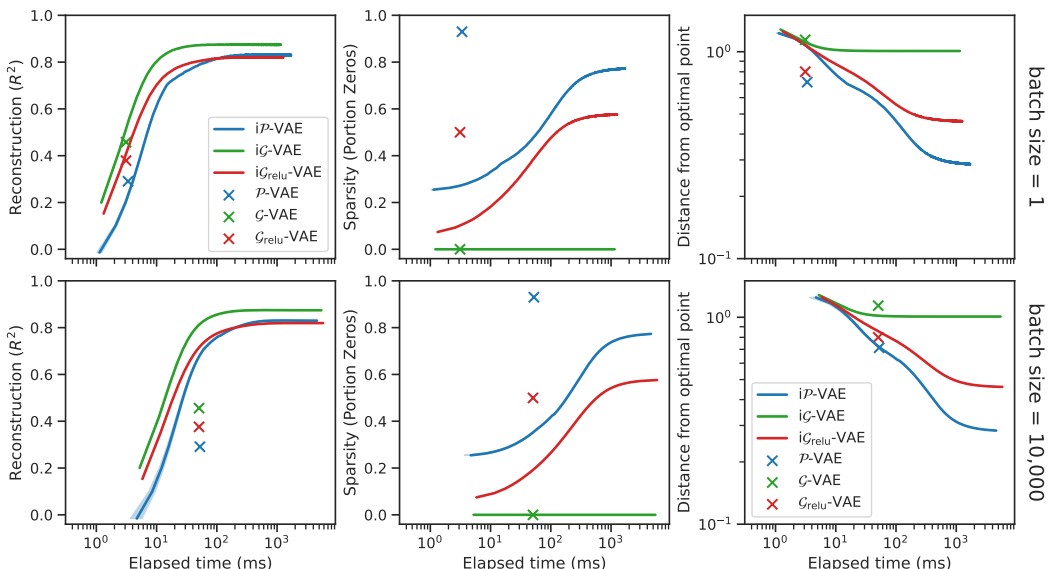

Figure 12: Runtime comparison between iterative and amortized VAEs for batch sizes of 1 (top) and 10,000 (bottom). Iterative models (solid lines) progressively improve over time, while amortized models (X markers) produce immediate but inferior results. **Left**: Reconstruction quality ($R^2$). **Middle**: Sparsity (portion zeros). **Right**: Overall performance (distance from optimal point, lower is better; see Fig. 4). For large batches, iterative models reach amortized-level performance at comparable speeds to amortized models. Iterative models then continue improving to achieve superior final performance across all metrics by $\sim 200$–$300$ms.

## C.8 i$\mathcal{P}$-VAE exhibits strong out-of-distribution (OOD) generalization

In the main paper, we compared our novel iterative VAE architectures, including the i$\mathcal{P}$-VAE, to their amortized counterparts, as well as neuroscience models such as sparse coding [39] and predictive coding networks (PCNs; [22, 90]). With the exception of PCNs, all other models we tested in the main paper employed a single linear layer decoder ($\hat{x} = \Phi z$, Fig. 5). In appendix B.6, we extended our theoretical framework to nonlinear decoders by deriving inference algorithms that incorporate the decoder's Jacobian (eq. (40), Fig. 6).

This appendix presents experiments with nonlinear decoders, where we replace the linear $\Phi$ with a deep neural network: $\hat{x} = \text{dec}(z; \theta)$, parameterized by $\theta$. We compare i$\mathcal{P}$-VAE to $\mathcal{P}$-VAE and hybrid iterative-amortized VAEs across various generalization tasks.

**Tasks and Datasets.** We evaluate performance primarily based on out-of-distribution (OOD; [98, 99]) generalization, measured through both reconstruction fidelity and downstream classification accuracy. We use MNIST as our primary training dataset, and test generalization on rotated MNIST, extended MNIST (EMNIST; [224]), Omniglot [225], and ImageNet32 [226], with the latter two resized to $28 \times 28$ for consistent comparisons.

**Architecture notation.** We experimented with both convolutional and multi-layer perceptron (MLP) architectures, denoted using the $\langle \text{enc}|\text{dec} \rangle$ convention. For example, $\langle \text{mlp}|\text{mlp} \rangle$ indicates both encoder and decoder networks are MLPs. We use $\langle \text{grad}|\text{mlp} \rangle$ to denote our fully iterative (non-amortized) i$\mathcal{P}$-VAE, which replaces the encoder neural network with natural gradient descent on variational free energy (section 4 and appendix B.6). We designed symmetrical architectures, ensuring that $\langle \text{mlp}|\text{mlp} \rangle$ has exactly twice as many parameters as $\langle \text{grad}|\text{mlp} \rangle$.

**Alternative models.** To isolate the impact of inference method in OOD settings, we compare i$\mathcal{P}$-VAE to the amortized $\mathcal{P}$-VAE and state-of-the-art hybrid methods that combine iterative with amortized inference: iterative amortized VAE (ia-VAE; [91]) in both hierarchical (h) and single-level (s) variants, and semi-amortized VAE (sa-VAE; [92]).

**Inference iterations.** For i$\mathcal{P}$-VAE, we explored different training iteration counts, $T_{\text{train}} \in \{4, 16, 32, 64\}$. Recall that during training, we differentiate through the entire sequence of iterations (learning to infer), which can lead to qualitatively different dynamics. For most OOD experiments, we use a $\langle \text{grad}|\text{mlp} \rangle$ i$\mathcal{P}$-VAE with $T_{\text{train}} = 64$. At test time, we evaluate with $T_{\text{test}} = 1,000$ iterations. For the hybrid models, we use their default iteration counts unless otherwise specified (sa-VAE: $T_{\text{train}} = T_{\text{test}} = 20$, ia-VAE: $T_{\text{train}} = T_{\text{test}} = 5$).

### C.8.1 Stability beyond the training regime and convergence for nonlinear decoders

An algorithm with strong generalization potential should learn inference dynamics that extend beyond the training regime. In the main paper (section 5), we found that all iterative VAEs converge when trained on natural image patches with linear decoders (Fig. 3). Here, we investigate whether this convergence holds for the more challenging case of nonlinear decoders.

We trained models on MNIST using different numbers of training iterations ($T_{\text{train}} \in \{4, 16, 32, 64\}$) with both $\langle \text{grad}|\text{mlp} \rangle$ and $\langle \text{grad}|\text{conv} \rangle$ architectures, then tested whether they continue to improve beyond $T_{\text{train}}$. As shown in Fig. 13a, even i$\mathcal{P}$-VAE models trained with as few as $T_{\text{train}} = 4$ iterations learn to keep improving during test time, eventually converging to stable solutions. Interestingly, models trained with more iterations initially show worse performance but ultimately converge to better solutions, suggesting that i$\mathcal{P}$-VAE learns temporally-extended inference dynamics that depend on the training sequence length but generalize well beyond it.

In contrast, hybrid models (sa-VAE and ia-VAE) begin with strong amortized initial guesses but plateau rapidly (Fig. 13a, right), converging to substantially higher MSE than i$\mathcal{P}$-VAE models despite having more parameters. We note that sa-VAE's authors acknowledged issues with the dominance of the iterative component on Omniglot and reported using techniques like gradient clipping to mitigate this (see footnote 6 in [92]), which may explain our observations on MNIST. More concerning, ia-VAE (single-level) begins to diverge outside its training regime and frequently resulted in numerical instabilities at test time when exceeding $T_{\text{train}}$.

Overall, i$\mathcal{P}$-VAE demonstrates better reconstruction performance and continues to improve outside the training regime, unlike other models. This extends our convergence results from linear decoders (Fig. 3) to the more general nonlinear case and reveals i$\mathcal{P}$-VAE's first form of OOD generalization: temporal generalization beyond the training horizon. Next, we examine whether i$\mathcal{P}$-VAE can generalize across visual domains.

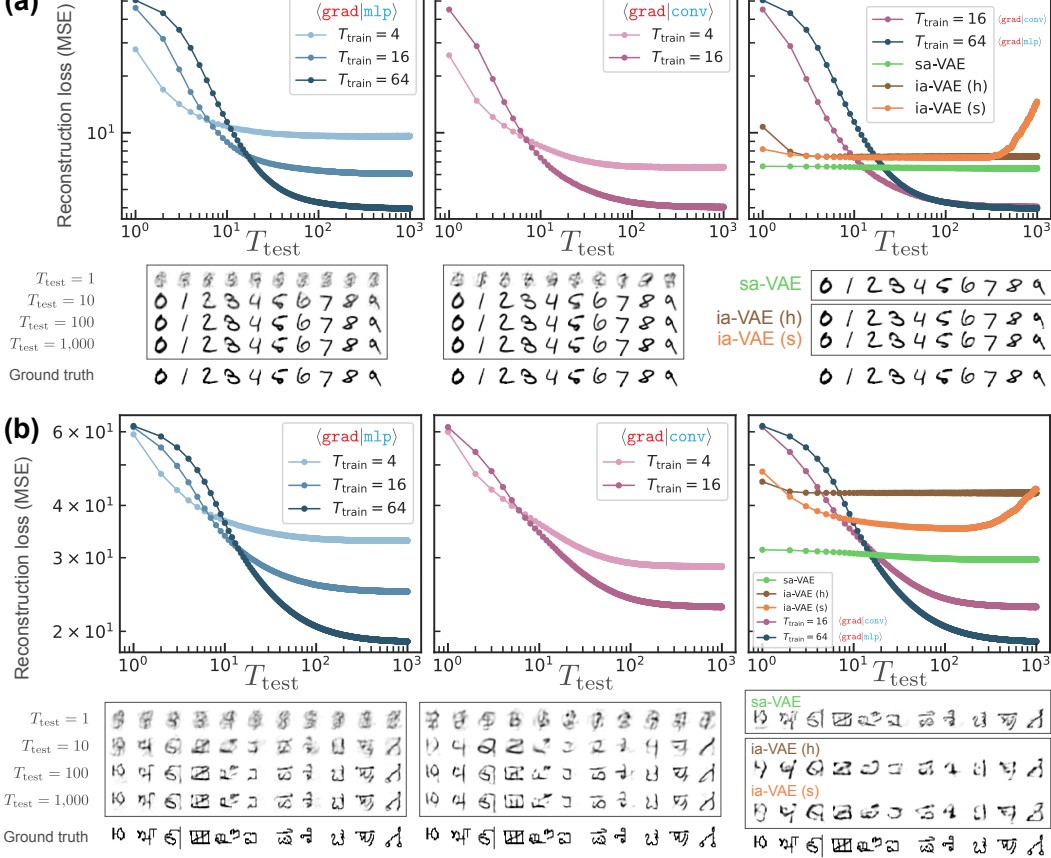

Figure 13: i$\mathcal{P}$-VAE generalizes beyond its training regime and across datasets. All models shown here are trained on MNIST and tested either on MNIST (panel a) or Omniglot (panel b). **(a)** i$\mathcal{P}$-VAE models trained with as few as $T_{\text{train}} = 4$ inference steps continue to improve well beyond the training regime ($T_{\text{test}} > T_{\text{train}}$). This temporal generalization holds across architectures: $\langle\text{grad}|\text{mlp}\rangle$ (left) and $\langle\text{grad}|\text{conv}\rangle$ (middle). In contrast, hybrid iterative-amortized models either plateau or diverge outside their training regime (right). **(b)** i$\mathcal{P}$-VAE trained exclusively on MNIST successfully generalizes to the structurally different Omniglot dataset, showing substantial improvement in reconstruction quality as inference iterations increase. With higher $T_{\text{train}}$, i$\mathcal{P}$-VAE models show initially worse but ultimately better performance, suggesting they learn more complex but effective inference dynamics.

### C.8.2    Out-of-distribution generalization

We evaluate two levels of generalization for MNIST-trained models: (1) within-dataset perturbations, and (2) transfer across related datasets.

**OOD generalization to within-dataset perturbation.**    We tested whether models trained on standard MNIST could generalize to rotated MNIST digits [98, 227]. At test time, we rotate digits between $0°$ and $180°$ in increments of $15°$. We evaluated (a) reconstruction quality of the rotated digits, and (b) whether the resulting latent representations support downstream classification (Fig. 14).

i$\mathcal{P}$-VAE and sa-VAE maintained relatively consistent performance across rotation angles, though i$\mathcal{P}$-VAE demonstrated superior performance in both reconstruction and classification. The amortized $\mathcal{P}$-VAE showed substantially worse reconstruction than iterative models, but its classification accuracy remained remarkably stable across angles, outperforming all models except i$\mathcal{P}$-VAE.

In contrast, both ia-VAE variants showed significant degradation in performance as the rotation angle increased, with substantial drops in both reconstruction fidelity and classification accuracy. Overall, i$\mathcal{P}$-VAE maintained the most stable and superior performance across all rotation angles, suggesting it learns rotation-invariant features that generalize effectively to unseen transformations.

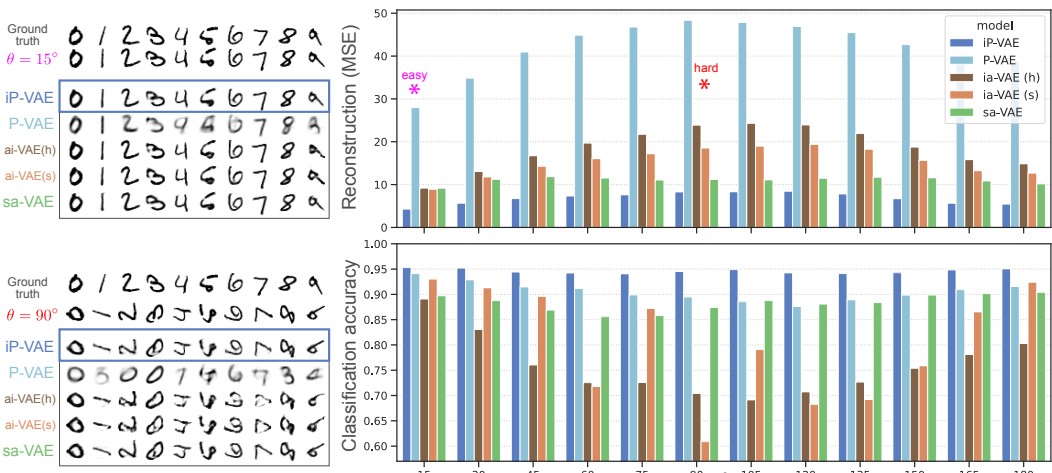

Figure 14: Robustness to rotational transformations of MNIST digits. **Left:** Visual comparison of reconstructions for moderately rotated ($\theta = 15°$) and severely rotated ($\theta = 90°$) digits across models. i$\mathcal{P}$-VAE maintains high reconstruction fidelity across rotation angles, while other models show progressive degradation. **Right:** Quantitative evaluation of reconstruction MSE (top) and downstream classification accuracy (bottom) across rotation angles from $0°$ to $180°$. i$\mathcal{P}$-VAE demonstrates remarkable stability in both metrics, with minimal performance degradation even at extreme angles. The solid lines represent means, while shaded regions show standard deviations across test samples. All models were trained only on standard (unrotated) MNIST, making this a challenging test of generalization to unseen transformations.

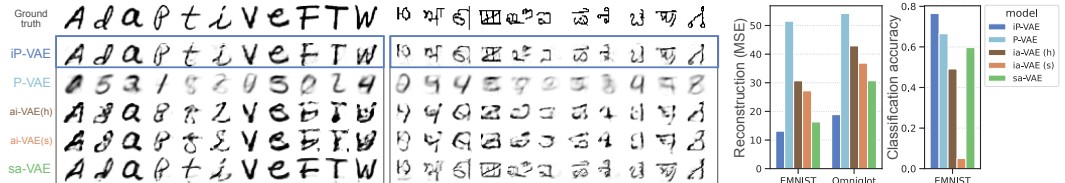

Figure 15: Cross-dataset generalization from MNIST to novel character datasets. **Left panels:** Visual comparison of original samples and their reconstructions for EMNIST (left) and Omniglot (center-left) across different models. All models were trained exclusively on MNIST. **Center-right panel:** Reconstruction MSE for EMNIST and Omniglot. Lower is better. i$\mathcal{P}$-VAE consistently achieves the lowest error on both datasets despite never seeing them during training. **Right panel:** Classification accuracy on EMNIST using a linear classifier trained on latent representations. i$\mathcal{P}$-VAE representations support significantly higher accuracy, suggesting they capture more discriminative features that transfer well to new character classes.

**OOD generalization across similar datasets.**    A model that learns compositional features with an effective inference algorithm should generalize to datasets structurally related to, but distinct from, the training distribution. To test this, we evaluated MNIST-trained models on EMNIST (Fig. 15) and Omniglot (Fig. 13b, Fig. 15), reporting both reconstruction MSE and classification accuracy (except for Omniglot, which has over 1,000 classes, making classification challenging).

i$\mathcal{P}$-VAE consistently demonstrated superior generalization, achieving lower reconstruction error and higher classification accuracy than all other models. The visual quality of its reconstructions on both EMNIST and Omniglot was noticeably better, preserving key structural elements of characters it had never seen during training. This suggests that i$\mathcal{P}$-VAE learns a more general and compositional latent space that captures fundamental structural elements shared across different character datasets.

**Extreme cross-domain generalization with i$\mathcal{P}$-VAE: from MNIST to natural images.**    Most remarkably, a $\langle$`grad`|`mlp`$\rangle$ i$\mathcal{P}$-VAE trained exclusively on MNIST still performs well when tested on cropped, grayscaled natural images from tiny ImageNet (Fig. 16), a domain vastly different

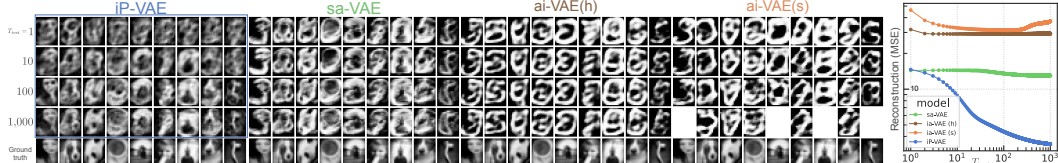

Figure 16: Extreme cross-domain generalization: MNIST-trained models tested on cropped, grayscaled natural images. **Left panels:** Ground truth grayscale ImageNet32 samples (resized to $28 \times 28$) alongside their reconstructions from various models. Despite being trained exclusively on MNIST digits, i$\mathcal{P}$-VAE captures significant semantic information and structural details of natural images. **Right panel:** Reconstruction MSE over inference iterations. i$\mathcal{P}$-VAE continues to improve with more iterations, while hybrid models plateau quickly. The ia-VAE variants fail to adapt to the new domain, while sa-VAE shows moderate adaptation. This challenging generalization task demonstrates i$\mathcal{P}$-VAE's ability to represent visual concepts far outside its training distribution. This capability is likely enabled by its Gabor-like primitive features that can be recombined to represent novel visual structures (Fig. 17).

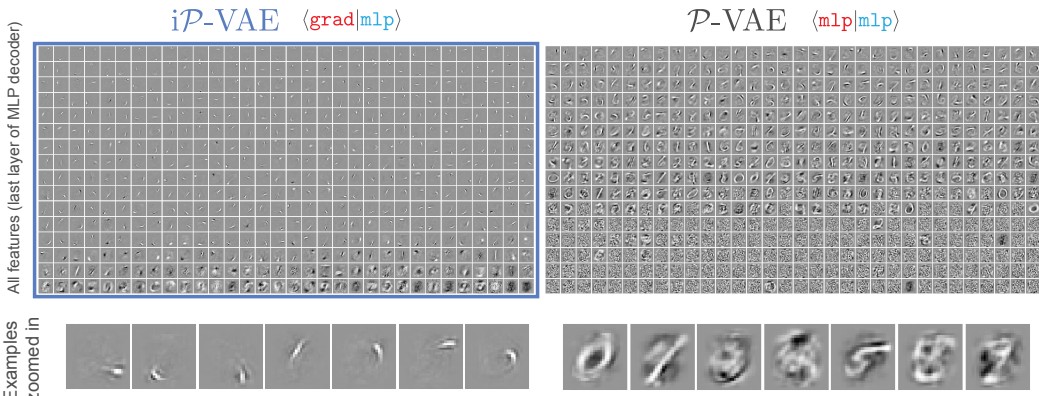

Figure 17: Visualization of learned features in the final decoder layer for i$\mathcal{P}$-VAE with a $\langle\texttt{grad}|\texttt{mlp}\rangle$ architecture (left), versus an $\langle\texttt{mlp}|\texttt{mlp}\rangle$ $\mathcal{P}$-VAE (right), both trained on MNIST. i$\mathcal{P}$-VAE learns Gabor-like features (fundamental visual primitives found in primate early visual cortex [107–110]) rather than digit-specific features. In contrast, $\mathcal{P}$-VAE learns more class-specific features resembling complete digits or digit fragments. Features are ordered by increasing kurtosis of their weight distributions to highlight the sparse nature of i$\mathcal{P}$-VAE's representations. The more general, primitive features learned by i$\mathcal{P}$-VAE likely contribute to its superior generalization capability across datasets, as these features can be recombined to represent novel visual structures not seen during training.

from handwritten digits. Despite never seeing natural images during training, i$\mathcal{P}$-VAE continues to improve its reconstruction quality with increasing inference iterations, capturing significant structural details; whereas, hybrid models like $\langle\texttt{conv}|\texttt{conv}\rangle$ sa-VAE and $\langle\texttt{mlp}|\texttt{mlp}\rangle$ ia-VAE plateau quickly or even diverge (Fig. 16, right panel). This extreme cross-domain transfer challenges common assumptions about generalization boundaries in representation learning and suggests that i$\mathcal{P}$-VAE captures something fundamental about visual structure that transcends specific datasets.

**Primitive visual codes enable generalization.** How does i$\mathcal{P}$-VAE achieve this unexpected generalization capability? We visualized the 512 learned features from the final layer of the $\langle\texttt{grad}|\texttt{mlp}\rangle$ i$\mathcal{P}$-VAE's decoder and compared them to those learned by an amortized $\langle\texttt{mlp}|\texttt{mlp}\rangle$ $\mathcal{P}$-VAE (Fig. 17). The contrast is striking: i$\mathcal{P}$-VAE learns Gabor-like features—fundamental visual primitives similar to those found in primary visual cortex—while $\mathcal{P}$-VAE learns features that resemble entire digits or digit fragments. Previous work has identified strokes as compositional sub-components of digits [228], but i$\mathcal{P}$-VAE learns an even more general and primitive visual code. These elementary Gabor-like features, combined with i$\mathcal{P}$-VAE's adaptive inference process, create a powerful compositional system that can flexibly recombine primitives to represent novel visual structures far outside the training distribution.

### C.8.3 A discussion of the out-of-distribution (OOD) generalization experiments

Empirically, i$\mathcal{P}$-VAE demonstrates promising adaptability and robustness to OOD samples while dynamically balancing computational cost and performance. It consistently outperforms both amortized models and hybrid iterative-amortized VAEs on the tasks we evaluated, despite using substantially fewer parameters.

These results are especially promising given the growing emphasis on OOD generalization in machine learning [98, 99, 227]. i$\mathcal{P}$-VAE 's ability to learn primitive, compositional features that transfer across domains suggests its potential value for representation learning, whether as a pretraining method for downstream tasks or as a backbone for diffusion models [229, 230].

That said, we acknowledge that more systematic evaluation is needed to fully characterize the compositional capacity of iterative VAEs. While our experiments focus primarily on reconstruction and classification, future work should explore more diverse tasks and evaluation metrics. The strong performance demonstrated here motivates a deeper investigation into the generalization capabilities of i$\mathcal{P}$-VAE and its potential applications across a broader range of domains.

### C.8.4 Methodological details for the OOD experiments

For our comparisons, we utilized the official implementations of sa-VAE [92], ia-VAE [91], and $\mathcal{P}$-VAE [50]. Across models, we maintained consistent train/validation splits and hyperparameters unless otherwise specified.

Since sa-VAE's original code used a Bernoulli observation model, we adapted it for Gaussian compatibility by removing the sigmoid in the decoder and replacing its reconstruction loss with MSE. For sa-VAE, we used the default configurations provided for Omniglot (batch size 50, 100 epochs), and adjusted appropriately for other datasets: MNIST (batch size 50, 32 epochs), and EMNIST (batch size 50, 16 epochs).

For ia-VAE, we used the authors' provided configurations for both Bernoulli and Gaussian observation models, as appropriate. The MNIST configuration was applied across our MNIST, EMNIST, and Omniglot experiments. Although the codebase hard-codes training to 2000 epochs, we halted training earlier, once the loss had converged (between 780 and 2000 epochs). In some runs, we encountered numerical instabilities that required restarting training from checkpoints.

Overall, the consistently strong performance of i$\mathcal{P}$-VAE across these generalization tasks suggests significant potential for applications in representation learning and transfer learning. We hope these results motivate further exploration of iterative inference models for representation learning.

### C.9 i$\mathcal{P}$-VAE scales up to complex, high-dimensional color image datasets

Theoretical neuroscience has produced many normative models, including predictive coding [22], sparse coding [39, 54], efficient coding [231], and the free-energy principle [24]. However, from a machine learning standpoint, existing implementations rarely meet modern data complexity requirements. Sparse coding, despite decades of development and widespread recognition of its advantages, still predominantly uses linear decoders and operates on small, grayscale image patches. While we previously demonstrated that i$\mathcal{P}$-VAE can perform *nonlinear* deep sparse coding (appendices B.6, C.8 and C.10), a critical question remains: can it scale to realistic, high-dimensional datasets?

Here we show that i$\mathcal{P}$-VAE successfully scales to modern color image datasets, including CIFAR-10 ($3 \times 32 \times 32$), tiny ImageNet ($3 \times 32 \times 32$), and CelebA ($3 \times 128 \times 128$).

**CelebA (cropped to $128 \times 128$).** We trained four models on CelebA to systematically compare iterative versus amortized inference and deep versus linear decoders:

1. i$\mathcal{P}$-VAE, $\langle \texttt{grad}|\texttt{conv} \rangle$: iterative encoding + 8-layer convolutional decoder
2. i$\mathcal{P}$-VAE, $\langle \texttt{grad}|\texttt{lin} \rangle$: iterative encoding + single-layer convolutional layer decoder
3. $\mathcal{P}$-VAE, $\langle \texttt{conv}|\texttt{conv} \rangle$: convolutional encoder + 8-layer convolutional decoder (amortized counterpart of #1)
4. $\mathcal{P}$-VAE, $\langle \texttt{conv}|\texttt{lin} \rangle$: convolutional encoder + single-layer convolutional decoder (amortized counterpart of #2)

All models used latent dimensionality $256 \times 16 \times 16$, with linear decoders using kernel size 38. We trained with $\beta = 1$ and evaluated reconstruction-sparsity performance. Overall performance is measured as Euclidean distance from the ideal point ($R^2 = 1$, sparsity $= 1$, marked by the gold star in Fig. 4); lower is better. Results are shown in Table 4.

Table 4: Reconstruction-sparsity performance on CelebA (cropped to $128 \times 128$). The $\langle \text{grad} | \text{conv} \rangle$ i$\mathcal{P}$-VAE with deep decoder (model #1) achieves the best overall performance while using 63% fewer parameters than its amortized counterpart (model #3).

| Model | Encoder | Decoder | $R^2 \uparrow$ | Sparsity $\uparrow$ | Performance $\downarrow$ | Params $\downarrow$ |
|---|---|---|---|---|---|---|
| (1) i$\mathcal{P}$-VAE | iterative | 8-layer conv | **0.95** | 0.93 | **0.09** | **0.7 M** |
| (2) i$\mathcal{P}$-VAE | iterative | linear conv | 0.90 | 0.91 | 0.13 | 1.1 M |
| (3) $\mathcal{P}$-VAE | conv | 8-layer conv | 0.86 | 0.96 | 0.15 | 1.9 M |
| (4) $\mathcal{P}$-VAE | conv | linear conv | 0.74 | **0.99** | 0.26 | 2.3 M |

**CIFAR-10 and Tiny ImageNet ($32 \times 32$).** To verify scalability across datasets, we trained a $\langle \text{grad} | \text{conv} \rangle$ i$\mathcal{P}$-VAE with 6 convolutional layers and latent dimensionality $256 \times 8 \times 8$ on both CIFAR-10 and tiny ImageNet ($\beta = 1$). Results are shown in Table 5.

Table 5: Reconstruction-sparsity performance on CIFAR-10 and tiny ImageNet ($32 \times 32$). The i$\mathcal{P}$-VAE consistently achieves strong performance across diverse natural image datasets.

| Dataset | $R^2 \uparrow$ | Sparsity $\uparrow$ | Performance $\downarrow$ |
|---|---|---|---|
| CIFAR-10 | 0.93 | 0.97 | 0.08 |
| Tiny ImageNet | 0.95 | 0.94 | 0.08 |

Across all three datasets, i$\mathcal{P}$-VAE achieves overall performance scores of 0.08–0.09, demonstrating consistent scalability to complex, high-dimensional color images. These results provide strong evidence that normative, brain-inspired models need not be limited to toy datasets: i$\mathcal{P}$-VAE bridges theoretical neuroscience and modern machine learning, achieving superior reconstruction–sparsity trade-offs with fewer parameters than amortized alternatives.

### C.10   i$\mathcal{P}$-VAE generates novel samples through an iterative procedure

The primary focus of this work (and of FOND more broadly) is on brain-like *inference* dynamics, where generation serves as a means to this end (i.e., "analysis-by-synthesis" [68]). Nevertheless, assessing the model's generative behavior provides useful insights into its learned representations.

To this end, we performed a qualitative evaluation of a $\langle \text{grad} | \text{conv} \rangle$ i$\mathcal{P}$-VAE trained on the MNIST dataset with $T_{\text{train}} = 16$. The generation procedure is as follows:

---
**Algorithm 1** Iterative Generation Procedure for i$\mathcal{P}$-VAE
---
1: Sample from initial prior: $z_0 \sim \mathcal{P}\text{ois}(z_0; \exp(u_0))$
2: **for** $t = 1$ to $T$ **do**
3:      Generate image: $\hat{x}_{t-1} \leftarrow f_\theta(z_{t-1})$            ▷ decoder output
4:      Treat generated image as input: $x_t \leftarrow \hat{x}_{t-1}$
5:      Run one inference step: update $u_t$ via eq. (7), where $x \equiv x_t$
6:      Sample new latent: $z_t \sim \mathcal{P}\text{ois}(z_t; \exp(u_t))$
7: **end for**

---

The model successfully produced coherent and recognizable digits (Fig. 18). Notably, during the iterative generation process, some initially ambiguous samples (e.g., resembling both '3' and '6') would converge to well-defined digits after several refinement steps. This progressive clarification mirrors the convergence behavior observed during inference (Fig. 3), suggesting a unified dynamical principle underlying both processes.

**(a)** Sample generation process across time (using a $\langle$grad$|$conv$\rangle$ i$\mathcal{P}$-VAE )

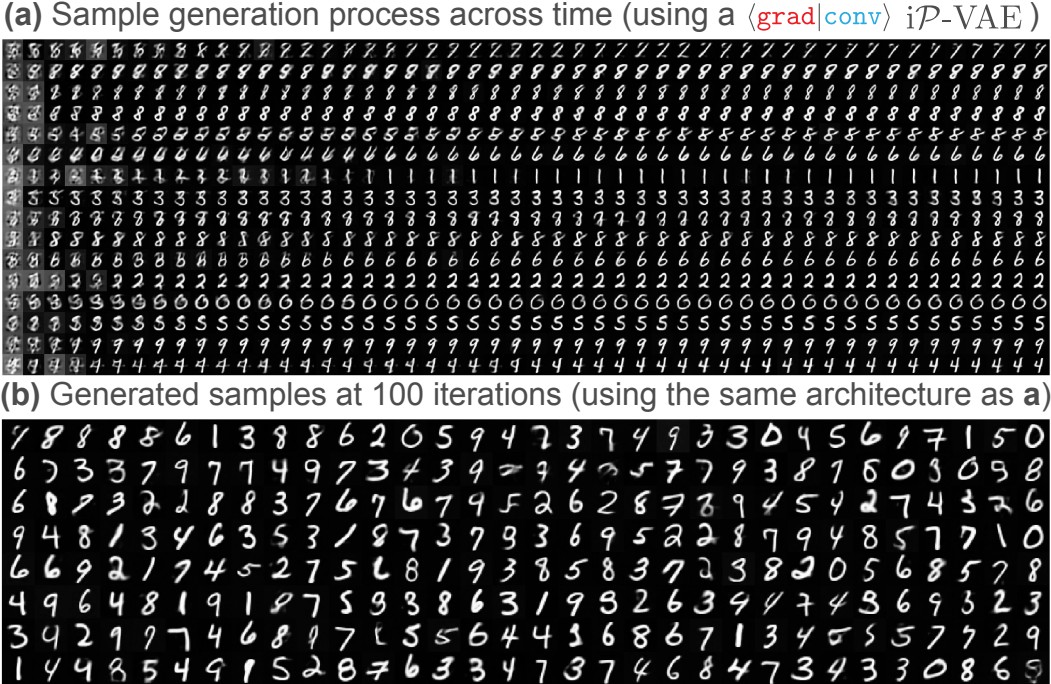

**(b)** Generated samples at 100 iterations (using the same architecture as **a**)

Figure 18: Generation results from a $\langle$grad$|$conv$\rangle$ i$\mathcal{P}$-VAE trained on MNIST, sampled using Algorithm 1. **(a)** Temporal dynamics of generation: horizontal axis shows iteration steps, vertical axis shows different sampled trajectories. **(b)** Final generated samples after 100 iterations.

However, we observed modest class imbalance in generated samples: certain digits (e.g., '8') appeared more frequently than their $\sim 10\%$ training prevalence. This bias likely reflects the learned prior distribution $\exp(\boldsymbol{u}_0)$ and could be mitigated through more expressive hierarchical priors.

## D  Extended Discussion

This appendix expands on key topics introduced in the main paper's discussion (section 6). We begin by summarizing FOND and discuss how it relates to existing literature (appendix D.1). We then analyze possible factors underlying i$\mathcal{P}$-VAE's effectiveness (appendix D.2), assess its potential for hardware implementation (appendix D.3), and explore relationships to sampling-based inference methods (appendix D.4).

We conclude with a discussion of limitations and future directions (appendix D.5). These include: accelerating iterative inference (appendix D.5.1), biologically plausible learning (appendix D.5.2), predictive dynamics (appendix D.5.3), hierarchical extensions (appendix D.5.4), and applications to explain neural data variance, thereby bringing theory and experiment closer together (appendix D.5.5).

### D.1  A summary of our prescriptive framework, and how it relates to BLR and BONG

In this work, we started by reviewing and synthesizing existing models—from VAEs in machine learning to sparse coding and predictive coding in neuroscience—as instances of variational free energy ($\mathcal{F}$) minimization. Drawing on this unification potential, we introduced FOND (*Free energy Online Natural-gradient Dynamics*): a framework that prescribes a clear sequence of choices to derive brain-like inference algorithms and architectures from first principles.

FOND establishes a three-step guideline for developing inference models:

1. Select the distributions that fully specify the variational free energy, $\mathcal{F}$
2. Identify the dynamic variables that parameterize the approximate posterior
3. Apply online natural gradient descent on $\mathcal{F}$ to derive the dynamics

The first two steps are flexible design choices, while the third is fixed: it is an algorithmic prescription that determines the equations governing the temporal evolution of the dynamic variables. Equivalently, this also fully determines the dynamics of the posterior distribution.

To fulfill the "online" component (the O in FOND), we require that inference is performed through continual belief updates, reflecting the adaptive [45–49] and recurrent [40–44] nature of neural processing. This iterative, sequential approach departs from the one-shot amortized inference typical in VAEs [37, 38, 217]. The "natural gradient" component [34, 35, 57] (the N in FOND) ensures that optimization respects the intrinsic geometry of probability distribution space [65, 76, 200].

**Situating FOND within existing literature: FOND vs. BONG vs. BLR.** Both FOND and BONG [60] are specialized applications of the Bayesian Learning Rule (BLR; [34]) in sequential settings, but they address fundamentally different problems through distinct technical approaches. We clarify these relationships here.

**Bayesian Learning Rule (BLR, [34])** — BLR provides a general mathematical framework for deriving learning algorithms as natural gradient descent on variational objectives. It unifies diverse optimization methods—from SGD to Adam—under a single principled lens. Both FOND and BONG instantiate this framework but diverge in their goals, scope, and technical implementation.

**BONG (Bayesian Online Natural Gradient, [60])** — BONG focuses on *learning* the parameters $\theta$ of Bayesian neural networks in supervised settings. Given labeled data $(x, y)$, BONG updates network weights online to improve predictive performance. Technically, BONG restricts itself to Gaussian approximate posteriors, which enables the use of analytical results—specifically Price's [232] and Bonnet's [233] theorems—to compute gradients of expectations (their Eq. 3). This Gaussian-specific approach yields closed-form update rules (their Eqs. 9-10) that BONG then approximates in various ways to achieve computational efficiency.

**FOND (Free energy Online Natural-gradient Dynamics)** — FOND, by contrast, focuses on *deriving inference dynamics* for variational parameters (e.g., membrane potentials $u$) in unsupervised settings. FOND's goal is to specify how neural activity evolves over time to perform posterior inference, not to learn network weights.

This fundamental difference in goals necessitates a different technical approach: FOND must be distribution-agnostic to derive models like i$\mathcal{P}$-VAE (Poisson) alongside i$\mathcal{G}$-VAE (Gaussian). Because Price's and Bonnet's theorems do not apply beyond Gaussians, FOND follows the standard VAE approach: approximate expectations via Monte Carlo sampling, then compute gradients of the resulting stochastic objective. This generality allows FOND to derive the spiking dynamics of i$\mathcal{P}$-VAE, which is impossible within BONG's Gaussian-restricted framework.

> **BONG vs. FOND: Key Differences**
>
> **Goal**: BONG learns network parameters ($\theta$); FOND derives neural dynamics ($\dot{u}$)
>
> **Setting**: BONG primarily targets supervised learning (though applicable to unsupervised settings via likelihood choice); FOND targets unsupervised inference
>
> **Scope**: BONG is Gaussian-specific; FOND is distribution-agnostic
>
> **Technical approach**: BONG uses analytical gradient computation via Price's and Bonnet's theorems; FOND uses Monte Carlo sampling then differentiation

These are not merely different "flavors" of the same algorithm but distinct mathematical procedures addressing complementary problems. BONG focuses on online supervised learning with Gaussian uncertainty, while FOND derives brain-inspired inference dynamics across diverse distributional assumptions. Both enrich the BLR framework by demonstrating its versatility in different domains.

**FOND's utility: application to derive iterative VAEs.** To demonstrate the viability and utility of FOND, we developed a novel iterative VAE model family, including a spiking variant, the i$\mathcal{P}$-VAE.

For this specific implementation, we made two biologically motivated choices:

1. Modeling neural activity as spike counts [180, 192, 193, 195, 234–236] via Poisson distributions [50], departing from the standard Gaussian assumptions in predictive coding [22, 33].

2. Assigning log-rate as the dynamic variable for each neuron. Log-rate is the canonical parameter of the Poisson distribution, and it is interpretable as the membrane potential of the neuron [70, 71].

With these choices fixed, the natural gradient descent on $\mathcal{F}$ unambiguously determined the form of the spiking neural dynamics in i$\mathcal{P}$-VAE.

The resulting architecture, i$\mathcal{P}$-VAE, shares theoretical and empirical similarities with LCA [39], effectively functioning as a spiking, stochastic extension of this classic model. Our empirical results showed that all iterative VAEs (i$\mathcal{P}$-VAE, i$\mathcal{G}$-VAE, i$\mathcal{G}_{\text{relu}}$-VAE) consistently outperformed their amortized counterparts in balancing reconstruction and sparsity, with i$\mathcal{P}$-VAE and LCA achieving the best overall trade-offs. Additionally, when evaluated against hybrid iterative-amortized VAEs in out-of-distribution generalization tasks, i$\mathcal{P}$-VAE showed strong performance.

Why is i$\mathcal{P}$-VAE effective? We explore this next by drawing connections to recent advances in sequence modeling and hierarchical VAE theory.

## D.2 Three possible explanations for the effectiveness of i$\mathcal{P}$-VAE

Several complementary factors contribute to i$\mathcal{P}$-VAE's effectiveness. First, we note interesting parallels with recent advances in sequence modeling. Recently, Beck et al. [114] introduced extended LSTM (xLSTM) that performs competitively with state-of-the-art transformers and state space models. xLSTM achieves this mainly through two key modifications: (1) replacing sigmoid gates with *exponential* nonlinearities, and (2) implementing a *normalization* scheme that stabilizes training. Similarly, i$\mathcal{P}$-VAE employs the $\exp(\cdot)$ nonlinearity to map membrane potentials to firing rates, providing a highly expressive activation function (Figs. 5 and 6). Furthermore, the theoretical derivations naturally gave rise to a multiplicative divisive normalization that stabilizes network dynamics and promotes convergence (eq. (8)).

Second, all our iterative VAEs, including i$\mathcal{P}$-VAE, benefit from effective *stochastic depth*. Child [115] demonstrated both theoretically and empirically that hierarchical VAEs gain their effectiveness from stochastic depth. When we unroll the iterative VAE encoding algorithms in time (e.g., Figs. 5 and 6), they resemble specialized hierarchical VAEs, but without the spatial coarse-graining found in typical hierarchical architectures [115, 125, 161]. Based on the arguments put forth by Child [115], we speculate that this temporal depth in i$\mathcal{P}$-VAE's stochastic representations contributes significantly to model performance.

In summary, i$\mathcal{P}$-VAE's success likely stems from at least three key elements: (1) the exponential nonlinearity providing expressiveness, (2) emergent normalization ensuring stability and convergence, and (3) effective stochastic depth through temporal unrolling.

A promising direction for future work would be extending i$\mathcal{P}$-VAE into a full hierarchical architecture with spatial coarse-graining, potentially further enhancing its already strong performance, which we discuss in detail below.

## D.3 Hardware-efficient implementation and real-world deployment potential of i$\mathcal{P}$-VAE

The computational characteristics of i$\mathcal{P}$-VAE make it particularly well-suited for efficient hardware implementation, especially on specialized accelerators and resource-constrained devices. This efficiency stems from three key properties: (1) sparsity, (2) weight reuse, and (3) integer-valued representations, as detailed below.

First, i$\mathcal{P}$-VAE generates sparse, integer-valued spike counts as representations. This sparse coding strategy aligns with emerging sparse tensor accelerators and neuromorphic computing architectures, which can achieve substantial improvements in energy efficiency compared to dense matrix operations [116]. The predominance of zeros in these representations (60%-95%, in natural image patch experiments, Fig. 4; and up to 84% in MNIST experiments, Table 3) enables both memory and computation savings through sparse storage formats [102] and compute-skipping optimizations [103].

Second, i$\mathcal{P}$-VAE's inference algorithm reuses the same set of weights across all iterations (Fig. 5), potentially reducing memory bandwidth requirements compared to deep neural networks with unique parameters at each layer. This is particularly significant for hardware deployment, as memory transfers often dominate energy consumption in neural network inference [103, 237, 238]. Where traditional deep networks must continuously load new parameters from memory, i$\mathcal{P}$-VAE's weight-sharing minimizes costly memory accesses and allows for efficient on-chip caching of the entire model.

Third, the discrete, integer-valued nature of spike counts eliminates the need for high-precision floating-point arithmetic [103–105], enabling deployment on simpler fixed-point hardware with reduced power consumption. Combined with the inherent sparsity, this positions i$\mathcal{P}$-VAE as a promising candidate for edge computing applications where energy and computational resources are severely constrained.

These properties are particularly valuable for real-world autonomous systems and robotics applications, where power budgets are limited and attaching power-hungry GPUs is impractical [103]. For instance, drones, mobile robots, and wearable devices could leverage i$\mathcal{P}$-VAE for efficient on-device perception without requiring cloud connectivity or massive batteries.

As noted above, i$\mathcal{P}$-VAE and LCA share key theoretical, computational, and empirical characteristics. Given that LCA has been successfully implemented both as a spiking neural network [123] and on neuromorphic hardware [239], we expect i$\mathcal{P}$-VAE to be similarly deployable on such platforms, potentially with even greater efficiency due to its principled stochastic dynamics.

Overall, i$\mathcal{P}$-VAE's combination of sparse integer representations, weight reuse, and alignment with neuromorphic computing principles offers a promising path toward deploying sophisticated perception capabilities in resource-constrained environments, bridging the gap between theoretical neuroscience models and practical applications.

## D.4 Relationship to sampling-based inference methods

Throughout this work, we focused on variational inference (VI) as our framework for approximating Bayesian posteriors. However, sampling-based methods, particularly Markov Chain Monte Carlo (MCMC; [240–243]), represent an equally important alternative approach. Like VI, sampling-based methods have been extensively explored in theoretical neuroscience [117–122, 244–252]. See Haefner et al. [253] for a balanced review.

Both VI and sampling address a key limitation of *maximum a posteriori* (MAP) inference by providing access to the full posterior distribution rather than just a point estimate. The i$\mathcal{P}$-VAE extends LCA [39] from a deterministic point estimate to a probabilistic representation with full posterior uncertainty. Similarly, recent sampling-based approaches have enhanced both sparse coding [120] and predictive coding [121, 122] by incorporating stochastic dynamics that explore the full posterior landscape rather than converging to a single point. These parallel developments in both VI and sampling frameworks reveal their complementary roles in addressing the limitations of traditional MAP estimation.

Sampling-based models all perform Bayesian posterior inference, but differ in whether they adopt a "prescriptive" approach, as outlined in appendix B.1. For a non-prescriptive example, Echeveste et al. [247] began with a well-established biophysical circuit—the stochastic *stabilized supralinear network* (SSN; [83, 254, 255])—and adapted it to perform posterior inference. Their optimized network exhibited key biological properties and achieved substantially faster sampling than standard Langevin samplers. These results compellingly demonstrate that biologically realistic circuits can perform Bayesian inference. However, this represents a bottom-up, descriptive strategy—starting from an existing circuit and adapting it for inference; whereas, other sampling-based approaches align more closely with the top-down, principle-driven methodology we advocate.

A particularly relevant example is the work of Masset et al. [119], who demonstrated that incorporating natural gradients into Langevin sampling significantly accelerates convergence. Their approach directly parallels ours: they derive a spiking neural network algorithm prescriptively from first principles, drawing inspiration from Ma et al. [256], who proposed a "complete recipe" for constructing MCMC samplers. This "complete recipe" serves a role analogous to the Bayesian learning rule (BLR; [34]) in our work, as they both provide a unifying framework for their respective domains. While BLR unifies learning algorithms under variational inference with natural gradient updates, the "complete recipe" provides a corresponding unifying framework for sampling-based MCMC methods.

Interestingly, our iterative VAE models, while fundamentally grounded in VI principles, incorporate sampling elements during inference by approximating the reconstruction loss through Monte Carlo sampling. This hybrid character hints at an opportunity for theoretical unification between VI and MCMC. Prior work has explored this direction [257, 258], but further research is needed to clarify the relationships between VI and sampling. This unification potential is particularly interesting in the context of brain-like inference models [23, 117, 253].

## D.5    Limitations and future work

In this final appendix, we discuss the limitations of our work and how future work can address them.

### D.5.1    Iterative inference: Generalization advantages with manageable computational costs

Our experiments consistently suggest that iterative inference outperforms amortized approaches in reconstruction-sparsity trade-offs (Fig. 4). Additionally, the fully iterative i$\mathcal{P}$-VAE also outperformed hybrid iterative-amortized VAEs in out-of-distribution (OOD) generalization experiments, including both reconstruction and downstream classification OOD tasks (Figs. 13 to 16).

Furthermore, we observed a clear pattern: the less amortized the inference process, the better the generalization capability. *Semi-amortized VAE* (sa-VAE; [92]), which relies partially on iterative updates, outperformed the more heavily amortized *iterative amortized VAE* (ia-VAE; [91]), suggesting that reducing dependence on learned amortization functions enhances adaptability to novel inputs.

**Iterative reasoning in other domains.**    Similar advantages of iterative inference have been observed in other domains as well. Recent advances in large language models employ iterative reasoning at test time, either explicitly through prompting techniques like *chain-of-thought* [259] and *tree-of-thought* [260], or through more sophisticated inference-time algorithms and architectures. These approaches generally fall into two categories: some focus on the inference algorithm, like the MCMC-based sampler from Karan and Du [261], which elicits latent reasoning from a base model. Others modify the architecture itself to be inherently more iterative. For instance, the brain-inspired *Hierarchical Reasoning Model* (HRM; [262]) introduces a novel recurrent structure for latent reasoning, while Koishekenov et al. [263] train a model to recursively apply its own reasoning-critical layers.

This principle has earlier roots in computer vision, where the brain-inspired *recursive cortical network* (RCN; [264, 265]) showed that a generative model using iterative message-passing, equipped with both feedback and lateral recurrent connections, could achieve remarkable data efficiency and generalization on challenging visual reasoning tasks, such as breaking text-based CAPTCHAs [264].

These diverse approaches—spanning prompting, sampling, and architecture—all converge on a similar principle: dedicating more test-time computational steps to a problem enables superior performance on complex reasoning tasks, despite the additional overhead.

The power of iterative inference is particularly evident on benchmarks designed to test the limits of abstract reasoning. On the challenging ARC-AGI benchmark [266], Liao and Gu [267] achieved strong performance ($34.75\%$ on training set, $20\%$ on evaluation set) by replacing encoder networks entirely with expectation-maximization (EM) iterative updates. This is conceptually similar to our approach, where iterative optimization replaces learned encoders. Remarkably, Liao and Gu [267] achieved these strong results without any pretraining, further supporting the idea that iterative inference may offer inherent advantages for generalization on difficult, abstract problems.

**The serial scaling hypothesis.**    These results align with the emerging *serial scaling hypothesis* [268], which posits that many challenging problems in machine learning, particularly those involving reasoning, planning, sequential decision-making, or physical simulation, are "inherently serial." Such problems possess computational dependencies that resist parallelization, meaning that simply scaling parallel compute (e.g., model width) is insufficient.

Therefore, progress fundamentally requires scaling *serial computation*: increasing the number of sequential processing steps (e.g., recurrent iterations or model depth). This perspective suggests the benefits of iterative methods might stem not just from mimicking biological processes, but from fulfilling a fundamental computational requirement for certain task classes.

**Re-evaluating the computational trade-off.**   Our runtime analysis (appendix C.7) challenges the assumption that iterative inference is prohibitively slow at test time. Iterative models reached amortized-level performance within comparable wall-clock times (especially with large-batch parallelization), then continued refining to achieve substantially better reconstruction-sparsity trade-offs. The primary cost is training time—roughly $2\times$ longer than amortized models (appendix C.1.4)— which is understandable given their increased effective depth, interpretable as training deeper ResNets with shared parameters (Fig. 5). This suggests that iterative models are particularly well-suited for applications where inference quality and generalization are central, and the one-time cost of longer training is acceptable.

Despite these encouraging runtime results, the general efficiency of iterative methods remains an active area of research. The growing adoption of iterative techniques in large-scale commercial models has spurred optimization efforts. Recent advances in hardware acceleration [269, 270], specialized architectures for recurrent computation [114, 271], and algorithmic improvements for iterative processes [272, 273] suggest that the efficiency gap may narrow in the coming years.

The fundamental question remains whether the generalization advantages of iterative inference, increasingly supported by both empirical results and theoretical arguments, will continue to justify any remaining computational costs as both approaches evolve.

### D.5.2   Biologically plausible learning rules

In this paper, we presented a prescriptive framework for deriving inference dynamics from first principles, but we did not address this question for learning the generative model. In other words, our work is prescriptive about $\dot{u}$, but not $\dot{\theta}$.

We optimized model parameters through a procedure that can be interpreted as backpropagation through time [93, 94]. This is not biologically plausible, because it still suffers from the weight transport problem [274]. It's even worse for temporal credit assignment because neurons lack a mechanism to store activation histories needed for precise backward passes through time [275, 276].

Other models, like predictive coding (PC), are prescriptive about both inference and learning aspects. Specifically, PC learning is effectively Hebbian [33, 277]. Biologically plausible synaptic learning rules exist for sparse coding as well. For instance, Zylberberg et al. [123] introduced SAILnet, a spiking network that performs sparse coding using only synaptically local information. Interestingly, the distribution of inhibitory connections learned by SAILnet is lognormal, matching experimental observations in cortex [278].

In general, we consider inference dynamics as more fundamental than learning, for several reasons. First, inference is far more experimentally accessible than synaptic weight dynamics. We can record from large populations of neurons while animals receive sensory stimulation, but we lack complete electron microscopy reconstructions of synapses before, during, and after learning. Second, what we call "learning" in machine learning likely corresponds to a complex mixture of evolution, development, and experience-dependent plasticity in biology [143]. Given this complexity, it is reasonable to achieve learning in neural networks using powerful but biologically implausible methods such as backpropagation [279–281].

With that said, future work should still explore biologically plausible prescriptive frameworks for learning synaptic weights, potentially building on emerging theories like eligibility traces [282], feedback alignment [283, 284], and e-prop [275].

**A natural extension: combining BONG and FOND for continual learning.**   Beyond biological plausibility, an exciting direction is to combine BONG with FOND. As we discussed in appendix D.1, BONG and FOND are complementary specializations of BLR. Using the same framing adopted here, BONG can be viewed as a prescriptive framework for deriving weight dynamics, $\dot{\theta}$, while FOND prescribes inference dynamics, $\dot{u}$. Since both frameworks minimize the same free energy objective, their combination is mathematically natural.

This BONG/FOND fusion would yield a fully Bayesian, principled algorithm that is prescriptive about both learning and inference: a continually learning, online Bayesian system that adapts both its beliefs and its generative model simultaneously—potentially enabling lifelong learning without catastrophic forgetting. We consider this unified framework a promising direction for future work.

### D.5.3 Predictive dynamics, with application to non-stationary sequences

Brains are prediction machines [146, 285, 286]. Overwhelming evidence, ranging from retina [287] to cortex [288], suggests that biological circuits are engaged in predictive computations, *anticipating* the future [289], rather than merely adapting to the past. However, our current framework lacks a concrete mechanism for such anticipatory computation. While we emphasized the importance of online inference for modeling biological perception, we left a crucial element unaddressed: online inference is most meaningful in the context of non-stationary inputs. Our work, however, explored only the simplest type of sequence: stationary inputs with the same sample repeated multiple times.

Future extensions should incorporate genuinely predictive updates that *evolve* the current posterior into a predictive prior distribution, rather than relying on the identity mapping we used in this work.

Several promising approaches could guide this development. In machine learning, Duran-Martin et al. [124] introduced *Bayesian online learning in non-stationary environments* (BONE), a unifying framework for probabilistic online learning in changing environments. Within neuroscience, Jiang and Rao [290] introduced *dynamic predictive coding*, in which higher-level neurons modulate the transition dynamics of lower-level ones, which is conceptually similar to *HyperNetworks* [291]. Millidge et al. [292] introduced *temporal predictive coding* networks, where recurrent connections are used to predict the network's own future activity, allowing it to approximate a Kalman filter with biologically plausible, local learning rules. Another notable contribution is the *polar prediction model* proposed by Fiquet and Simoncelli [293]. Inspired by the temporal "straightening" hypothesis [294], the model performs linear extrapolation of phase in a learned complex-valued latent space, enabling next-frame prediction with strong performance, interpretability, and computational simplicity.

We believe predicting the future and adapting to the past represent two distinct yet complementary computations employed by biological systems. When modeling these processes, both should be clearly motivated and, ideally, derived from first principles. In this work, we addressed adaptation but left prediction largely unexplored. We aim to develop the missing predictive component in future work, while evaluating the resulting architectures on non-stationary datasets such as natural videos.

### D.5.4 Hierarchical extensions with realistic synaptic transmission delays

Another dimension of being prescriptive lies in architectural design. In this paper, we focused on single-layer models and did not provide prescriptions for multi-layer, hierarchical architectures [5, 295]. Yet this is an important direction, given growing evidence that hierarchical VAEs exhibit cortex-like organization [125, 126]. Future extensions should address two key components [296]: (1) the structure of the generative model, and (2) information flow during inference.

For the generative model, a fundamental decision concerns the probabilistic dependencies between latent variables across different layers. Should the joint distribution be structured in a Markovian chain, where each layer interacts only with its immediate neighbors? Or might a more flexible *heterarchical* structure [297–299] with an arbitrary graph topology [300] better capture the complex causal structure of natural signals?

For information flow during inference, several questions arise: what information propagates between layers, and in which directions [301]? Each choice leads to different computational trade-offs and makes distinct predictions about neural dynamics. For example, predictive coding typically propagates prediction errors upward in the hierarchy. Crucially, biological systems do not transmit raw signals between hierarchical levels, but instead apply various forms of pooling [302] or coarse-graining [125] operations that transform lower-level representations into increasingly abstract features at higher layers. This pooling serves both computational efficiency and representational purposes, enabling the extraction of invariant features while reducing dimensionality [303]. In the future, our framework could be extended to derive normative pooling operations through free energy minimization under appropriate architectural constraints, potentially explaining why the visual system employs specific receptive field pooling strategies rather than alternatives.

Finally, realistic neural models should account for finite signal propagation delays in biological axons [304–306]. In future work, we plan to incorporate such transmission delays into our framework to enhance biological realism and potentially uncover computational mechanisms uniquely enabled by temporally structured information flow [307].

### D.5.5  Applications to explain neural data variance

The ultimate test of a model is its ability to explain empirical data. i$\mathcal{P}$-VAE demonstrated several emergent behaviors that mimic cortical properties at a phenomenological level (Figs. 3, 8 and 11), but this is only a first step toward rigorous validation against neural data. A comprehensive assessment would require testing our model's capacity to explain neural response variance [308, 309], for example, in frameworks like Brain-Score [127].

Our philosophical approach differs from purely descriptive models that are directly optimized to fit neural responses [128, 310]. Instead, we advocate for normative models that are *validated* against neural data, rather than trained explicitly to *reproduce* it, in line with critiques of purely descriptive approaches [311]. This approach offers greater interpretability, as the model's behavior stems from principled objectives (e.g., free energy minimization) rather than parameter fitting [312]. From this point of view, there are two promising paths forward.

First, we can systematically explore different prescriptive choices within our framework to identify which configurations best explain neural data. For instance, comparing how alternative inference dynamics or distributional assumptions affect the model's ability to predict neural responses to novel stimuli. Second, we might consider a semi-supervised training approach [313], where free energy minimization is augmented with a neural regularization term that encourages alignment between i$\mathcal{P}$-VAE spike patterns and recorded neural activity in response to identical stimuli.

The spiking nature of i$\mathcal{P}$-VAE makes it particularly appropriate for comparison with neural recordings, offering closer parallels to biological data than non-spiking alternatives. Beyond explaining variance, i$\mathcal{P}$-VAE's spiking representation also enables in silico experiments that can potentially inform future in vivo studies [128, 314]. Finally, this compatibility with spike train data creates an opportunity for rigorous comparisons between variational inference approaches like ours and sampling-based spiking models [118, 119, 245, 247, 249–252, 305] (appendix D.4), in terms of their ability to account for empirical neural responses.

In the future, we are excited to test a fully predictive, hierarchical version of i$\mathcal{P}$-VAE against neural data to put our framework to the ultimate test. Such experiments could validate the theory and help identify which aspects of neural computation are most effectively captured by variational principles.

