# OpenReview forum: "Brain-like Variational Inference"
_NeurIPS.cc/2025/Conference — NeurIPS 2025 poster_

### Official Review · Reviewer_fucC · 2025-06-25

**Clarity:** 4
**Significance:** 3
**Originality:** 3
**Rating:** 5
**Confidence:** 4

**Summary:**

The paper proposes a unifying framework, Free energy Online Natural-gradient Dynamics (FOND), for deriving brain-like inference dynamics by minimizing variational free energy (F). It introduces the iterative Poisson variational autoencoder (iP-VAE), a spiking neural network whose membrane potential dynamics perform variational inference via natural gradient descent. The work claims both theoretical and empirical advances, including emergent normalization, hardware efficiency, and improved reconstruction-sparsity trade-offs compared to standard VAEs and predictive coding models.

**Questions:**

I liked that the authors (obviously) compared against the P-VAE, and showed superior performance. Do you believe the good results are a consequence of the iterative process, or is there something specific about the Poisson distribution? More precisely, do you believe that in general iterative methods should perform better than amortized ones, and that the gap in performance you observed in simply because of that?

**Ethical Concerns:**

["NO or VERY MINOR ethics concerns only"]

**Limitations:**

Yes

**Quality:**

4

**Strengths And Weaknesses:**

Pros:

1) he paper provides a clear and compelling synthesis of variational inference in machine learning (ELBO maximization) and theoretical neuroscience (free energy minimization), highlighting their mathematical equivalence and bridging the two fields. It was also a very interesting read as it connected this work well with the recent literature on BLR and Poisson VAEs.
2) The derivation from first principles is interesting.


Cons:

1) While the theoretical contributions are strong, the empirical results are somewhat limited in scope. The main experiments focus on small-scale datasets (natural image patches, MNIST). What is the limit to which this algorithm can be pushed? FashionMNIST? Colored images like CIFAR10? It would be nice to add a small discussion on the problem of scale.
2) While the unification is well-argued, the core idea of iterative inference via natural gradients is closely related to recent work (e.g., Bayesian Learning Rule, BONG), and the paper could more clearly delineate what is fundamentally novel in FOND and iP-VAE versus extensions of these frameworks.

---

> ### Author Rebuttal · Authors · 2025-07-30
>
> We sincerely thank Reviewer fucC for their encouraging evaluation and constructive feedback. We are glad they found our work to be a "clear and compelling synthesis" that connects well with recent literature, as this was a primary goal of our manuscript.
>
> While the reviewer acknowledges our theoretical contributions, they rightly point to the limited scope of our initial experiments and ask for clarification on novelty. Below, we address these important points with new experimental results and a clearer framing of our contributions.
>
> ---
> ---
> ---
>
> > While the theoretical contributions are strong, the empirical results are somewhat limited in scope. The main experiments focus on small-scale datasets (natural image patches, MNIST). What is the limit to which this algorithm can be pushed?
>
> We thank the reviewer for raising this concern, which was shared by several other reviewers. We agree that addressing this limitation will substantially improve the final paper. To this end, we performed experiments on three color-image datasets: ```CIFAR-10``` (3 x 32 x 32), ```tiny ImageNet``` (3 x 32 x 32), and cropped ```CelebA``` (3 x 128 x 128).
>
> Due to time constraints during the rebuttal period, we performed only limited hyperparameter search, and evaluated the models in their sparsity-reconstruction performance. We plan to conduct more extensive experiments, such as classification performance using a linear probe,  and include these as supplemental results in the final paper. Below, we report results from this preliminary exploration.
>
> # Scaling up iP-VAE: experiments on more complex, colored image datasets
>
> Here, we demonstrate how iP-VAE can be readily scaled up to more complex datasets, such as faces and natural images.
>
> ## ```CelebA``` (cropped to 128 x 128)
>
> We trained the following models:
>
> 1. iP-VAE with an iterative encoding algorithm and a deep decoder with 8 convolutional layers (interpreted as "deep sparse coding," see Figure 6 and Appendix B.6).
> 2. iP-VAE with an iterative encoding algorithm and a single layer convolutional decoder.
> 3. P-VAE with a deep convolutional encoder and decoder. This is the amortized counterpart of model # 1.
> 4. P-VAE with a deep convolutional encoder and single layer convolutional decoder. This is the amortized counterpart of model # 2
>
> All models shared a latent dimensionality of 256 * 16 * 16, and the linear decoders (models # 2 and 4) had a kernel size of 38. We trained all models using $\beta = 1$, and evaluated them in their sparsity-reconstruction performance. We also report overall performance, defined as the Euclidean distance from the ideal point (golden star; $R^2$ = 1, Sparsity = 1), where a lower score is better. Here are the results:
>
> |   | model | encoder | decoder | $R^2$ ↑ | Portion zeros ↑ | Overall performance ↓ | # params ↓ |
> |---|-------|---------|---------|-----|---------------|----------|----------|
> | 1 | iP-VAE | iterative | conv (deep) | 0.95 | 0.93 |  0.09 | 0.7 M |
> | 2 | iP-VAE | iterative | conv (lin) | 0.90 | 0.91 |  0.13 | 1.1 M |
> | 3 | P-VAE | conv | conv (deep) | 0.86 | 0.96 | 0.15 | 1.9 M |
> | 4 | P-VAE | conv | conv (lin) | 0.74 | 0.99 |  0.26 | 2.3 M |
>
>
> ## ```CIFAR-10``` and ```tiny ImageNet``` (32 x 32)
>
> For these datasets, we trained model #1:  iP-VAE with a deep decoder but now with 6 layers. The latent dimensionality was 256 * 8 * 8. We trained all models using $\beta = 1$. Results:
>
> | Dataset | model | $R^2$ ↑ | Portion zeros ↑ | Overall performance ↓ |
> |---------|-------|-----|---------------|---------------------|
> | ```CIFAR-10``` | iP-VAE (deep conv decoder) | 0.93 | 0.97 | 0.08 |
> | ```tiny ImageNet``` | iP-VAE (deep conv decoder) | 0.95 | 0.94 | 0.08 |
>
> In conclusion, these new results demonstrate that iP-VAE scales effectively to complex, high-dimensional color image datasets, achieving a superior reconstruction-sparsity trade-off compared to its amortized counterpart, all while using fewer parameters. We believe this substantially strengthens the empirical support for our claims and addresses the concerns about the limited scope of our initial experiments.
>
> ---
> ---
> ---
>
> > While the unification is well-argued, the core idea of iterative inference via natural gradients is closely related to recent work (e.g., Bayesian Learning Rule, BONG), and the paper could more clearly delineate what is fundamentally novel in FOND and iP-VAE versus extensions of these frameworks.
>
> We thank the reviewer for this crucial point, which was shared by others. The reviewer correctly points out that our FOND framework shares its core mathematical machinery with BONG and the broader Bayesian Learning Rule (BLR) framework. Our primary contribution is not a new mathematical rule, but a new **recipe** for how this machinery can be used to derive and interpret **brain-like dynamical systems** from first principles.
>
> To address this, we will add a new section to the revised manuscript dedicated to clarifying this relationship. Below is a sketch of our planned explanation.
>
> # The relationship between BLR, BONG, and FOND
>
> We see a clear lineage connecting these frameworks:
>
> - BLR provides the general principle for deriving learning rules from variational updates,
> - BONG applies BLR to sequential online inference with a single update per observation.
>     - The authors of BONG specifically state that: "BONG can be seen as a special case of BLR with one update step per observation and learning rate 1".
> - FOND applies BLR to derive the *governing dynamics* of a neural system.
>
> The novelty of FOND lies in its explicit emphasis on **dynamics** (the "D"). While BONG and BLR provide a recipe for optimizing variational parameters, FOND provides a recipe for deriving the governing dynamics equations of a neural system.
>
> As we outline in Appendix B.2, this is inspired by prescriptive theories in physics, where one derives dynamics by (1) first identifying the **dynamical variables** (e.g., membrane potentials $\mathbf{u}$) and then (2) defining the principles that govern their evolution (i.e., **natural gradient descent on free energy**).
>
> This prescriptive lens is what allows us to start from an abstract objective ($\mathcal{F}$) and arrive at a concrete, interpretable neural circuit model, such as a spiking network with lateral competition (iP-VAE), or other familiar models. This focus on deriving circuit dynamics is the key differentiator. The goal of BLR/BONG is primarily to perform efficient Bayesian inference. The goal of FOND is to ask:
>
> *"If a neural system were performing inference by minimizing free energy, what would its dynamics look like?"*
>
> The result is a canonical model that is both biologically plausible and empirically effective, a connection not emphasized by prior work.
>
> We will clarify this distinction in the revised manuscript. We also plan to add a Venn diagram, illustrating that BLR is the general mathematical framework, containing both BONG and FOND as two distinct, specialized applications. BONG focuses on sequential inference, while FOND focuses on deriving biologically plausible neural dynamics.
>
> Thank you for pushing us to clarify this relationship. We believe it will significantly strengthen the paper’s framing.
>
> ---
> ---
> ---
>
> > Do you believe the good results are a consequence of the iterative process, or is there something specific about the Poisson distribution? More precisely, do you believe that in general iterative methods should perform better than amortized ones, and that the gap in performance you observed in simply because of that?
>
> We thank the reviewer for this great question. It gets to a key point of our paper.
>
> Yes, we believe the better results come from the iterative process in general, not just from using the Poisson distribution.
>
> The results in Figure 3b show that all of the iterative models perform better than their amortized counterparts. This is true for all the models we tested.
>
> We think this shows an important trade-off that is worth more attention. Our work provides clear evidence that iterative inference, while slower, can close the known "amortization gap" and produce better results. This leads to an interesting question for both AI and neuroscience: how does the brain handle this? The brain's inference needs to be both *accurate* (like iterative methods), and *fast* (like amortized ones). This suggests that a *hybrid* approach might be at play, which is an exciting direction for future work.

---

> > ### Comment · Reviewer_fucC · 2025-08-04
> > **Acknowledgement**
> >
> > I thank the authors for the detailed answer, and for the nice paper!

---

### Official Review · Reviewer_yjVk · 2025-06-29

**Clarity:** 4
**Significance:** 2
**Originality:** 2
**Rating:** 4
**Confidence:** 4

**Summary:**

The authors modify the (recent) Poisson variational autoencoder (P-VAE) to use iterative, rather than amortized, inference (iP-VAE).  That is, they compute the natural parameters of the Poisson recognition model with gradient descent, rather than a forward pass through a deep neural network.  The authors show that, at least in certain settings, it achieves a better trade-off between reconstruction error and sparsity than models using amortized inference.  It also learns receptive (really, projective) fields that resemble the receptive fields of V1 neurons, as in classic sparse-coding models.  Much of manuscript is devoted to situating the procedure in the context of free-energy minimization (or ELBO maximization).

**Questions:**

- The receptive fields for the P-VAE in Fig. 16 are much less Gabor-like than those in the original paper (ref. [50], Fig. 4).  Can the authors explain this discrepancy?

 - The disappearance of the recognition-model entropy from the inference procedure is justified slightly differently in the BONG paper.  Can the authors explain the relationship between the justifications?  Are they two ways of saying the same thing?

 - Training vs. inference:
	 - Why use the "Poisson reparameterization algorithm" during training but the STE during inference?
	 - If I have understood correctly, the free-energy gradient in Eq. 6 is used in training, even though the beta-dependent term is effectively dropped during inference.  Is that right?

 - One of the traditional complaints about iterative inference is that it takes too long.  Can the authors say anything about this?  Can they relate it to biologically relevant time scales?

- It appears that, on the basic metric of Fig. 3 (or Fig. 9), the iP-VAE performs about the same as LCA.  Can the authors comment on what they see as the advantage of the iP-VAE?

- I don't understand the authors' explanation for the amortization gap in the appendix (near line 1376).

**Ethical Concerns:**

["NO or VERY MINOR ethics concerns only"]

**Final Justification:**

In short, I am not persuaded that the difference in emphasis between FOND and BONG (e.g., laid out in the response to Reviewer fucC) is sufficiently significant to justify publication here--even though I agree with the authors on the importance of a prescriptive framework for learning.

The authors have laid out a good plan to address many of the other shortcomings of the MS, but too much is promissory (this year's proscription on uploading a one-page pdf is unfortunate). The MS would really benefit from these changes and another round of review.

**Limitations:**

See "Strengths and Weaknesses" above.

**Quality:**

3

**Strengths And Weaknesses:**

The core claims appear sound, the manuscript is very clearly written, and this variation of the Poisson VAE is convincingly more biologically plausible than the original.  The contrast-dependent response latency is compelling (perhaps could have been promoted to the main text) and yet intuitive under the model.


Weaknesses:
My main concern is that the contribution is not substantial enough:

(1) The algorithm is more or less (just) an application of BONG [ref. 60]/the Bayesian learning rule [34] to the existing Poisson VAE.

(2) This is perhaps obscured because much---really, too much---of the MS is given over to preliminaries and didactic material.  (This reviewer shares the authors' enthusiam for the free energy as a unifying principle, but it crowds out the actual contributions of the MS.)  The new material doesn't really begin till page 5, and then there are only two figures of results in the main text.

(3) As the authors know, the datasets are primitive by modern standards.  This is particularly problematic because amortized inference is supposed to be most useful in the setting of large, complex datasets, which makes the comparisons arguably unfair.  Generation is not assayed at all.  The quality of the latent variables is not quite as good as the P-VAE (Table 2), although MNIST is not a very discriminative dataset.

(Along these lines: the models employing amortized inference use deep neural networks and > 10x as many parameters as the iterative-inference networks [Table 2].  Is this really a good choice when the encoder is linear?)


Minor:
"In machine learning, diffusion models were originally motivated by non-equilibrium statistical mechanics [29], only later understood as a form of ELBO maximization."  Altough the work cited [29] emphasizes Jarzynski's inequality, it also (Sec. 1.2) makes explicit the connection to ELBO maximization, including VAEs, and invokes Hinton's original Science paper on free-energy minimization.

---

> ### Author Rebuttal · Authors · 2025-07-31
>
> We sincerely thank Reviewer yjVk for their constructive comments and encouraging feedback. We appreciate that they found our core claims to be sound, the manuscript to be "very clearly written," and iP-VAE to be "convincingly more biologically plausible than the original."
>
> We address the comments and questions below.
>
> ---
> ---
> ---
>
> > The contrast-dependent response latency is compelling (perhaps could have been promoted to the main text)
>
> We share the reviewer's enthusiasm, as this empirical result connects nicely to the emergent normalization in the theory (eq. 9), and the model exhibits this interesting dynamics even though it was trained only on static images. We will consider promoting this figure to the main text for the final version.
>
> ---
> ---
> ---
>
> > (1) The algorithm is more or less (just) an application of BONG [ref. 60]/the Bayesian learning rule [34] to the existing Poisson VAE.
>
> This is a crucial point, which was shared by others. The reviewer correctly points out that our FOND framework shares its core mathematical machinery with BONG and the broader Bayesian Learning Rule (BLR) framework. Our primary contribution is not a new mathematical rule, but a new **recipe** for how this machinery can be used to derive and interpret **brain-like dynamical systems** from first principles.
>
> To address this, we plan to add a new section to the revised manuscript, dedicated to clarifying this relationship. We provide a sketch of this planned section in our rebuttal to Reviewer ```fucC```. (We apologize for redirecting your attention to the other rebuttal - this should have been a global rebuttal response, but those are not allowed this year.)
>
> ---
> ---
> ---
>
> > (2) …too much of the MS is given over to preliminaries and didactic material … The new material doesn't really begin till page 5 …
>
> We agree that the current balance can be improved. Our goal was to provide a synthesis connecting ML and neuroscience concepts, but we see that it delayed the introduction of our novel contributions. In the revised version, we will streamline the background, move more didactic material to the appendix, and present our FOND framework and the empirical results much earlier in the paper.
>
> ---
> ---
> ---
>
> > (3) As the authors know, the datasets are primitive by modern standards. This is particularly problematic because amortized inference is supposed to be most useful in the setting of large, complex datasets, which makes the comparisons arguably unfair.
>
> We thank the reviewer for raising this concern, which was shared by several other reviewers. We believe that addressing this limitation will substantially improve the final paper. To this end, we performed experiments on three color-image datasets: ```CIFAR-10``` (3 x 32 x 32), ```tiny ImageNet``` (3 x 32 x 32), and cropped ```CelebA``` (3 x 128 x 128).
>
> We report these results in our rebuttal to Reviewer ```fucC```. (Once again, apologies for redirecting your attention to the other rebuttal - this should have been a global rebuttal response, but those are not allowed this year.)
>
> ---
> ---
> ---
>
> > Generation is not assayed at all.
>
> This was deliberate because the main focus of the paper and FOND is brain-like **inference** dynamics, where generation serves as a means to this end (i.e., *"analysis-by-synthesis"*). However, we take your point that assessing the model's generative capabilities is informative. To this end, we performed a qualitative evaluation of a <grad|conv> iP-VAE trained on MNIST ($T_\text{train} = 16$). Our generation algorithm works as follows:
>
> - Draw samples from the learned prior
> - Pass them through the trained decoder, generate an image
> - Treat the generated image as an input sample for the next time point: perform inference on it, update the posterior
> - Draw samples from the current posterior, pass through decoder, generated the next image
> - Repeat
>
> Since we cannot include images in the rebuttal this year, we offer a verbal description of the results:
>
> The model successfully generated coherent and recognizable digits. We observed an interesting dynamic during the iterative generation process: some initial samples appeared ambiguous (e.g., a shape resembling both a '3' and a '6'), but would resolve into a single, well-defined digit after a few more iterations. This suggests the generative process mirrors the kind of refinement and convergence we see during inference. However, we also identified a class imbalance in the generated samples; for example, the digit '8' appeared more frequently than its ~10% representation in the training set. This could be addressed in future work using more expressive hierarchical priors.
>
> Overall, these promising results confirm the model can generate realistic samples through an iterative procedure (at least on MNIST), and that its dynamics are interesting in their own right. We will include this as a supplemental fig, but we respectfully maintain that the core contribution of this paper is the derivation and analysis of the inference algorithm itself.
>
> ---
> ---
> ---
>
> > The quality of the latent variables is not quite as good as the P-VAE (Table 2), although MNIST is not a very discriminative dataset.
>
> The reviewer is correct that in our MNIST results (Table 2), the amortized P-VAE, with its deep convolutional encoder, achieves slightly higher downstream classification accuracy than the iterative iP-VAE.
>
> We believe this demonstrates a classic trade-off: the powerful amortized encoder is highly effective at learning the specific features of the MNIST training set, but this comes at the cost of out-of-distribution (OOD) generalization. Our OOD experiments in the appendix support this conclusion. While P-VAE is competitive on the standard MNIST test set, its classification accuracy is lower than iP-VAE's on rotated MNIST digits (Figure 13), and drops substantially when transferring to the EMNIST dataset (Figure 14).
>
> To provide a definitive answer on more complex datasets, we will perform the same downstream classification analysis on our newly trained ```CIFAR-10``` and ```tiny ImageNet``` models for the final manuscript. This will clarify the true quality of the latent variables on more challenging benchmarks.
>
> ---
> ---
> ---
>
> > (Along these lines ... Is this really a good choice when the encoder is linear?)
>
> This is an excellent point, which we addressed with new controlled experiments on ```CelebA```. We tested iterative (iP-VAE) and amortized (P-VAE) models with both deep (8 convolutional layers) and shallow (single layer) decoders. **In all matched comparisons, iP-VAE outperformed P-VAE**, confirming its advantage is **robust to decoder complexity** and not an architectural artifact. We will add a full analysis on all new datasets to the final paper.
>
> ---
> ---
> ---
>
> > The receptive fields … are much less Gabor-like than those in the original paper ... Can the authors explain this discrepancy?
>
> One possible explanation is a **different choice of hyperparameters**. For P-VAE, the reported receptive fields were obtained from a model trained using the standard $\beta = 1$; however, our iP-VAE results correspond to $T_\{train} = 16$ and $\beta = 24$. It’s possible that a different combination of these hyperparameters would make the receptive fields more or less Gabor-like.
>
> > The disappearance of the recognition-model entropy from the inference procedure is justified slightly differently in the BONG paper ... Are they two ways of saying the same thing?
>
> As we show in Appendix B.10, this is an inevitable consequence of performing a single update per time point. BONG frames it as a prescription, but it follows directly from the online, single-update setup. Ultimately, these are **different descriptions of the same mathematical result.**
>
> > Why use the "Poisson reparameterization algorithm" during training but the STE during inference?
>
> The two methods serve different goals. We use STE to derive the inference update rule itself. We use the Poisson reparameterization algorithm for the separate task of training the decoder's weights via backpropagation through time.
>
> > If I have understood correctly, the free-energy gradient in Eq. 6 is used in training, even though the beta-dependent term is effectively dropped during inference. Is that right?
>
> That is correct. The beta term is used for training the decoder. We also experimented with multi-step inference where the term would be active, but saw no significant performance difference, so we adopted the simpler single-step BONG prescription.
>
> > One of the traditional complaints about iterative inference is that it takes too long. Can the authors say anything about this? Can they relate it to biologically relevant time scales?
>
> We agree and address runtime quantitatively in our response to Reviewer ```iEYy```. We also believe **slowness can be a feature, scaling with task difficulty.** For instance, the macaque visual cortex takes ~30 ms longer to recognize “challenge” images ([Kar et al., 2019](https://www.nature.com/articles/s41593-019-0392-5)).
>
> An iterative framework has the potential to capture this time-dependent difficulty, a connection we plan to explore in future work.
>
> > It appears that, on the basic metric of Fig. 3 (or Fig. 9), the iP-VAE performs about the same as LCA. Can the authors comment on what they see as the advantage of the iP-VAE?
>
> While empirically similar, iP-VAE has key advantages:
>
> 1. It is a **stochastic** probabilistic model, while LCA is deterministic.
> 2. It uses **compressed, integer spike count representations**, unlike LCA's continuous values.
> 3. It naturally extends to **deep decoders** (Figure 6, Appendix B.6).
>
> The fact that iP-VAE performs on par with LCA despite these more constrained, biologically plausible representations is a strength.
>
> > I don't understand the authors' explanation for the amortization gap in the appendix (near line 1376).
>
> Thank you for pointing this out. Our current explanation is cryptic. We will revise that section to enhance clarity.

---

> > ### Comment · Reviewer_yjVk · 2025-08-08
> >
> > (I apologize for the late reply.)
> >
> > I appreciate the thorough responses.  Still, my reservations on originality and significance remain, and I retain my score: I am not persuaded that the difference in emphasis between FOND and BONG (e.g., laid out in the response to Reviewer fucC) is sufficiently important (even though I agree with the authors on the importance of a prescriptive framework).
> >
> > The authors have laid out a good plan to address many of the other shortcomings of the MS, but too much is promissory (this year's proscription on uploading a one-page pdf is unfortunate).  The MS would really benefit from these changes and another round of review.

---

### Official Review · Reviewer_UGdd · 2025-06-29

**Clarity:** 3
**Significance:** 2
**Originality:** 3
**Rating:** 3
**Confidence:** 5

**Summary:**

The authors introduce a family of iterative variational autoencoders (VAEs) inspired by Bayesian theories of probabilistic inference in the brain and biological plausibility. Their broader goal is to propose a prescriptive framework called Free energy Online Natural-gradient Dynamics (FOND) for specifying and learning latent variable models based on variational inference that model the brain’s perceptual inference processes.

Their main model — the iterative Poisson VAE (iP-VAE) — uses a prior and posterior to model the latent variable hierarchically, mimicking neural activity with components such as continuous membrane potential and firing rate that includes temporal dynamics, and a Gaussian likelihood to model how the latents generate observed images. They also include versions with Gaussian priors and posteriors with and without an additional nonlinearity.

All their models are trained using natural gradient descent to maximize the variational lower bound (ELBO) of images.

They evaluate performance on the van Heteren and MNIST datasets showing improvements in reconstruction error, learned latent sparsity, out-of-distribution generalization compared to existing amortized VAEs.

**Questions:**

1. Model details: The authors use $x$ to denote observations and $z$ for latent variables. They introduce "dynamic variables" $u$ and $r$ (Lines 167, 168) and use them in subsequent equations (including Eq 3), but they do not explicitly state the mathematical relationships between $x$, $z$, $u$, and $r$. The manuscript would benefit from clearly defining these relationships upfront in the main text, preferably accompanied by a graphical model, before proceeding with their usage. The main text would also benefit from stating the exact forms of the posterior and prior (just the way Gaussian-linear likelihood is stated, but the prior and posterior are pushed to the appendix). Given that the models being proposed are essentially VAEs with specific choices of prior, posterior and likelihood, stating the above quantities upfront makes the necessary equations (ELBO, gradients) follow much more naturally. Also from the appendix, I do not understand the assumed posterior well. It is stated as $Pois(z; r)$, but shouldn’t it be conditioned on $x$? I would like to see the function that relates $x$ to $z$ in the posterior. Including these details upfront would greatly improve the flow of the paper and clarity.
2. Line 335: what are PCNs? I do not think this is introduced in the main text.
3. Re Figure 3, panel b: is this a reasonable metric? The gold-star — although I understand at first glance — is not only impossible but also ill-defined (no latent variables are ever active?).
4. Re evaluation: a more rigorous metric than reconstruction error would be to compare the log-likelihood (ELBO) of images under the given models. It would be informative to see that benchmarked against baselines. Mere reconstruction error only gives us half the picture.

**Ethical Concerns:**

["NO or VERY MINOR ethics concerns only"]

**Final Justification:**

Among the raised issues, the authors have (1) addressed scaling to modern datasets and shown that their results hold, and (2) substantially clarified the motivation behind FOND, agreeing to reframe it as a promising brain-inspired approach rather than a definitive prescription.

However, the central issue remains. While FOND is well-motivated from reasonable first principles, the work does not provide rigorous enough evaluation to substantiate the claim that its derived models are truly “brain-like”. The only metrics reported are image reconstruction error and latent sparsity, compared to simpler versions of the generative models. These evaluations do not convincingly evaluate brain-like properties or biological plausibility. Evaluation on neuroscience datasets, showing that FOND-derived models reproduce or predict neural phenomena concretely is warranted.

While the approach has promise, this has yet to be demonstrated conclusively. I therefore maintain my score and encourage resubmission with broader and more targeted neural evaluations, alongside the updated positioning and careful reframing of FOND in light of the rebuttal discussions.

**Limitations:**

The authors mention their limitations bundled with future work in a single line: "[L356] Future work should explore acceleration techniques for iterative inference [106], biologically plausible learning rules [115], predictive dynamics for non-stationary sequences [116], hierarchical extensions [117, 118], and neural data applications [119, 120]."

This is too brief and doesn't sincerely inform the reader critically of the limitations--of either FOND or the developed VAEs--such as strong assumptions made, scope and computational efficiency. I understand there exists a detailed appendix, but I would strongly encourage the authors to substantiate it in the main text.

I would particularly encourage the authors to substantiate limitations of natural gradient descent and iterative updates as opposed to amortized (these two aspects focusing on FOND), and why these are "prescriptions" for more "brain-like" architecture; the choices of prior, posterior and likelihood of their proposed models; and perhaps most critically, reflect on their choice of evaluation (please see my point on **Impact** under weaknesses).

**Quality:**

2

**Strengths And Weaknesses:**

## Strengths
- The paper boasts a **strong technical foundation**, building on variational methods in modern probabilistic machine learning and the free energy principle (FEP) of theoretical neuroscience. It also aims to connect several approaches, traditionally siloed, of theoretical neuroscience such as FEP, predictive coding, sparse coding and potentially neural sampling and reviews diverse set of works well.
- The specification of models (choices of variables, learning algorithms and derivations thereof) are **principled** from a probabilistic machine learning perspective, and their core idea of biologically motivating the structure of the latent space is conceptually appealing.
- The paper is **well-written**.

## Weaknesses

- **Unclear significance of FOND**: I’m struggling to see substantial novelty or contribution in the proposed prescriptive framework, FOND. As presented, FOND primarily prescribes latent variables that are inspired by the brain’s neural activity—such as time varying firing rates and membrane potentials—and employs natural gradient descent to optimize the ELBO. From a machine learning or statistical standpoint, this appears to be a standard application of variational inference tailored to a specific setting or just latent variable modeling in general. It’s unclear whether this warrants being framed as a new “prescriptive framework.” In my view, the work would be better positioned as introducing a novel, brain-inspired family of VAEs, rather than proposing a new framework. Additionally, it's not clear what exactly is being prescribed and why: the latent variables and their distributions? Online updates? Natural gradient descent? Perhaps most crucially, FOND seems to not prescribe how to evaluate the supposed brain-like models. I would appreciate clarification from the authors regarding their intended contribution and how FOND offers distinctive value. I’m open to correction if I have misunderstood or missed a key aspect.
- **Unclear novelty of proposed models**: While the models are well-motivated and principled, they come across as incremental extensions with only modestly novel components. Beyond their iterative inference structure, it’s unclear how these models substantially differ from prior work—for example, the Poisson-VAE by Vafaii et al. (2024). The proposed models may offer added expressiveness or biological plausibility (e.g., resembling spike counts and membrane potential to some extent), but the manuscript would benefit from a clearer articulation of what is substantively new and how these aspects advance existing approaches.
- **Limited impact of proposed models and insufficiently compelling evaluation**: the main text primarily focuses on evaluating the models on image reconstruction error and sparsity (figure 2 and 3) on MNIST and van Hateren and compare the models to amortized baselines. Are these (1) strong enough evaluation criteria and (2) strong enough baselines? Given that the main contribution centers on creating models with brain-like dynamics and biological plausibility, evaluations on real neuroscience data are strongly warranted. In fact, I would have expected their purported framework to include this as a "prescription". In that case, the corresponding baselines should also include competitive neural predictive models, at least to note the gaps. Alternatively, if the models are intended as normative, what neural phenomena do they explain that remain difficult for existing models to capture, and how are those insights attributable to the prescriptions of FOND or specific choices of the generative model (beyond commonly observed effects such as the emergence of Gabor-like features from the Gaussian likelihood). Setting aside biological relevance, it's unclear why a rigorous metric such as log-likelihood (ELBO) of images is also left out. Seeing it as an image generative model, it's unclear how it performs compared to state of the art models such as diffusion models. I see this overall as the primary weakness of the work.
- **Limited choice of datasets**: Given that the proposed models build on modern probabilistic techniques, not having evaluations on more challenging and larger benchmarks such as CIFAR or ImageNet raises concerns over computational efficiency and scalability.

## Overall assessment

Edit: note that this assessment was written prior to the rebuttal phase. Please see below for final justification.

This paper presents a principled, biologically motivated VAE family, but its impact is limited by the unclear significance and originality of their proposed "FOND" framework, modest model novelty, and most crucially, insufficient empirical evaluation to support the paper's claims.

**To raise my scores**, I encourage authors (where feasible) to **(1)** more clearly articulate the originality and distinctive value of the FOND framework and how their developed VAE models substantially advance prior works, **(2)** expand it to include tasks grounded in real neural data to support biological and experimental relevance, **(3)** train the VAEs on more modern datasets, and evaluating using rigorous metrics such as log-likelihood (ELBO), or provide a clear justification for their current choices, **(4)** improve the clarity in terms of defining components of the generative models upfront, and lastly **(5)** discuss limitations.

---

> ### Author Rebuttal · Authors · 2025-07-31
>
> We sincerely thank Reviewer UGdd for their constructive, thorough, and well-thought-out feedback, along with their clear directions to improve. We appreciate that they pointed out the paper's strengths, including its "strong technical foundation," principled and "conceptually appealing" model design, and that it is "well written."
>
> ---
> ---
> ---
>
> # Scope clarification
>
> >  "In my view, the work would be better positioned as introducing a novel, brain-inspired family of VAEs, rather than proposing a new framework."
>
> We agree that two distinct contributions are in play:
>
> - **(i)** Introducing a prescriptive framework (FOND) for deriving brain-like inference dynamics from first principles, and
> - **(ii)** Applying FOND to derive a family of concrete, brain-inspired VAEs.
>
> We also share the reviewer’s enthusiasm for evaluating the brain-inspired models on real neural data to demonstrate biological relevance. However, the primary goal of this manuscript is to establish **FOND**, a rigorous theoretical framework for deriving such biologically interpretable neural inference dynamics from first principles.
>
> ---
> ---
> ---
>
> We appreciate that the reviewer provided us with a clear roadmap with 5 concrete action items for improvement. We believe we can address four of them quite well, but the point about neural data will have to wait for a future submission.
>
> ## Item (1): Originality of FOND
>
> FOND is not a new mathematical rule, but a **recipe** for how to derive and interpret **brain-like dynamical systems** from first principles. To clarify the utility of such a framework, we plan to add a new section to the revised manuscript.
>
> We provide a sketch of this planned section in our rebuttal to Reviewer ```fucC```. (We apologize for redirecting your attention to the other rebuttal - this should have been a global rebuttal response, but those are not allowed this year.)
>
> > it's not clear what exactly is being prescribed and why
>
> FOND simply makes explicit what choices you have when building a model based on Free Energy ($\mathcal{F}$) minimization. You must choose your likelihood and approximate posterior, and how to parameterize it. From that, the **dynamics** of the variational parameters are fully determined as natural gradient descent on $\mathcal{F}$. We show how different models result from different choices, and that new models can be derived. We are actively using this framework to make more capable brain-inspired VAEs that go beyond iP-VAE.
>
> ## Item (2): Applications to real neural data
>
> > … support biological and experimental relevance … evaluations on real neuroscience data … what neural phenomena do they explain that remain difficult for existing models to capture
>
> In the current work, we tested contrast-dependent response latency and found that iP-VAE exhibits cortex-like dynamics (Appendix C.3, Figure 8). But what about other neural phenomena?
>
> [Zhu & Rozell (2013)](https://doi.org/10.1371/journal.pcbi.1003191) demonstrated that LCA sparse coding exhibits a variety of neural phenomena, such as end-stopping, surround suppression, cross-orientation suppression, and so on. Given the strong theoretical and empirical similarities between LCA and iP-VAE, we believe iP-VAE would likely reproduce most of the neural phenomena exhibited by LCA. We will include additional evaluations of phenomenology in the final manuscript. The key advantage is that iP-VAE achieves this with stochastic, spiking neurons, whereas LCA is deterministic and continuous.
>
> Moreover, we are currently pursuing direct comparisons to neural data, such as anchoring model latents with real neural data during training, or finding correlations between real neurons and extensions of iP-VAE neurons (similar to [Vafaii et al., 2023](https://openreview.net/forum?id=1wOkHN9JK8)). We do outline this direction in Appendix D.5.5 (Applications to explain neural data variance). While this is an exciting next step, it is beyond the scope of our current contribution.
>
> Overall, the present work establishes FOND, which offers a clear recipe to derive brain-like inference algorithms that go beyond this, including architectures with forward-predictive and hierarchical components, that we will evaluate against real neural data.
>
> ## Item (3a): Training the VAEs on more modern datasets
>
> > … not having evaluations on more challenging and larger benchmarks such as CIFAR or ImageNet raises concerns over computational efficiency and scalability
>
> This concern was shared by several reviewers. We believe that addressing this limitation will substantially improve the final paper. To this end, we performed experiments on three color-image datasets: ```CIFAR-10``` (3 x 32 x 32), ```tiny ImageNet``` (3 x 32 x 32), and cropped ```CelebA``` (3 x 128 x 128).
>
> The results, presented with tables in our response to Reviewer ```fucC```, confirm that iP-VAE readily scales up to more complex datasets, and consistently outperforms its amortized counterpart on ```CelebA```. (Once again, apologies for redirecting your attention to the other rebuttal - this should have been a global rebuttal response, but those are not allowed this year.)
>
> ## Item (3b): Why not report log-likelihood
>
> > … it's unclear why a rigorous metric such as log-likelihood (ELBO) of images is also left out … Mere reconstruction error only gives us half the picture
>
> We thank the reviewer for this important question. Our decision to focus on the reconstruction-sparsity trade-off (Figure 3) instead of a single log-likelihood value was a principled one, motivated by two main factors.
>
> **1. A more informative, disaggregated metric:**
>
> First, we follow the recommendation of [Alemi et al. (2018)](https://arxiv.org/pdf/1711.00464) in *Fixing a Broken ELBO*, who "strongly encourage future work to report rate and distortion values independently, rather than just reporting the log likelihood."
>
> Our $R^2$-sparsity plot is designed in this spirit. Reconstruction $R^2$ serves as a proxy for inverse distortion, while sparsity serves as a proxy for the inverse coding rate. This connection is grounded in the theory of Poisson models, where the KL-divergence (rate) term acts as a penalty that directly promotes sparsity (see Figure 10 for an empirical demonstration). This disaggregated view provides deeper insight into model behavior than a single ELBO score and also connects to practical benefits like hardware efficiency.
>
> **2. The challenge of comparing online vs. static ELBO:**
>
> Second, reporting a single, comparable log-likelihood for our online model is non-trivial. Unlike static models with one ELBO, our iterative approach generates a sequence of ELBOs, one for each time step (eq. 54; Figure 7). Averaging these is not directly comparable to the ELBO of a static model, making a fair benchmark against standard VAEs challenging.
>
> **Action plan:**
>
> That said, we agree that providing this data is valuable for completeness. In the revised paper, we will add an appendix section containing (1) time-averaged log-likelihood scores and (2) plots of the full time-dependent log-likelihood curves for our main models. This will provide a more complete picture, as requested.
>
>
> ## Item (4): Improving clarity
>
> > improve the clarity in terms of defining components of the generative models upfront
>
> We will clearly define the full set of generative model components upfront.
>
> ## Item (5): Discussing limitations
>
> > … too brief and doesn't sincerely inform the reader critically of the limitations … I would strongly encourage the authors to substantiate it in the main text … limitations of natural gradient descent and iterative updates as opposed to amortized … why these are "prescriptions" for more "brain-like" architecture …  most critically, reflect on their choice of evaluation
>
> We thank the reviewer for this detailed feedback. We agree that the limitations section should be more substantial, and will make the following changes in the final paper:
>
>
> - We will use the additional page to bring more of the critical discussion of limitations from Appendix D.5 into the main text.
> - Specifically, we will expand on the primary limitation of iterative inference (its computational cost versus its generalization benefits) as we discuss in Appendix D.5.1.
> - We will also add discussion clarifying in what sense our prescriptions are "brain-like," focusing on their emergent dynamics and representations.
> - Finally, regarding our choice of evaluation, we will create a new appendix section detailing our principled reasoning (as outlined in our response to item (3b) above), and will refer to it from the main text.
>
>
> ---
> ---
> ---
>
>
> > Seeing it as an image generative model, it's unclear how it performs compared to state of the art models such as diffusion models
>
> Thank you for pointing this out, which allows us to clarify our main focus. We are interested in brain-like inference dynamics, and view the generative capacity of the model only in service of inference (i.e., *"analysis-by-synthesis"*).
>
> That said, we agree that assessing the model's generative capacity is informative. We performed a qualitative generation experiment on MNIST and we detail the results in our response to Reviewer ```yjVk```.
> > Line 335: what are PCNs?
>
> Predictive Coding Networks. We will revise the main text to introduce them before using the acronym.
>
> > Re Figure 3, panel b: is this a reasonable metric? The gold-star — although I understand at first glance — is not only impossible but also ill-defined (no latent variables are ever active?).
>
> We believe that the metric is reasonable in this application, as its goal is to highlight the tradeoff between sparsity (proxy for energy use on neuromorphic hardware), and reconstruction loss $R^2$ (proxy for inverse distortion).
>
> Although the gold star is not practically achievable in most scenarios, in high artificial constructs where the data is constant (like a single image dataset), it is possible to achieve full sparsity with perfect reconstruction loss.

---

> ### Comment · Reviewer_UGdd · 2025-08-05
> **Follow-up comments and questions**
>
> Thanks for the detailed explanation and additional experiments. Below are some follow-up comments and questions on selected points. I have no more questions on other points as of now.
>
> **Re originality and significance of FOND**: Thank you for clarifying your intent with FOND. However, your description reinforces my concern that the “framework” largely repackages standard variational inference applied to a setting---pick a likelihood, an approximate posterior, a prior, then apply natural gradient descent and maximize ELBO. These are conventional steps, not a substantive new prescriptive framework. What’s worse, the setting here, which is “brain-like” dynamics, is so loosely defined (if defined at all) that I do not know how to use it "rigorously". FOND offers no concrete guidance on critical modeling choices such as distributions, architectures, posterior structure, or likelihood family, nor does it specify what variables should actually be modeled and why exactly these—spikes, membrane potentials, natural images, natural videos, behavior etc. These are nontrivial decisions in a vast design space, and without clear prescriptions, the framework provides little actionable constraint. Especially without principled evaluation benchmarks, such as using neural data (either for normative explanations or phenomenological fitting), the framework appears permissive enough to accommodate countless designs, making it unclear what it truly constrains or contributes beyond existing practice.
>
> **Re improving clarity**: I appreciate your willingness to provide model details upfront in the manuscript, but this does not currently clarify your models or help me assess my evaluation. Could you please address this explicitly in the rebuttal, referring to my question 1 in the original review?

---

> ### Author Response · Authors · 2025-08-06
>
> We sincerely thank the reviewer for the detailed follow-up and for deeply engaging with our work. These are important critiques that get to the core of our paper's contribution. We will address both points below.
>
> ## Re: Originality and Significance of FOND
>
> We appreciate the detailed explanation of your concerns. It helps us see that our use of the term "prescriptive framework" has caused confusion. You are correct that the individual steps (choosing distributions, using natural gradient descent, etc.) are established practice in variational inference.
>
> The novelty we aimed to convey with FOND is not in inventing these steps, but in **formalizing them into a specific derivational pathway** to generate and interpret neural network **dynamics**. The goal of FOND is to provide a bridge from the high-level, abstract principle of Free Energy minimization to concrete, testable neural circuit models.
>
> > FOND offers no concrete guidance on critical modeling choices such as distributions, architectures, posterior structure, or likelihood family, nor does it specify what variables should actually be modeled and why exactly these
>
> You rightly point out that "brain-like" is a vast design space and FOND does not constrain all possible choices. This is intentional. What FOND does is that it determines the **circuit dynamics equations** after a researcher makes a few, clearly defined modeling choices.
>
> Our paper then shows that making one set of specific choices (Poisson distributions, membrane potential parameterization) inevitably leads to the derivation of a canonical circuit model (a spiking, leaky-integrate-and-fire network with lateral competition). Here, the value is in demonstrating that this familiar circuit model can be **derived from first principles**, rather than **constructed descriptively**.
>
> To make the utility of FOND as a generative framework more concrete, we wish to briefly share two unpublished examples of other distinct models we are actively developing using its principles:
>
> 1. A **hierarchical** iP-VAE that includes top-down spatial modulations, and incorporates a **forward-predictive** component that learns to anticipate future states. This model goes beyond the current iP-VAE’s flat latent space and Dirac-delta transition probabilities.
> 2. An iterative Gaussian VAE with a compressive nonlinearity (e.g., a sigmoid) that, when trained on raw natural images, learns **center-surround** receptive fields: a different but equally canonical neural representation compared to the Gabor-like receptive fields learned by iP-VAE.
>
> Critically, all of these models, and their resulting dynamics equations, are derived within FOND.
>
> These additional examples illustrate our central point: FOND is a framework precisely because it can derive multiple, distinct dynamics. By making different principled choices, we can produce models with different dynamics equations that learn qualitatively different—but still biologically relevant—representations (e.g., center-surround vs. Gabor). Thus, a core value of FOND is that it provides a **unified theoretical lens** showing how these **diverse outcomes**, with **diverse dynamics**, originate from the **same principle** of free energy minimization.
>
> This reinforces that our paper's first contribution is the framework (or recipe) itself, with iP-VAE serving as one important (but not exclusive) demonstration of its utility as a separate contribution. We hope these concrete examples help clarify our intended contribution.
>
> ### Action plan
>
> We will, of course, incorporate this reasoning into the revised manuscript to make the framing clearer. We will also replace instances of **“prescriptive framework”** in the text with phrases like **“recipe”** and **“framework”** to more accurately present the contribution.
>
> ## Re improving clarity
>
> You are absolutely right, and we sincerely apologize for not addressing your Q1 in our initial rebuttal. This was an oversight. We will answer it explicitly now and will ensure this is clarified upfront in the revised manuscript.
>
> The core of the issue is that the posterior's dependence on $x$ is implicit in our iterative model. In the notation for the approximate posterior, $q_u(z|x) = \text{Pois}(z; r)$, the conditioning on $x$ happens via the membrane potentials $u$:
>
> - $u = u(x)$
> - $r = r(x) = \exp(u(x))$
> - $z(r(x)) \sim \text{Pois}(z; r(x))$
>
> The explicit dynamic update rule that defines the $u(x)$ relationship appears in **Equation 8**, which as you noted, comes much later in the manuscript. We agree this is too late and causes confusion.
>
> In the final version, we will make these dependencies explicit from the start. This includes clarifying that in **Equation 3**, the posterior membrane potentials $u$ (the red ones) are a function of $x$, while the prior ones $u_0$ are not. We will also correct our slight abuse of notation from $q_u(z|x)$ to the more accurate $q(z|u(x))$ to be perfectly clear.

---

> > ### Comment · Reviewer_UGdd · 2025-08-06
> > **Follow-up comments and questions II**
> >
> > I thank the authors again for the detailed explanations. These explanations certainly help better clarify the work, but I do have some follow-up questions that I believe will further help clarify the paper's position.
> >
> > **Re originality and significance of FOND**: From my perspective, when modeling with the ELBO as the objective, the hard part lies in selecting the distributions and functional forms. Your response confirms that these choices are left to the scientist by design. Once they are made, what remains is specifying how those choices are used to maximize the ELBO or minimize free energy.
> >
> > Am I correct in understanding that FOND’s primary prescription occurs at this stage—specifically, the use of online natural gradient descent with Fisher preconditioning? If so, this is not entirely clear from the main text, especially since you also prescribe the use of membrane potential and, later, the definition of firing rate (Lines 167–169 and the paragraph titled "Prescriptive choices", and also Line 45 where you mention that specifying $q_\lambda$ is a prescriptive choice).
> >
> > This raises a key question (as I noted under **Limitations** in my original review): Why use natural gradients of free energy, and why apply them in an online setting? Additionally, why iterative inference? You identify these as the main “prescriptions” beyond the model components, but I do not see a clear first-principles justification for why either step would produce more “brain-like” dynamics. As I understand it, the message is: choose your prior, posterior, and likelihood freely, then make inference “brain-like” by deriving dynamics via natural gradients, applied online. How much does this specific choice of update rule meaningfully contribute to biological plausibility? And are established approaches, the more de-facto choices in mainstream machine learning—such as using KL divergence directly, standard gradient descent, adaptive optimizers like Adam—explicitly ruled out in learning the parameters of the specified distributions? If so, please clarify the rationale and the first principle reasons for doing so.
> >
> > Regarding the two unpublished examples: while I am genuinely enthusiastic about going through them, they cannot factor into my evaluation, as the current submission must stand on its own.
> >
> > **Re improving clarity of model components**: I follow the generative model much more clearly now. Am I correct in understanding that the posterior over $u$ is linear w.r.t $x$? Do you retain this relationship when you make the likelihood (decoder) nonlinear as well?

---

> ### Author Response · Authors · 2025-08-06
>
> We genuinely appreciate your deep engagement, and the follow-up. This discussion is helping us understand and clarify our paper's contributions in real-time (i.e., *online*).
>
> ## Re originality and significance of FOND:
>
> > From my perspective, when modeling with the ELBO as the objective, the hard part lies in selecting the distributions and functional forms.
>
> We agree. This is something that is not discussed as openly and frequently as it should. For example, one can read free energy papers and leave with the impression that this literature is exclusively about Gaussian distributions.
>
> Our paper tries to clearly outline what choices are to be made (lines 91-93), and that Gaussians are not mandatory (lines 1282-1285). We then demonstrate how alternative choices like Poisson lead to relatively more biologically plausible outcomes.
>
> > Am I correct in understanding that FOND's primary prescription occurs at this stage---specifically, the use of online natural gradient descent with Fisher preconditioning?
>
> Yes, that is exactly right. We state this in lines 165-166:
>
> *"The key prescription in FOND is that once the dynamic variables are specified, their dynamics are governed by natural gradient descent on $\mathcal{F}."*
>
> ...and further reinforce it later in the manuscript, lines 196-197:
>
> *"FOND prescribes that neural dynamics follow natural gradients of the free energy"*
>
> > If so, this is not entirely clear from the main text, especially since you also prescribe the use of membrane potential and, later, the definition of firing rate … also Line 45 …
>
> We fully agree. Your feedback helped us realize that our original "Prescriptive choices" paragraph confusingly bundled this core rule of FOND with the specific modeling choices made for iP-VAE, without specifying that some of those are about FOND (lines 163-166), while others are about deriving iP-VAE (lines 167-172).
>
> While the second set of prescriptions about iP-VAE start with *"For a neurally plausible model…"* (line 167), we agree that we should have delineated this more clearly.
>
> Thank you for pointing this out. In the revisions, we will clearly distinguish FOND's prescription (fixed) with the other prescriptive choices (to derive iP-VAE).
>
>
> > This raises a key question … Why use natural gradients of free energy, and why apply them in an online setting? Additionally, why iterative inference?
>
> Thank you for these questions. They get to the heart of our work. We provide detailed justifications for these three core prescriptions in a new official comment addressed more globally (right below the abstract).
>
> ## Re the two unpublished examples
>
> Thank you for your enthusiasm. We mentioned those only to make a single point concrete: that FOND (the framework) is a contribution distinct from specific models (iP-VAE, those two examples, etc.) that can be derived and understood within FOND.
>
> ## Re improving clarity of model components:
>
> > the posterior over $u$ is linear w.r.t $x$?
>
> Yes, that is correct for a single update step, even when the decoder is nonlinear:
>
>
> - In the linear decoder case, the update to $u$ depends on $\Phi^T x$
> - In the nonlinear case, this is replaced by $\mathbf{J}[z_t] x$, where $\mathbf{J}[z_t]$ is the decoder's Jacobian, evaluated at spike samples, $z_t$, drawn from the current prior.
>
> In both cases, the update to $u$ is a linear function of the input $x$, where the transformation is either fixed ($\Phi^T$) or state-dependent ($\mathbf{J}[z_t]$). This is shown in the legends (rightmost panels) in Figures 5 and 6, but we will clarify this in the text in the revision.

---

> > ### Comment · Reviewer_UGdd · 2025-08-07
> >
> > I thank the authors for their detailed responses. I do not have follow-up comments or questions as of now.

---

### Official Review · Reviewer_iEYy · 2025-07-02

**Clarity:** 2
**Significance:** 3
**Originality:** 3
**Rating:** 5
**Confidence:** 3

**Summary:**

This paper introduces FOND (Free-energy Online Natural Gradient Dynamics), a framework for deriving inference procedures as brain-inspired dynamics from online natural gradient descent on variational free energy. Implementing FOND with Poisson latent variables yields the iterative Poisson VAE (iP-VAE), a recurrent spiking network whose dynamics implement iterative variational Bayesian inference. Empirical evaluation demonstrates that iP-VAE achieves superior reconstruction–sparsity trade-offs compared to amortized VAEs and standard sparse/predictive coding baselines on small-scale benchmarks, with robust out-of-distribution performance.

**Questions:**

- How sensitive are the reported performance metrics (reconstruction error and sparsity) to the dimensionality of the latent variable space (K)?

- Would replacing natural gradient updates with standard gradient descent significantly affect the observed performance?

**Ethical Concerns:**

["NO or VERY MINOR ethics concerns only"]

**Final Justification:**

I thank the authors for appropriately addressing my points. I believe this is a good contribution and thus will increase my score and recommend acceptance.

**Limitations:**

The authors give an adequate explanation of the works limitation in appendix D.5. A concise overview of these points should be brought into the main paper for a final version.

**Quality:**

3

**Strengths And Weaknesses:**

Strengths:

- The paper is technically sound and will have broad appeal to researchers across machine learning and theoretical neuroscience.

- The work successfully unifies multiple influential frameworks (variational inference, sparse coding, natural gradient descent) within a coherent theoretical framework, which is further used to derive novel architectures.

- While the empirical results presented in the main paper are limited in scope (i.e., very simple datasets), the results regarding sparsity and OOD performance are compelling.

- The conceptual integration of online natural gradient descent, variational inference, and biologically plausible neural dynamics is insightful and (to my knowledge) novel.


Weaknesses:

- While the writing is clear, the structure of the manuscript could be improved for readability. The main text frequently cross-references an extensive appendix, splitting core methodological details across sections and appendices.

- A quantitative comparison of runtime (training and inference) between the iterative and amortized algorithms is lacking.

- A concise algorithmic description or pseudocode detailing the iterative algorithm steps would significantly enhance clarity.

---

> ### Author Rebuttal · Authors · 2025-07-31
>
> We sincerely thank Reviewer iEYy for their constructive feedback and encouraging evaluation. We appreciate that they found our work to be a novel and insightful unification of key frameworks, with broad appeal and compelling empirical results (although "limited in scope").
>
> Below, we address the scope, other concerns and questions, and the comment regarding limitations.
>
> ---
> ---
> ---
>
> > While the writing is clear, the structure of the manuscript could be improved for readability. The main text frequently cross-references an extensive appendix, splitting core methodological details across sections and appendices.
>
> We will restructure the manuscript and make the contents of the main paper more coherent and self-contained. Specifically, given the additional 1 page, we plan to bring all the necessary details for deriving the main theoretical result into the main paper. This should further enhance the clarity of the core derivations and methodological details.
>
> ---
> ---
> ---
>
> > A quantitative comparison of runtime (training and inference) between the iterative and amortized algorithms is lacking.
>
> We agree with the reviewer. A significant challenge with iterative methods is their slower training and inference time, and it is important to provide a quantitative measure of runtime.
>
> #  Runtime comparison between iterative and amortized VAEs
>
> Here, we will focus only on comparing iP-VAE and P-VAE runtimes. For the final revised manuscript, we will also explore the Gaussian and Gaussian + ReLU models.
>
> ## Training time comparison
>
> We partially address this in **Appendix C.1.4 (Training compute details)**, lines 1894-1898:
>
> *"The computational requirements for iterative VAEs scale with the number of inference iterations ($T_\text{train}$). For example, training the models presented in Fig. 2 with $T_\text{train} = 16$ requires approximately three hours per model on an NVIDIA A6000 GPU. Memory usage and training time both scale approximately linearly with $T_\text{train}$, as each additional inference iteration requires maintaining the computational graph for backpropagation."*
>
> But as the reviewer pointed out, we do not compare to the amortized counterparts. To address this, we examined the P-VAE counterpart which achieves the best overall performance in Figure 3. This model uses a five-layer ResNet (lines 1864-1865), but is otherwise identical to the aforementioned iP-VAE.
>
> This amortized P-VAE takes ~1.5 hours to train (versus ~3h for iP-VAE).
>
> Therefore, at least for the van Hateren dataset, the amortized training time is roughly 2x faster. We expect this gap to widen with (1) increased $T_\text{train}$, and (2) dataset scale. For the final paper, we will explore this question more thoroughly and quantify relative training times as a function of various hyperparameters and datasets.
>
> ## Inference time comparison:
>
> To quantify this, we looked at how many test-time iterations does it take for iP-VAE to outperform its P-VAE counterpart, in terms of both sparsity and reconstruction scores. To this end, we used the distance from the ideal gold star ($R^2$ = 1, Sparsity = 1; introduced in Figure 3).
>
> We found that at only 13 iterations, iP-VAE surpasses the performance of the amortized P-VAE.
>
> In conclusion, while iterative inference is slower, it achieves superior performance in a relatively small number of steps. For the final paper, we plan to perform more comprehensive comparisons, calculate actual inference runtimes in seconds, and include them in a dedicated appendix section.
>
> ---
> ---
> ---
>
> > A concise algorithmic description or pseudocode detailing the iterative algorithm steps would significantly enhance clarity.
>
> We agree. For the revised manuscript, we plan to add both an algorithm box and a graphical model to clearly and unambiguously describe the full iterative algorithm.
>
> ---
> ---
> ---
>
>
> > How sensitive are the reported performance metrics (reconstruction error and sparsity) to the dimensionality of the latent variable space (K)?
>
> Thank you for this suggestion. We performed additional experiments, exploring latent dimensionalities $K \in \[32, 64, 128, 256, 512, 768, 1024, 2048\]$. We report $R^2$, sparsity (i.e., portion_zeros), and the overall performance as measured by the Euclidean distance from the gold star (i.e., $R^2$ = 1, Sparsity = 1; introduced in Figure 3). We also report the average total number of active neurons, defined as K * (1 - portion_zeros).
>
> Here are the results:
>
> | Latent dim (K) | $R^2$ ↑ | Portion zeros ↑ | Overall perf. ↓ | Avg. num. active neurons |
> |---------|------|---------------|---------------------|--------------|
> | 32 | 0.24 | 0.28 | 1.05 | 23.2 |
> | 64 | 0.42 | 0.28 | 0.93 | 46.4 |
> | 128 | 0.64 | 0.32 | 0.77 | 87.4 |
> | 256 | 0.77 | 0.55 | 0.51 | 115.6 |
> | 512 | 0.83 | 0.77 | 0.28 | 116.1 |
> | 768 | 0.83 | 0.86 | 0.22 | 110.4 |
> | 1024 | 0.83 | 0.90 | 0.20 | 106.9 |
> | 2048 | 0.84 | 0.95 | 0.17 | 103.1 |
>
> We observe two key results:
>
> - First, the overall performance consistently improves as the dictionary becomes more overcomplete, corroborating a known desirable feature of sparse coding models.
> - Second, and more surprisingly, the total number of active neurons plateaus around ~110, even as the latent space grows to 2048 dimensions. This suggests the model learns an efficient code that uses a consistent budget of active neurons, regardless of the available capacity.
>
> We thank the reviewer for this excellent question, which revealed these new results. We will include a full analysis in the appendix of the final paper.
>
> ---
> ---
> ---
>
>
> > Would replacing natural gradient updates with standard gradient descent significantly affect the observed performance?
>
> Yes. We explored this in our early experiments and found that standard gradient descent updates, which do not account for the geometry of the parameter space, are unstable. Without the Fisher preconditioning provided by natural gradients, the updates are poorly scaled. This caused the inference dynamics to quickly diverge.
>
> We will add a sentence stating this in the revised manuscript to further highlight the importance of natural gradients for our framework.
>
> ---
> ---
> ---
>
>
> > The authors give an adequate explanation of the works limitation in appendix D.5. A concise overview of these points should be brought into the main paper for a final version.
>
> We agree. With the additional 1 page for the final version, we plan to more clearly and explicitly discuss limitations.

---

> > ### Comment · Reviewer_iEYy · 2025-08-08
> >
> > I thank the authors for appropriately addressing my points. I believe this is a good contribution and thus will increase my score and recommend acceptance.

---

### Author Response · Authors · 2025-08-06
**Three key prescriptions of FOND**

This brief section grew out of our insightful and rich discussion with Reviewer `UGdd`. We decided to share it here for a broader visibility.

FOND prescribes that neural dynamics (1) follow **natural gradients** of the free energy, (2) in an **online inference** setting, (3) through an **iterative** procedure.

Below, we provide a first-principles justification for each of these core prescriptions of FOND.

## **Why natural gradients?**

This prescription is "mathematics-inspired" rather than directly "brain-inspired." Natural gradient descent is, formally, the path of *steepest descent* in the space of probability distributions ([Amari (1998)](https://doi.org/10.1162/089976698300017746)). By steepest descent, we mean following the direction that minimizes free energy most steeply. That is, achieving the greatest reduction in free energy per unit of distance traveled, with distance measured using the Fisher information metric ([Martens (2020)](https://jmlr.org/papers/v21/17-678.html)).

Through adopting natural gradients, we ensure our dynamics follow the most efficient path to minimize free energy, providing a principled foundation for the optimization process.

## **Why online inference?**

This is a crucial prescription for modeling adaptive systems like the brain, as static inference cannot account for phenomena where recent history shapes current belief. We provide two examples:

1. **Low-level visual perception:** In `serial dependence`, perception is systematically biased by recently seen stimuli ([Fischer & Whitney (2014)](https://www.nature.com/articles/nn.3689)). This requires a prior that evolves online as sensory evidence streams. This time-evolving mechanism is unavailable to static models that would process each stimulus identically and independently.

2.  **High-level cognitive updating:** This principle extends to higher-level cognition, such as `learned helplessness`, where an organism’s beliefs about its own agency are updated in real-time by experience ([Seligman & Maier (1967)](https://doi.org/10.1037/h0024514), see also this ~7 min [YouTube video](https://www.youtube.com/watch?v=gFmFOmprTt0) for a demonstration). Such rapid belief updating can be mathematically formalized as an evolving prior *$u$* in our model.

These phenomena demonstrate that online inference is a fundamental principle of biological intelligence across multiple timescales. In contrast, static models struggle to account for how past experience continuously reshapes ongoing processing.

## **Why iterative inference?**

This prescription is motivated by the computational nature of inference itself. According to the *"analysis-by-synthesis"* idea, perception is an inherently iterative, error-correcting loop: a hypothesis generates a prediction (synthesis), this is compared to data (analysis), and the error refines the hypothesis. This is precisely what iterative inference does. Amortized inference, by contrast, is an open-loop guess that can be brittle, especially for novel inputs.

This computational view is strongly supported by neuroscience. Decades of research show that the brain's initial *feedforward sweep* is followed by waves of *recurrent processing* that are critical for robust perception. In fact, recent work argues that **recurrence is required** to capture the brain's representational dynamics and successfully perform challenging recognition tasks. For example:

- Macaque visual cortex takes an **additional ~30 ms of recurrent processing** to recognize "challenge" images ([Kar et al. (2019)](https://www.nature.com/articles/s41593-019-0392-5)),
- Neural code for faces in macaques dynamically evolves over hundreds of milliseconds—a process termed "**code switching**" ([Shi et al. (2023)](https://www.biorxiv.org/content/10.1101/2023.12.06.570341v2)).

Our own empirical results further validate this principle: we found that iterative inference clearly and unambiguously outperforms amortized inference, leading to more adaptive algorithms that better generalize out-of-distribution.

Therefore, the prescription for iterative inference is motivated by a convergence of evidence:

1. It is the **natural algorithmic form of analysis-by-synthesis**,
2. Neuroscience shows it is **essential for robust perception**, and
3. Our **empirical results** confirm it leads to more generalizable models.

While the brain likely employs a hybrid iterative-amortized strategy to achieve both accurate and fast inferences, these principles justify iteration as a core component of any model aiming for brain-like inference.

## **Action plan**

We will further substantiate these first-principles justifications and include them in a dedicated Appendix section. We thank Reviewer `UGdd` once again for their deep reading of our paper and their questions, which pushed us to clarify our contributions.

---

> ### Comment · Reviewer_UGdd · 2025-08-07
> **On the prescriptions of FOND**
>
> I thank the authors for their thoughtful and detailed explanations. They clarify the intent and scope of the work better, and they strengthen the overall contribution.
>
> Since FOND is positioned as a core contribution offering “prescriptions motivated by first principles,” I would encourage the authors to clearly articulate why these particular prescriptions are justified and what those first principles are—just as they have done in this rebuttal. While I understand the plan to include this in the appendix, I believe it would be more impactful to present this reasoning in the main text, given its importance. As reviewers iEYy and yjVK have also noted, a large part of the main text is currently devoted to reviewing prior work which could be trimmed and reallocated to this more central motivation, which would help readers better understand the "why" behind FOND upfront.
>
> Lastly, I would strongly suggest framing FOND not as a fully prescriptive or definitive framework, but rather as a promising, brain-and first-principles-inspired approach. This would help set appropriate expectations while highlighting the genuine value of the contribution. I believe such positioning would resonate more effectively with readers and enhance the clarity of the paper’s message.

---

> > ### Author Response · Authors · 2025-08-07
> >
> > We are genuinely grateful for your detailed follow-up and constructive guidance. This discussion has been incredibly helpful for us. It pushed us to articulate the core principles behind our work, culminating in the justifications we presented in our last response above.
> >
> > Following your advice, we will make two key changes to the final manuscript:
> >
> > - We will move the **"first-principles" justifications** for our core prescriptions (natural gradient, online, iterative) to a more central position in the **main text**, making space by shifting some background material to the appendix (in doing so, simultaneously addressing similar points raised by Reviewers `iEYy` and `yjVK`).
> > - We will soften the framing, adopting your suggestion to present FOND as a **"promising, brain- and first-principles-inspired approach"** rather than a "fully prescriptive or definitive" framework.
> >
> > Thank you again for your rigorous and thoughtful engagement. This will substantially improve the paper.

---

### Author Response · Authors · 2025-08-08
**BONG and FOND are related, but distinct, frameworks**

We are following up on the **FOND/BONG similarity question** to provide a final, clarifying analysis. As can sometimes happen in parallel scientific development, we developed our FOND framework before we were aware of BONG. When we discovered BONG while writing the paper, we cited it for its clear conceptual similarities. However, we recognize now that we did not sufficiently articulate the significant technical differences between the two approaches in our manuscript.

The insightful comments from Reviewers `fucC` and `yjVk` prompted us to conduct a much more thorough analysis of the BONG paper. This has led to a new clarity, and we are now convinced that the two frameworks are fundamentally distinct in their goals, scope, and, most critically, their technical approach.

We have identified three high-level conceptual differences:

1. **Different goals:** BONG is a rule for learning the parameters $\theta$ of a Bayesian Neural Network. In contrast, FOND is a recipe for deriving the inference dynamics of variational parameters (e.g., membrane potentials, $u$).
2. **Different settings:** BONG is optimizing neural network parameters for supervised learning. FOND is about deriving neural dynamics in unsupervised settings.
3. **Different scopes:** BONG's derivations and approximations are specific to Gaussian approximate posteriors. FOND is distribution-agnostic, allowing for other posteriors like the Poisson distribution used to derive iP-VAE (but also Gaussians for iG-VAE).

These conceptual differences stem from a **fundamental technical divergence** in how each framework handles the gradient of the expected log-likelihood term in the objective:

- **BONG's approach:** By restricting itself to **Gaussian approximate posteriors**, BONG can leverage specific mathematical results (e.g., Price's and Bonnet’s Theorems) to find an analytical expression for the gradient of the expectation in their Equation 3. They apply these theorems to derive Gaussian-specific update rules shown in their Equations 9 and 10 ([link to BONG paper](https://openreview.net/pdf?id=E7en5DyO2G)). Given this setup, the main technical contribution of BONG is then to explore various ways to approximate those *Gaussian-specific* expressions.
- **FOND's approach:** Because FOND must be distribution-agnostic, it cannot use these Gaussian-specific theorems. Instead, it follows the standard and more general approach used in the VAE literature: first, approximate the expectation with a Monte Carlo sample, and *then* compute the gradient of the resulting stochastic objective.

This is beyond just a different "flavor" of the same algorithm. Rather, they are different mathematical procedures.

One direct consequence is that **a spiking model like iP-VAE, with its Poisson approximate posterior, cannot be derived within the BONG framework**. The theorems BONG relies upon do not apply to the Poisson distribution, whereas FOND's more general gradient estimation technique handles it naturally.

We are grateful to the reviewers (especially `yjVk`) for pushing us on this point. This deeper analysis has been invaluable, and it will allow us to precisely and confidently articulate the relationship between these frameworks in our revised manuscript.

---

### Note · Authors · 2025-08-11

We are grateful to the reviewers for the deep and constructive discussions that have significantly clarified our paper's main contributions. We would like to summarize the two key outcomes for the AC's consideration.

### **1. Clarified theoretical contribution (FOND):**

The discussions prompted us to articulate that our primary theoretical contribution is FOND, a recipe for deriving brain-like neural dynamics from first principles. We clarified that **FOND is distinct from related work like BONG** in its:

- **`Goals`** (inference dynamics vs. parameter learning),
- **`Scope`** (unsupervised vs. supervised; distribution-agnostic vs. Gaussian-specific), and
- **`Technical approach`** (in approximating the expected log-likelihood).

Additionally, our final rebuttals provide a detailed, first-principles justification for FOND's core prescriptions—the use of **`online`**, **`iterative`**, and **`natural-gradient`**-based inference—to model adaptive biological intelligence.

###  **2. Strengthened empirical contribution (iP-VAE):**

FOND's utility is demonstrated through the derivation of iP-VAE, a novel spiking VAE. In direct response to reviewer feedback, we conducted **new experiments on complex color-image datasets** (**`CelebA`**, **`CIFAR-10`**, **`tiny ImageNet`**). These new results show that our iterative model consistently and significantly outperforms its powerful amortized counterpart, addressing initial concerns about the scope and scalability of our evaluation.

### **Summary**

Our work offers a dual contribution:

1. a **theoretical recipe** for deriving principled neural models; and
2. a high-performing model family supported by **strong empirical evidence**, both:
    - **existing** (sparsity-reconstruction trade-off, OOD generalization) and
    - **new** (scalability to complex, color-image datasets).

We thank the reviewers and AC for their time and guidance, which will substantially improve the final manuscript.

---

### Decision · Program_Chairs · 2025-09-17

**Decision:**

Accept (poster)

**Comment:**

This paper introduces FOND, a principled recipe for deriving brain-like inference dynamics as online natural-gradient descent on free energy, and demonstrates it via iP-VAE, a spiking recurrent network that performs iterative variational inference. The model shows appealing theoretical properties—such as emergent normalization and efficient spike-based representations—and strong empirical results, including favorable sparsity–reconstruction trade-offs, robustness to out-of-distribution inputs, and scaling to larger datasets (CelebA, CIFAR-10, tiny ImageNet). The work bridges machine learning and neuroscience by turning abstract free-energy principles into concrete neural circuit dynamics.

The main strengths lie in its conceptual clarity and unification, the empirical effectiveness of iP-VAE, and the broader relevance to both ML and neuroscience communities. Reviewers appreciated the principled derivation and the improved generalization compared to amortized baselines. The weaknesses are that FOND is framed more as a “recipe” than a fundamentally new algorithm, biological plausibility is supported only indirectly (without direct neural data evaluation), and initial experiments were limited in scope (later addressed with new scaling results).

During rebuttal, the authors provided substantial clarifications distinguishing FOND from BONG/BLR, offered first-principles justifications for online natural-gradient inference, and added runtime, sensitivity, and scalability experiments. While some concerns about originality and biological validation remain, the paper is technically solid, intellectually stimulating, and strengthened by rebuttal results. Overall, I recommend acceptance as a poster, as the contributions will be of clear interest to both machine learning and neuroscience audiences.